# A Likely Geometry of Generative Models

## Abstract

The geometry of generative models serves as the basis for interpolation, model inspection, and more. Although certain generative models admit an implicit geometric structure, there is no broadly applicable framework that captures a principled notion of geometry across generative models without imposing restrictive assumptions on the model class or data dimensionality. In this paper, we show how to equip generative models with a general geometry compatible with different metrics and probability distributions to analyze generative models. Our method does not require additional training. We consider curves analogous to geodesics constrained to a suitable data distribution aimed at targeting high-density regions learned by generative models. We formulate this as a (pseudo-)metric and prove correspondence to a Newtonian system on a Riemannian manifold. We show that shortest paths can here be characterized by a system of ordinary differential equations, which, along the optimal path, locally correspond to geodesics under a suitable Riemannian metric. Numerically, we derive a novel algorithm to efficiently compute interpolation and generalized Fréchet means. Quantitatively, we show that curves using our metric traverse regions of higher likelihood areas than baselines across a range of models and datasets.

## 1 Introduction

Generative models learn the data distribution but rarely specify how to compute statistics such as interpolation and means on the generated samples. Such actions require a *model geometry*. However, geometries are commonly defined *ad hoc*, e.g., through spherical and linear interpolation (Song & Ermon, 2020; Song et al., 2021a; Zheng et al., 2024). Although these interpolation schemes are simple, they assume that the prior distribution is isotropic Gaussian, where spherical interpolation also assumes that the space is high-dimensional. This does not generalize well across generative models.

Other works equip generative models with a Riemannian metric to model the data manifold (Tosi et al., 2014; Arvanitidis et al., 2018; Shao et al., 2018; Karczewski et al., 2025; Béthune et al., 2025), but this geometry is specified on a *per-model* basis. In general, Riemannian manifolds provide an operational framework for the analysis of generative models (Hauberg, 2019) and naturally appear in latent space models (Shao et al., 2018; Arvanitidis et al., 2018; Hauberg, 2019). For Riemannian manifolds, interpolation corresponds to *geodesics*, which are locally-length minimizing curves. Geodesics can be locally characterized by an ordinary differential equation (ODE) (do Carmo, 1992, Lemma 2.3)

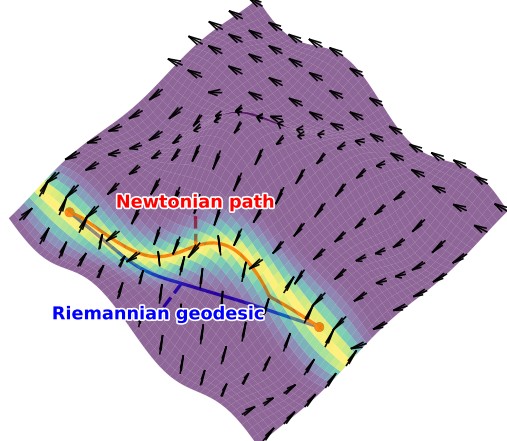

Figure 1: A conceptual illustration of our method, where we consider the geometry corresponding to a Newtonian system on a Riemannian manifold. The gradient vector field of the density "pushes" the Riemannian geodesic to areas of high likelihood.

$$\ddot{\gamma}^k(t) + \Gamma_{ij}^k\left(\gamma(t)\right)\dot{\gamma}^i(t)\dot{\gamma}^j(t) = 0, \quad \Gamma_{ij}^k = \sum_{kl}\frac{1}{2}g^{kl}\left(\frac{\partial g_{jl}}{\partial x^i} + \frac{\partial g_{ij}}{\partial x^j} - \frac{\partial g_{il}}{\partial x^l}\right). \tag{1}$$

where $\left\{\Gamma_{ij}^k\right\}_{ijk}$ denotes the Christoffel symbols derived from the Riemannian metric $g$ with inverse $\{g^{kl}\}_{kl}$. However, geodesics for generative models are not always within high likelihood regions of the data distribution, and for geodesics to have high likelihood, it requires imposing restrictions on the underlying generative model to incorporate the uncertainty of the data into the metric (Hauberg, 2019; Tosi et al., 2014).

Rather than modeling the geometry of the data manifold as a Riemannian manifold, we show in this paper how to construct a force field that points to regions of high-likelihood as illustrated in Fig. 1. We show that enforcing geodesics to be within high-likelihood areas does not correspond to a Riemannian metric but instead corresponds to a Newtonian system on a Riemannian manifold. Then, shortest curves are given by (Cariñena & Muñoz-Lecanda, 2023)

$$\ddot{\boldsymbol{\gamma}}^k(t) + \Gamma_{ij}^k\left(\boldsymbol{\gamma}(t)\right)\dot{\boldsymbol{\gamma}}^i(t)\dot{\boldsymbol{\gamma}}^j(t) = \nabla_x^k S(x), \tag{2}$$

where $\nabla_x^k$ is the $k$th element of the Riemannian gradient of a function $S : \mathcal{M} \to \mathbb{R}$. We start by stating the requirements for geodesics that they should be restricted to high-likelihood areas and formulate this as a (pseudo-)metric to define the notion of interpolation and a mean value. We further show that along the shortest curve, our approach can locally be seen as a geodesic under a Riemannian metric. We derive a novel algorithm to compute interpolation and means using our geometric framework and prove that our algorithm converges to a local minimum and has local asymptotic quadratic convergence under sufficient regularity, enabling fast computation of statistics for generative models. Empirically, we demonstrate that our approach obtains curves with a higher likelihood compared to baselines on various datasets and generative models.

## 2 Background and related work

**Riemannian manifolds** provide an operational framework suitable for analyzing data and generative models (Hauberg, 2019) and appear naturally in latent space models like the variational-autoencoder (VAE) (Shao et al., 2018; Arvanitidis et al., 2018). A Riemannian manifold can be viewed as a smooth $n$-dimensional surface in a sufficiently high-dimensional Euclidean space (Nash, 1954; Whitney, 1936). In practice, rather than working in the ambient Euclidean space, computations are performed intrinsically in local coordinates of the manifolds using the Riemannian metric. For example, the sphere and torus admits embeddings in $\mathbb{R}^3$, yet all computations can naturally be carried out in two-dimensional coordinate charts of the manifold. Formally, a Riemannian manifold is a differentiable manifold $\mathcal{M}$, equipped with a Riemannian metric, in the sense that it defines a smoothly varying inner product $g : T_{\boldsymbol{x}}\mathcal{M} \times T_{\boldsymbol{x}}\mathcal{M} \to \mathbb{R}$. The inner product is a quadratic form $g_{\boldsymbol{x}}(\boldsymbol{v}, \boldsymbol{w}) = \boldsymbol{v}^\top \boldsymbol{G}(\boldsymbol{x})\boldsymbol{w}$, where $\boldsymbol{v}, \boldsymbol{w} \in T_{\boldsymbol{x}}\mathcal{M}$ denote elements in the tangent space of $\mathcal{M}$ in $\boldsymbol{x} \in \mathcal{M}$ and $\boldsymbol{G}$ is a symmetric and positive definite matrix known as the metric matrix function (do Carmo, 1992, Page 38). The tangent space at $\boldsymbol{x} \in \mathcal{M}$ is a vector space that consists of the tangent to all curves at $\boldsymbol{x}$. The Riemannian metric gives rise to the notion of curves that locally minimize the Riemannian distance:

$$\text{dist}(\boldsymbol{a}, \boldsymbol{b}) := \min_{\boldsymbol{\gamma}} \int_0^1 \sqrt{\dot{\boldsymbol{\gamma}}(t)^\top \boldsymbol{G}(\boldsymbol{\gamma}(t))\dot{\boldsymbol{\gamma}}(t)}\, \mathrm{d}t,$$

where $\dot{\boldsymbol{\gamma}}$ denotes the time-derivative of a smooth curve $\boldsymbol{\gamma}$ on $\mathcal{M}$ with $\boldsymbol{\gamma}(0) = \boldsymbol{a} \in \mathcal{M}$ and $\boldsymbol{\gamma}(1) = \boldsymbol{b} \in \mathcal{M}$. Geodesics are locally length minimizing and can be computed by minimizing the energy functional

$$\mathcal{E}(\boldsymbol{\gamma}) = \frac{1}{2}\int_0^1 \dot{\boldsymbol{\gamma}}(t)^\top \boldsymbol{G}\left(\boldsymbol{\gamma}(t)\right)\dot{\boldsymbol{\gamma}}(t)\, \mathrm{d}t, \tag{3}$$

or equivalently solving the Euler-Lagrange equations in Eq. 1. In general, we will assume that the manifolds studied in this paper are *geodesically complete* in the sense that between any two points $\boldsymbol{a}, \boldsymbol{b} \in \mathcal{M}$, there exists at least one length-minimizing geodesic connecting the boundary points. Simple examples of Riemannian manifolds are the sphere used in climate data (Karpatne et al., 2019; Mathieu & Nickel, 2020) and the $n$-dimensional torus used in protein modeling (Lovell et al., 2003; Murray et al., 2003). More abstract examples of Riemannian manifolds are learned manifolds using latent space models as the VAE, where the decoder under sufficient regularity learns a smooth $d$-dimensional immersion (Shao et al., 2018). For a further discussion on Riemannian geometry and generative models, we refer to (Hauberg, 2025).

**Generative models** aim to approximate the underlying data distribution. Although this objective is mostly shared across generative models, different models solve this problem in very different ways. VAE's learn the data distribution by maximizing a lower bound of the likelihood (Kingma & Welling, 2014), autoregressive models model the conditional distribution (Uria et al., 2016), normalizing flows use the change of variables theorem (Papamakarios et al., 2021), while score-based diffusion models learn the data distribution using the probability flow ODE by learning the score function (Song et al., 2021b).

All of these generative models can generate samples that approximate the data distribution, but do not allow elaborate statistics on the generated samples, such as interpolation, mean, and so on. Different works equip generative models with a geometry, but this is typically based on strong assumptions about the data dimension and the generative model. Shao et al. (2018); Arvanitidis et al. (2018); Wang & Ponce (2021) propose using the learned metric in the latent space for the VAE (Kingma & Welling, 2014) and the generative-adversarial network (GAN) (Goodfellow et al., 2014) to compute geodesics for interpolation and extraction of model information. Although these approaches provide insight into latent space models, they assume a latent space equipped with a Riemannian metric, and it has been observed that for the geodesic to have a high likelihood with respect to the data distribution, it requires incorporating data uncertainty directly into the generative model (Hauberg, 2019; Tosi et al., 2014; Kalatzis et al., 2020; Arvanitidis et al., 2022).

Diffusion models do not immediately learn a latent representation of the data, and the above constructions do not apply. Rather than applying the ambient Euclidean geometry, diffusion models usually compute interpolation using spherical and linear interpolation in the noise distribution that arises as the limit of the forward dynamics (Song et al., 2021b; Song & Ermon, 2020; Du & Mordatch, 2019), where Zheng et al. (2024); He et al. (2024); Yang et al. (2024); Guo et al. (2024) exploits the specific structure of diffusion models to generate highly realistic transitions between samples. These interpolants are based on strong assumptions of diffusion models and do not generalize to other generative models, such as Riemannian diffusion models, due to their ad hoc nature (Bortoli et al., 2022; Huang et al., 2022; Jo & Hwang, 2024). Alternatively, Wang & Golland (2023); Yang et al. (2025) train a diffusion model to generate realistic transitions between data, but the approach requires additional training and does not directly relate to any underlying generative model. In general, non-Riemannian methods do not extend beyond interpolation to support statistical measures such as the means or principal components. This limits their use in post hoc data analysis, which is often the long-term goal of geometric constructions.

Other works equip diffusion models with a Riemannian structure (Yu et al., 2025; Karczewski et al., 2025; Saito & Matsubara, 2025; Béthune et al., 2025), where Karczewski et al. (2025) considers an information geometric approach inspired by the Fisher-Rao metric. Lebanon (2002); Hartmann et al. (2022); Yu et al. (2025); Béthune et al. (2025) construct a generic Riemannian metric for generative models based on a probability density function, where Béthune et al. (2025) considers a conformal Riemannian metric designed for energy-based models. Rather than incorporating the probabilistic structure into the Riemannian metric, we opt to consider the metric structure of the data used in training and penalize with a scalar field. Not only do we find that this gives better results compared to baselines but also reduces computational costs.

## 3 Our likely geometry

In this section, we describe how to equip generative models with our geometric framework. We do so by first stating the properties of interpolation and use this to define the notion of interpolation and means, as illustrated in Fig. 2. In general, generative models often reside in a Riemannian manifold (Bortoli et al., 2022; Davidson et al., 2018; Bjerregaard et al., 2025; Rozo et al., 2025). For example, earthquake data used in Riemannian diffusion models (Bortoli et al., 2022) are restricted to the sphere, while image data used to train diffusion models reside in Euclidean space (Song et al., 2021b; Ho et al., 2020). We aim to construct a generic geometric framework that favors high-density regions learned by any generative model, given the metric structure of the data. More formally, we will consider a Riemannian manifold $\mathcal{M}$, in which the generative process resides, and consider any function $S : \mathcal{M} \to \mathbb{R}$ that is bounded from below.

In this paper, we set the metric for all experiments set to the metric of the ambient space on which the generative model relies and $S$ as the negative log-likelihood learned by the generative model, but this is not restrictive. For a Riemannian manifold, geodesics can be found by minimizing the energy functional in Eq. 3,

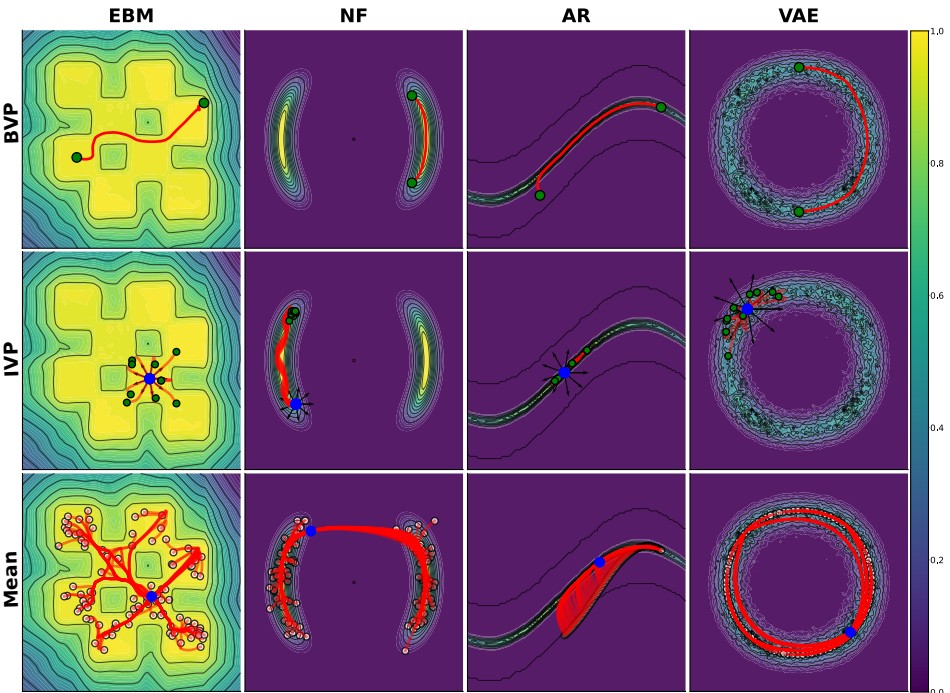

Figure 2: Examples of the three fundamental concepts derived from the (pseudo-)metric in Eq. 7 with $\lambda = 20.0$: boundary-value curves (row 1), initial value curves (row 2), and a generalized version of the Fréchet mean (row 3). The concepts are applied to an energy-based model (EBM) (LeCun et al., 2006) for a checkerboard, normalizing flow (NF) (Papamakarios et al., 2021) for the two-moon distributions, autoregressive model (AR) (Uria et al., 2016) for a sinus curve, and a VAE (Kingma & Welling, 2014) for circle data. Red denotes the connecting curves between the green boundary points. For the generalized Fréchet mean, blue denotes the mean, while the white points are 100 sampled data points. The black arrows denote the initial directions of the curves. The regularization function for each model is described in Section 4.

but this disregards the learned data distribution. To ensure that the interpolating curves are attracted to the data distribution, we propose the following constrained minimization problem

$$
\begin{aligned}
\min_{\gamma} \quad & \frac{1}{2} \int_0^1 \dot{\gamma}(t)^\top \boldsymbol{G}\left(\gamma(t)\right) \dot{\gamma}(t)\, \mathrm{d}t, \\
\text{s.t.} \quad & \int_0^1 S\left(\gamma(t)\right)\, \mathrm{d}t \leq \bar{S},
\end{aligned}
\tag{4}
$$

where $\bar{S} \in \mathbb{R}$ is a suitable bound and $\gamma(0) = \boldsymbol{a} \in \mathcal{M}$ and $\gamma(1) = \boldsymbol{b} \in \mathcal{M}$. Minimizing this constrained optimization problem can be computationally expensive depending on $S$. We therefore consider the following minimization problem of the regularized energy functional, which corresponds to a Lagrange relaxation of the constraint in Eq. 4.

$$
\gamma^* = \min_{\gamma} \mathcal{E}_\lambda(\boldsymbol{a}, \boldsymbol{b}) \quad \text{where} \quad \mathcal{E}_\lambda = \frac{1}{2} \int_0^1 \dot{\gamma}(t)^\top \boldsymbol{G}\left(\gamma(t)\right) \dot{\gamma}(t)\, \mathrm{d}t + \lambda \int_0^1 S\left(\gamma(t)\right) \mathrm{d}t,
\tag{5}
$$

where $\lambda > 0$ is a dual variable. If $\boldsymbol{u}^\top \boldsymbol{G}(\boldsymbol{z})\boldsymbol{u}$ and $S(\boldsymbol{z})$ are convex in $(\boldsymbol{z}, \boldsymbol{u})$, then there exists a $\lambda$ such that the solution to this minimization problem Eq. 5 is a solution to the minimization problem in Eq. 4. If this is not the case, there could be duality gaps between Eq. 5 and 4, i.e., the solution to Eq. 5 is an optimal solution, but there is not necessarily a $\lambda$ for all values of $\bar{S}$ in Eq. 4. For the rest of the paper, we will consider the minimization problem in Eq. 5 to reduce computational complexity and implicitly assume that a critical point is a minimum point.

**Metric.** The energy minimization in Eq. 5 provides a direct trade-off between smooth curves with respect to the geometry and targeting high-likelihood areas defined by $S$, which is directly controlled by $\lambda$. Inspired

by Eq. 5, we define the following (pseudo-)metric structure of the Riemannian manifold $\mathcal{M}$ and the tangent space $T_{\boldsymbol{z}}\mathcal{M}$ for $\boldsymbol{z} \in \mathcal{M}$.

**Definition 3.1** (Metric structure). *Let $(\mathcal{M}, g)$ be a Riemannian manifold and let $S : \mathcal{M} \to \mathbb{R}$ be bounded from below. We define the following (pseudo-)inner product in the tangent space $T_{\boldsymbol{z}}\mathcal{M}$ for $\boldsymbol{z} \in \mathcal{M}$*

$$F_{\boldsymbol{z},\lambda}(\boldsymbol{v}, \boldsymbol{w}) = \boldsymbol{v}^{\top} \boldsymbol{G}(\boldsymbol{z}) \boldsymbol{w} + \lambda S(\boldsymbol{z}), \tag{6}$$

*Using Eq. 6, we define the following (pseudo-)metric on $\mathcal{M}$*

$$\text{dist}_{\lambda}^{2}(\boldsymbol{a}, \boldsymbol{b}) = \min_{\boldsymbol{\gamma}} \mathcal{E}_{\lambda}(\boldsymbol{a}, \boldsymbol{b}) \tag{7}$$

*where $\boldsymbol{a}, \boldsymbol{b} \in \mathcal{M}$ with $\boldsymbol{\gamma}(0) = \boldsymbol{a}$ and $\boldsymbol{\gamma}(1) = \boldsymbol{b}$ and $\mathcal{E}_{\lambda}(\boldsymbol{a}, \boldsymbol{b})$ is the regularized energy Eq. 5.*

Since we have assumed that $\mathcal{M}$ is geodesically complete for the Riemannian background metric in which the data reside and that $S$ is bounded from below, there exists a solution to Eq. 7. Note that in Eq. 7 we have removed $1/2$ to simplify the computations. Since the scale between $S$ and $G$ might be very different, we show in Appendix E how to heuristically normalize $\lambda$, which we apply for all computations. Note also that the phrasing, *inner product*, in Definition 3.1 is not necessarily correct. It is easily seen that $F$ generally does not satisfy positive-definiteness, i.e., $F_{\boldsymbol{z},\lambda}(v, v) > 0$ for $v \neq 0$. In this way, we can consider $S$ as a shift of the Riemannian inner product. Similarly, the (pseudo-)metric in Eq. 7 is not a metric, as it can be negative depending on $S$. To simplify the wording, we will denote Eq. 6 and Eq. 7 as an inner product and metric, respectively. However, we show that we have a local representation in terms of a Riemannian metric along the optimal curve.

**Proposition 3.2** (Local Metric). *Let $\tilde{S}$ denote the lower bound of $S$. Assume that $\lambda$ is sufficiently large so that $S(\cdot)$ is close to $\tilde{S}$ in Eq. 4. Let $\boldsymbol{\gamma}^{*}$ denote the optimal solution to Eq. 7. Then the regularized energy can locally along the optimal curve $\boldsymbol{\gamma}^{*}$ be estimated as*

$$\mathcal{E}(\boldsymbol{\gamma}) \approx \tilde{S} + \int_{0}^{1} (\dot{\boldsymbol{\gamma}}^{*})^{\top}(t) \boldsymbol{U}(\boldsymbol{\gamma}^{*}(t)) \dot{\boldsymbol{\gamma}}^{*}(t) \, \mathrm{d}t,$$

*where $\boldsymbol{U}(\boldsymbol{\gamma}^{*}(t)) = \boldsymbol{G}(\boldsymbol{\gamma}^{*}(t)) + \frac{\lambda}{2}\boldsymbol{H}(S)(\boldsymbol{\gamma}^{*}(t))$ and $\boldsymbol{H}(S)$ denotes the Euclidean Hessian of $S$. Thus, if $\boldsymbol{U}(\boldsymbol{\gamma}^{*}(t))$ is positive definite, we can interpret it as a local representation of a Riemannian metric along the optimal curve.*

*Proof.* See Appendix A.1. $\qquad\square$

In case $S = -\log p$, we see that in Proposition 3.2 we get the Hessian of the log-likelihood similar to the Fisher-Rao metric (Nielsen, 2020). Thus, we can interpret our method as a trade-off between the naturally length-minimizing curves for the given metric structure of the space for $\lambda = 0$ and the curves strongly restricted to the data distribution as $\lambda$ increases. The exact choice of $\lambda$ is therefore a preference for the user that balances the smoothness and likelihood of the curves depending on the use-case, as illustrated in Fig. 3 for simple synthetic data with a Gaussian mixture model (GMM) and a kernel density estimator (KDE) with $G = I$ and $S = -\log p$. Throughout the paper, we will consider $\lambda = 20.0$ and use the normalizing described in Appendix F, which we find to nicely balance high-likelihood and smoothness.

In the following, we will derive and define the properties of the metric in Eq. 7, as well as computational methods for fast and accurate computations for statistical analysis.

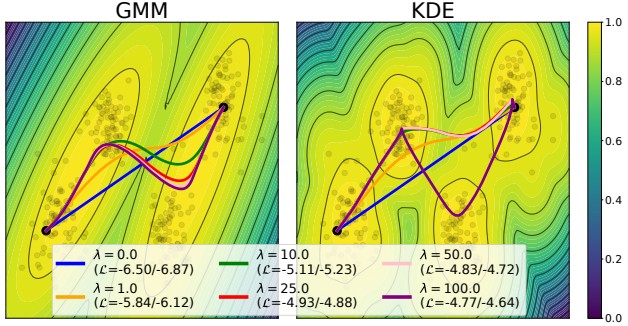

Figure 3: Interpolation curves computed for different values of $\lambda$ in Eq. 5 with an Euclidean background metric and $S(\boldsymbol{z}) = -\log p(\boldsymbol{z})$. The data density, $p$, is approximated by a GMM and KDE shown in the background color for synthetic data (black). The mean log-likelihoods of the estimated curves are denoted $\mathcal{L}$, where the first number is for GMM and the second for KDE.

**Minimizing curves.** In the Riemannian case, geodesics serve as the foundation for elaborate statistical models and are deeply rooted in the exponential and logarithmic map that generalizes the notion of vector addition and subtraction to Riemannian manifolds (Hauberg, 2025, Chapter 9). The exponential and logarithmic map, when it exists, can be found by solving the ODE in Eq. 1 as an initial-value problem (IVP) or boundary-value problem (BVP), respectively. Given the metric in Eq. 7, we derive an ODE that defines a generalized geodesic.

**Proposition 3.3** (First variation). *Consider the regularized energy in Eq. 7 for a Riemannian manifold $(\mathcal{M}, g)$. Let $g_{ij}$ and $g^{ij}$ denote the local coordinates of the metric matrix function and its inverse, respectively. The first variation gives the following* ODE

$$\ddot{\boldsymbol{\gamma}}^k(t) + \Gamma_{ij}^k \dot{\boldsymbol{\gamma}}^s(t)\dot{\boldsymbol{\gamma}}^j(t) = \frac{\lambda}{2} g^{kp} \partial_p S\left(\boldsymbol{\gamma}(t)\right), \tag{8}$$

*where $\Gamma_{ij}^k = \frac{1}{2} g^{kj}\left(\partial_l g_{pj} + \partial_j g_{pl} - \partial_p g_{ij}\right)$ denotes the Christoffel symbols of the Levi-Civita connection. Note that Eq. 8 can equivalently be written as*

$$\dot{\boldsymbol{\gamma}}(t) = -\frac{1}{2}\boldsymbol{G}(\boldsymbol{\gamma}(t))\mu(t), \quad \dot{\boldsymbol{\mu}}(t) = \frac{1}{4}\nabla_x\left(\boldsymbol{\mu}^\top(t)\boldsymbol{G}(x)\boldsymbol{\mu}(t)\right)\big|_{x=\boldsymbol{\gamma}(t)} - \lambda\nabla_x S(x)\big|_{x=\boldsymbol{\gamma}(t)}, \tag{9}$$

*where $\boldsymbol{\mu}(0) = -2\boldsymbol{G}(\boldsymbol{a})\dot{\boldsymbol{\gamma}}(0)$.*

*Proof.* See Appendix A.2 for the proof of Eq. 8. For the proof of Eq. 9, we refer to Rygaard (2026). □

If $\lambda = 0$, we see from Eq. 8 that we immediately get the corresponding ODE for Riemannian geodesics Eq. 1. Note that Eq. 8 is similar to the ODE governing motion on a Riemannian manifold under external forces, where $\frac{\lambda}{2} g^{kp} \partial_p S\left(\boldsymbol{\gamma}(t)\right)$ can be interpreted as the external force (Cariñena & Muñoz-Lecanda, 2023). In this way, we can interpret $S$ as a force that pushes the curve into the data distribution, where the "push" is controlled by $\lambda$ corresponding to a Newtonian system on a Riemannian manifold. Note that the ODE in Eq. 9 can be suitable in high-dimensions as it avoids the cumbersome computation of the Christoffel symbols and only requires the gradient of an inner product. However, for all computations we will use the ODE in Eq. 8. Note also that if Eq. 7 is locally convex for the solution to Eq. 8, the solution to Eq. 8 is a local minimum to Eq. 7.

Using Eq. 8, we can easily define the Exponential map corresponding to the metric in Eq. 7 as $\mathrm{Exp}_{x,\lambda}(v) = \boldsymbol{\gamma}(1)$, where $\boldsymbol{\gamma}(1)$ is the solution to Eq. 8 with $\boldsymbol{\gamma}(0) = \boldsymbol{x}$ and $\dot{\boldsymbol{\gamma}}(0) = v$. Similarly, we can define, when it exists, the logarithmic map as $\mathrm{Log}_{x,\lambda}(\boldsymbol{y}) = \dot{\boldsymbol{\gamma}}(0)$ as the solution to Eq. 8 with $\boldsymbol{\gamma}(0) = x$ and $\boldsymbol{\gamma}(1) = \boldsymbol{y}$. The exponential map corresponds to an IVP that can be estimated numerically using standard ODE solvers (Dormand & Prince, 1980; Bogacki & Shampine, 1989; Hairer et al., 1993; Hairer & Wanner, 1996; Shampine & Reichelt, 1997; Hindmarsh, 1983). The logarithmic map corresponds to a BVP problem which, in principle, can be solved similarly using BVP solvers (Noakes, 1998; Kaya & Noakes, 2008; Hennig & Hauberg, 2014; Arvanitidis et al., 2019), but below we will derive a fast and efficient algorithm to solve the BVP rather than using ODE solvers.

**Geometric statistics.** Using the modified version of the exponential and logarithmic map, we can compute geodesics targeting high-density regions. The metric structure in Eq. 7 lets us go further and compute geometric statistics, such as means and principal components, on the manifold (Pennec, 2006; Fletcher et al., 2004). This allows generative models to be incorporated into data analysis pipelines. Using Eq. 7, we define the discrete mean-value as

$$\mu, \{\gamma_s\}_{i=1}^{N_{\mathrm{data}}} = \underset{\substack{\boldsymbol{y}\in\mathcal{M} \\ \{\gamma_s\}_{i=1}^{N_{\mathrm{data}}}}}{\arg\min} \sum_{i=1}^{N_{\mathrm{data}}} w_i \mathcal{E}_\lambda\left(\boldsymbol{a}_i, \boldsymbol{y}\right), \tag{10}$$

where $\{\boldsymbol{a}_i\}_{i=1}^{N_{\mathrm{data}}} \subset \mathcal{M}$ are data points with weights $\{w_i\}_{i=1}^{N_{\mathrm{data}}} \subset \mathbb{R}_+$ and shortest curves $\{\gamma_i\}_{i=1}^{N_{\mathrm{data}}}$ connecting the data points and mean $\mu$. If $\lambda = 0$, then this is equivalent to the Fréchet mean (Fréchet, 1948). Generally, uniqueness and existence of the Fréchet mean is not guaranteed even on Riemannian manifolds. In

Appendix C, we summarize the conditions for existence and uniqueness on Riemannian manifolds and state results for when this holds for Eq. 10. From the definition of the mean in Eq. 10, we can easily generalize well-known geometric statistics such as *geodesic regression* (Fletcher, 2011) and *principal geodesic analysis* (PGA) (Fletcher et al., 2004).

**Algorithm.** Solving the IVP using the ODE in Eq. 8 is computationally cheap. However, computing geodesics as a BVP can even in the Riemannian case be computationally difficult and often scales poorly in high dimensions or exhibits slow convergence. Rygaard & Hauberg (2025) circumvent this through an iterative scheme to estimate geodesics with fast convergence using optimal control known as the *GEORCE-* algorithm. However, their method does not support regularization of the geodesic. We generalize *GEORCE* to compute boundary-value geodesics under regularization and prove that our extended method converges to a local minimum (global convergence) and exhibits local asymptotic quadratic convergence under certain additional conditions similar to Rygaard & Hauberg (2025). We formulate the regularized geodesic problem in Eq. 5 as a discrete control problem using a first order approximation of the velocity along the curve in Eq. 5.

$$\min_{(\boldsymbol{z}_s, \boldsymbol{u}_s)} E(\boldsymbol{z}) := \min_{(\boldsymbol{z}_s, \boldsymbol{u}_s)} \sum_{s=0}^{N_{\text{grid}}-1} \left( \boldsymbol{u}_s^\top \boldsymbol{G}(\boldsymbol{z}_s) \boldsymbol{u}_s + \lambda S(\boldsymbol{z}_s) \right)$$
$$\text{s.t.} \quad \boldsymbol{z}_{s+1} = \boldsymbol{z}_s + \boldsymbol{u}_s, \quad s = 0, \ldots, N_{\text{grid}} - 1,$$
$$\boldsymbol{z}_0 = \boldsymbol{a}, \, \boldsymbol{z}_{N_{\text{grid}}} = \boldsymbol{b}. \tag{11}$$

With this formulation, $\boldsymbol{z}_{0:N_{\text{grid}}}$ denotes the state variables, while $\boldsymbol{u}_{0:(N_{\text{grid}}-1)}$ denotes the control variable such that the control variables correspond to a discretization of the tangent vectors at the grid points of the curve. Since the regularizing function depends only on the state $\boldsymbol{z}_{0:N_{\text{grid}}}$, we can apply a similar approach as Rygaard & Hauberg (2025) to decompose the optimal control problem in Eq. 11 into convex subproblems in the control variables. Using this, we show that

**Proposition 3.4.** *The necessary conditions for a minimum in Eq. 11 are*

$$2\boldsymbol{G}(\boldsymbol{z}_s)\boldsymbol{u}_s + \boldsymbol{\mu}_s = 0, \quad s = 0, \ldots, N_{\text{grid}} - 1,$$
$$\boldsymbol{z}_{s+1} = \boldsymbol{z}_s + \boldsymbol{u}_s, \quad s = 0, \ldots, N_{\text{grid}} - 1,$$
$$\nabla_{\boldsymbol{y}} \left[ \boldsymbol{u}_s^\top \boldsymbol{G}(\boldsymbol{y})\boldsymbol{u}_s + \lambda S(\boldsymbol{y}) \right]\big|_{\boldsymbol{y}=\boldsymbol{z}_s} + \boldsymbol{\mu}_s = \boldsymbol{\mu}_{s-1}, \quad s = 1, \ldots, N_{\text{grid}} - 1. \tag{12}$$
$$\boldsymbol{z}_0 = \boldsymbol{a}, \boldsymbol{z}_{N_{\text{grid}}} = \boldsymbol{b},$$

*where $\boldsymbol{\mu}_s \in \mathbb{R}^d$ for $s = 0, \ldots, N_{\text{grid}} - 1$.*

*Proof.* See Appendix A.3. □

The necessary conditions in Eq. 12 can generally not be solved with respect to $\boldsymbol{u}_{0:(N_{\text{grid}}-1)}$ and $\boldsymbol{\mu}_{0:(N_{\text{grid}}-1)}$. However, we can circumvent this iteratively. In iteration $k$ consider the state and control variables $\boldsymbol{z}_{0:N_{\text{grid}}}^{(k)}$ and $\boldsymbol{u}_{0:(N_{\text{grid}}-1)}^{(k)}$. We fix $\boldsymbol{G}(\cdot)$ and the gradient term in iteration $k$ to define the following variables.

$$\boldsymbol{\nu}_s := \nabla_{\boldsymbol{y}} \left( \boldsymbol{u}_s^\top \boldsymbol{G}(\boldsymbol{y})\boldsymbol{u}_s + \lambda S(\boldsymbol{y}) \right)\big|_{\boldsymbol{y}=\boldsymbol{z}_s^{(k)}, \boldsymbol{u}_s=\boldsymbol{u}_s^{(k)}}, \quad s = 1, \ldots, N_{\text{grid}} - 1,$$
$$\boldsymbol{G}_s := \boldsymbol{G}\left(\boldsymbol{z}_s^{(k)}\right), \quad s = 0, \ldots, N_{\text{grid}} - 1. \tag{13}$$

Inserting these variables into Eq. 12, we can solve the equations in closed-form, which naturally leads to the following update scheme.

**Proposition 3.5** (Update Scheme). *The update scheme for $\boldsymbol{u}_s, \boldsymbol{\mu}_s$ and $\boldsymbol{z}_s$ is*

$$
\begin{aligned}
\boldsymbol{\mu}_{N_{\text{grid}}-1} &= \left( \sum_{s=0}^{N_{\text{grid}}-1} \boldsymbol{G}_s^{-1} \right)^{-1} \left( 2(\boldsymbol{a} - \boldsymbol{b}) - \sum_{s=0}^{N_{\text{grid}}-1} \boldsymbol{G}_s^{-1} \sum_{j>s} \boldsymbol{\nu}_j \right), \\
\boldsymbol{u}_s &= -\frac{1}{2} \boldsymbol{G}_s^{-1} \left( \boldsymbol{\mu}_{N_{\text{grid}}-1} + \sum_{j>s} \boldsymbol{\nu}_j \right), \quad s = 0, \dots, N_{\text{grid}} - 1, \\
\boldsymbol{z}_{s+1} &= \boldsymbol{z}_s + \boldsymbol{u}_s, \quad s = 1, \dots, N_{\text{grid}} - 1, \\
\boldsymbol{z}_0 &= \boldsymbol{a}.
\end{aligned}
\tag{14}
$$

Some generative models, such as diffusion models, operate directly in the high-dimensional ambient Euclidean space, using the Euclidean metric throughout training and generation. In this case $\boldsymbol{G} = \boldsymbol{I}$ and storing any matrix would be intractable. However, for a Euclidean metric, the update formulas reduce to

**Corollary 3.6** (Euclidean Update Scheme). *Assume $\boldsymbol{G} = \boldsymbol{I}$. The update scheme for $\boldsymbol{u}_s$ and $\boldsymbol{z}_s$ is*

$$
\begin{aligned}
\boldsymbol{u}_s &= \frac{\boldsymbol{b} - \boldsymbol{a}}{N_{\text{grid}}} + \frac{1}{2} \left( \frac{1}{N_{\text{grid}}} \sum_{k=0}^{N_{\text{grid}}-1} \sum_{j>k} \boldsymbol{\nu}_j - \sum_{j>s} \boldsymbol{\nu}_j \right) \quad s = 0, \dots, N_{\text{grid}} - 1, \\
\boldsymbol{z}_{s+1} &= \boldsymbol{z}_s + \boldsymbol{u}_s, \quad s = 0, \dots, N_{\text{grid}} - 1, \\
\boldsymbol{z}_0 &= \boldsymbol{a},
\end{aligned}
\tag{15}
$$

*where $\boldsymbol{\nu}_s := \lambda \nabla_{\boldsymbol{y}} S(\boldsymbol{y})\big|_{\boldsymbol{y}=\boldsymbol{z}_s}$*

Thus, for a Euclidean metric, the algorithm has the same complexity as a gradient descent method. Given the update scheme in Proposition 3.5, the new iteration is found by applying line-search with backtracking similar to (Rygaard & Hauberg, 2025). We denote the algorithm *ProbGEORCE* (Probabilistic GEORCE) and display it in pseudo-code in Appendix B, where we apply line-search in the update scheme using backtracking with parameter $\rho = 0.5$ similar to (Rygaard & Hauberg, 2025). We note that compared to the original *GEORCE*-algorithm for unregularized geodesic construction, the only difference between the algorithms is that $\boldsymbol{\nu}_s$ depends on $\lambda S(\boldsymbol{z}_s)$. Note that $\boldsymbol{u}_0$ can be directly interpreted as the logarithmic map modulo scaling. If $S$ or $\boldsymbol{G}$ are expensive to evaluate, then repeatedly evaluating the energy functional for line-search can be time-consuming and possibly intractable. In Appendix A.5, we show how to adaptively estimate $\alpha$ using an adaptive update scheme similar to *ADAM* to avoid repeated evaluation of $\boldsymbol{G}$ and $S$. If $S$ stems from a neural network, we will use the adaptive update scheme derived in Appendix A.5 and otherwise we will use line-search to estimate the steps-size $\alpha$. In general, we set $N_{\text{grid}} = 100$, while for image interpolation we set $N_{\text{grid}} = 10$.

In Table 1, we show the complexity of our method compared to applying other methods for minimizing Eq. 11. Note that since the Hessian of the first order approximation of the energy with respect to all grid points is a diagonal is sparse, and that the Hessian of $S$ with respect to all grid points is a diagonal of block matrices, then the complexity of the Newton method is not $\mathcal{O}\left(N_{\text{grid}}^3 d^3\right)$. Also note that if $G = I$, then *ProbGEORCE* has complexity $\mathcal{O}\left(N_{\text{grid}} d\right)$, since the metric matrix inversion is avoided.

Table 1: Complexity of different optimization algorithms.

| Method | Complexity |
|---|---|
| Gradient descent | $\mathcal{O}\left(N_{\text{grid}} d\right)$ |
| Quasi-Newton method | $\mathcal{O}\left(N_{\text{grid}}^2 d^2\right)$ |
| (Sparse) Newton method | $\mathcal{O}\left(N_{\text{grid}} d^3\right)$ |
| *ProbGEORCE* | $\mathcal{O}\left(N_{\text{grid}} d^3\right)$ |

In Appendix A.4 we show that *ProbGEORCE* exhibits converge to a local minimum (global convergence) similar to the original *GEORCE*-algorithm (Rygaard & Hauberg, 2025). Similar to (Rygaard & Hauberg, 2025), we also show that *ProbGEORCE* has locally asymptotic quadratic convergence if $S$ is an affine function. In practice, $S$ will not be an affine function, and therefore our method will be fast for sufficiently small values of $\lambda$, but as $\lambda$ increases our method will not necessarily be faster than optimizers. This results seems intuitive as when $\lambda$ increases the Riemannian energy is negligible, and therefore the problem is to

minimize a general function $S$. For the generalized Fréchet mean in Eq. 10, we can similarly consider the discretized version

$$\min_{(\boldsymbol{z}_{s,i}, \boldsymbol{u}_{s,i})} \sum_{i=1}^{N_{\text{data}}} \sum_{s=0}^{N_{\text{grid}}} w_i \left( \boldsymbol{u}_{s,i}^\top \boldsymbol{G}(\boldsymbol{z}_{s,i}) \boldsymbol{u}_{s,i} + \lambda S(\boldsymbol{z}_{s,i}) \right), \tag{16}$$

where $\boldsymbol{z}_{s+1,i} = \boldsymbol{u}_{s,i} + \boldsymbol{z}_{s,i}$ and $\boldsymbol{z}_{0,i} = \boldsymbol{a}_i \in \mathcal{M}$ denote the data points for $i = 1, \dots, N_{\text{data}}$ and $\boldsymbol{z}_{N_{\text{grid}},i} = \boldsymbol{y} \in \mathcal{M}$ denote the candidate mean. Following a similar approach, we show in Appendix A.6 how to estimate the mean in Eq. 10 by generalizing the approach from Rygaard et al. (2025).

**Application to diffusion models.** Using our framework, we have defined a (pseudo-)metric that deviates from the background Riemannian metric with an external force field. A natural choice for this force is to set $S = -\log p$, but this quantity is not available for diffusion models without having to solve the probability flow ODE, which is expensive. However, $\nabla_{\boldsymbol{z}} S = -\nabla_{\boldsymbol{z}} \log p$, is the only one needed for the ODE in Eq. 8, solving the BVP in Eq. 14 and computing the mean in Appendix C. For diffusion models $\nabla_{\boldsymbol{z}} S = -\nabla_{\boldsymbol{z}} \log p$ is exactly the *score* function and, therefore, our method is directly applicable to this case. However, for diffusion models, the score depends also on time. Given our framework, we can compute the regularized geodesics in the following two ways:

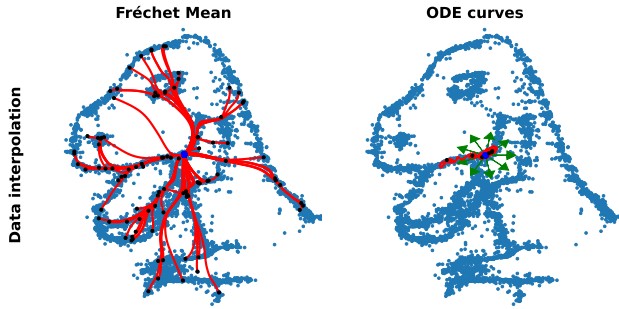

Figure 4: The Fréchet mean (blue) for the dino-dataset using a DDPM computed in data space as well as 10 curves (red) in ten different directions (green arrows) for the corresponding IVP (8).

- Curves in noise space: Given the limiting distribution of the diffusion model, we compute the interpolation in noise space and, using the time-reversal dynamics, transport the geodesic to data space. The regularization function can be set as the log-likelihood of the limiting distribution.

- Curves in data space: With our framework, we can also directly compute the geodesics in data space by applying the score function for time sufficiently close to 0 using the adaptive update scheme for *ProbGEORCE* in Appendix A.5.

Both approaches provide different pros and cons. Computing curves in noise space can be computationally cheaper, since the limiting distributions, and hence its density, are often known in closed form. However, these curves will only be smooth if the sampling scheme of the diffusion model is deterministic as for the probability flow ODE (Chen et al., 2018) or DDIM sampling (Song et al., 2021a). Note also that high density in noise space does not necessarily imply high density in the data space by the change of variable formula. In contrast, computing the curves in the data space will result in smooth curves and uses the learned score function directly. However, this is computationally more expensive, as it requires evaluating the diffusion model repeatedly in the optimization, and be numerically unstable if the initial curve is not sufficiently close to data to have a proper estimate of the score. In Fig. 4, we show the corresponding Fréchet mean and IVP curves for the dino-DDPM computed in the data space. In Appendix D, we provide more details on diffusion models, the difference between interpolation in noise and data space, and also how our method can be incorporated into image editing methods such as NoiseDiffusion (Zheng et al., 2024).

# 4 Experiments

**Runtime.** Initially, to illustrate the efficiency of *ProbGEORCE*, Fig. 5 considers the BVP between two points in a local chart of the $n$-sphere. We set $S = -\log p$, where $p$ denotes the data density of three randomly weighted isotropic Gaussian distributions with random means, and $\lambda = 1.0$. We terminate the algorithms if the norm of the gradient of Eq. 11 is less than $10^{-4}$. We see that our algorithm obtains the lowest regularized energy across the different dimensions. We also see that *ProbGEORCE* with line-search (LS) is significantly faster compared to using the adaptive scheme. This is expected as the evaluation of $S$ is computationally cheap.

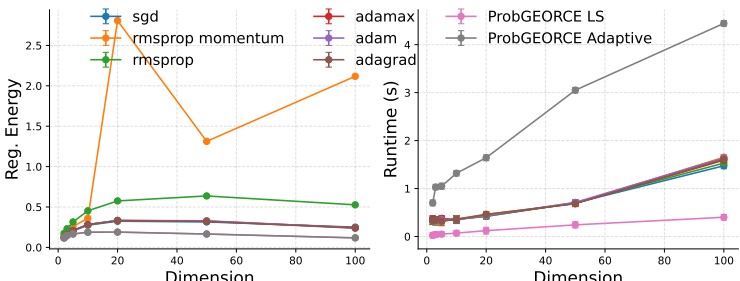

Figure 5: The regularized energy and runtime for different methods minimizing Eq. 7 for $\lambda = 1.0$ for the $n$-sphere and $S = -\log p$, where $p$ is the density of three Gaussian randomly weighted with random mean values and identity as the covariance matrix. Note that the regularized energy for *ProbGEORCE* with line-search (LS) and adaptive update scheme (Adaptive) are very similar.

In Appendix G, we show the runtime and estimates for other manifolds, and also for the models in Fig. 2 for different optimizers and values of $\lambda$, where $S$ is expensive to evaluate. In this case, the adaptive update scheme performs significantly better than line-search as expected. In Appendix G, we also show the runtime and estimates for varying $\lambda$, where we see that if $\lambda$ is sufficiently high, then our method is similar to other optimizers. This is expected as when $\lambda$ is sufficiently high, then the energy term in Eq. 5 is negligible, and the problem is to minimize a general function $S$.

Table 2: Negative log-likelihood (NLL) and runtime results for the energy-based model, normalizing flow, autoregressive model and variational autoencoder used in Fig. 2. For the VAE, we report the ELBO. We write $-$ if the Riemannian metric is not invertible or if the generative model returns nans for the likelihood. In Appendix E, we provide details on the benchmarks and any hyper-parameters.

| Method | EBM | | NF | | AR | | VAE | |
|---|---|---|---|---|---|---|---|---|
| | NLL ↓ | Runtime ↓ | NLL ↓ | Runtime ↓ | NLL ↓ | Runtime ↓ | NLL ↓ | Runtime ↓ |
| **IVP** | | | | | | | | |
| Ours ($\lambda = 20.0$) | **8.483** | $4.90 \pm 0.09$ | $\mathbf{1.20 \times 10^3}$ | $101.08 \pm 2.17$ | $\mathbf{1.37 \times 10^4}$ | $7.66 \pm 0.08$ | **-144.673** | $64.01 \pm 0.49$ |
| Euclidean | 30.010 | $\mathbf{0.00 \pm 0.00}$ | $4.01 \times 10^3$ | $\mathbf{0.00 \pm 0.00}$ | $1.06 \times 10^5$ | $\mathbf{0.00 \pm 0.00}$ | -105.053 | $\mathbf{0.00 \pm 0.00}$ |
| Spherical | 16.536 | $0.00 \pm 0.00$ | $4.10 \times 10^3$ | $0.00 \pm 0.00$ | $8.25 \times 10^4$ | $0.00 \pm 0.00$ | -85.552 | $0.00 \pm 0.00$ |
| Fisher-Rao | 29.682 | $12.61 \pm 0.09$ | - | - | - | - | - | - |
| Jacobian (Saito & Matsubara, 2025) | 29.682 | $18.50 \pm 0.03$ | - | - | $6.03 \times 10^{32}$ | $54.93 \pm 0.58$ | - | - |
| Jacobian (Reg) | 29.682 | $25.43 \pm 0.07$ | - | - | $5.30 \times 10^{29}$ | $61.82 \pm 2.22$ | - | - |
| Inverse Density (Yu et al., 2025) | 28.673 | $9.14 \pm 0.06$ | $2.78 \times 10^3$ | $255.21 \pm 1.63$ | - | - | - | - |
| Generative (Kim et al., 2024) | 29.184 | $9.70 \pm 0.08$ | $3.87 \times 10^3$ | $225.10 \pm 2.41$ | $1.08 \times 10^5$ | $17.06 \pm 0.12$ | -120.631 | $212.38 \pm 2.05$ |
| Monge (Hartmann et al., 2022) | 29.682 | $12.64 \pm 0.05$ | $2.81 \times 10^3$ | $452.26 \pm 2.78$ | $9.51 \times 10^4$ | $31.83 \pm 0.23$ | -121.875 | $399.59 \pm 3.87$ |
| **BVP** | | | | | | | | |
| Ours ($\lambda = 20.0$) | **-1.892** | $7.89 \pm 0.02$ | **94.492** | $217.49 \pm 0.37$ | **28.611** | $11.53 \pm 0.20$ | **-24.382** | $167.97 \pm 3.81$ |
| Euclidean | 2.435 | $\mathbf{0.00 \pm 0.00}$ | 273.797 | $\mathbf{0.00 \pm 0.00}$ | $6.05 \times 10^3$ | $\mathbf{0.00 \pm 0.00}$ | 18.142 | $\mathbf{0.00 \pm 0.00}$ |
| Spherical | 141.120 | $0.00 \pm 0.00$ | 103.706 | $0.00 \pm 0.00$ | $1.43 \times 10^4$ | $0.00 \pm 0.00$ | 948.467 | $0.00 \pm 0.00$ |
| Fisher-Rao | 0.863 | $21.99 \pm 0.04$ | - | $1.49 \pm 0.00$ | $1.73 \times 10^{14}$ | $37.51 \pm 0.36$ | - | - |
| Jacobian (Saito & Matsubara, 2025) | 2.435 | $2.52 \pm 0.02$ | - | - | $7.84 \times 10^5$ | $6.90 \times 10^3 \pm 23.44$ | - | - |
| Jacobian (Reg) | 2.435 | $2.41 \pm 0.01$ | - | - | $7.15 \times 10^5$ | $6.98 \times 10^3 \pm 53.38$ | - | - |
| Inverse Density (Yu et al., 2025) | -1.451 | $16.41 \pm 0.01$ | 273.797 | $0.39 \pm 0.00$ | $6.05 \times 10^3$ | $0.04 \pm 0.00$ | - | - |
| Generative (Kim et al., 2024) | -1.256 | $16.63 \pm 0.35$ | 274.235 | $158.04 \pm 0.70$ | $6.05 \times 10^3$ | $0.04 \pm 0.00$ | 21.332 | $116.09 \pm 0.41$ |
| Monge (Hartmann et al., 2022) | 2.490 | $24.69 \pm 0.23$ | 174.743 | $553.14 \pm 1.95$ | $3.50 \times 10^6$ | $38.31 \pm 0.15$ | -6.788 | $314.86 \pm 5.48$ |
| **Mean** | | | | | | | | |
| Ours ($\lambda = 20.0$) | **-79.383** | $9.78 \pm 0.04$ | $\mathbf{2.14 \times 10^4}$ | $188.16 \pm 4.97$ | $\mathbf{2.54 \times 10^4}$ | $8.80 \pm 0.38$ | $\mathbf{-2.28 \times 10^3}$ | $166.96 \pm 0.10$ |
| Euclidean | 153.896 | $\mathbf{0.00 \pm 0.00}$ | $1.18 \times 10^5$ | $\mathbf{0.00 \pm 0.00}$ | $5.62 \times 10^5$ | $\mathbf{0.00 \pm 0.00}$ | $2.07 \times 10^3$ | $\mathbf{0.00 \pm 0.00}$ |
| Spherical | $7.52 \times 10^3$ | $0.11 \pm 0.00$ | - | $0.11 \pm 0.00$ | $1.26 \times 10^7$ | $0.11 \pm 0.00$ | -558.952 | $0.14 \pm 0.00$ |
| Fisher-Rao | 28.670 | $30.32 \pm 0.12$ | - | $1.68 \pm 0.02$ | $7.20 \times 10^{18}$ | $0.12 \pm 0.00$ | - | - |
| Jacobian (Saito & Matsubara, 2025) | 153.896 | $247.24 \pm 0.76$ | - | - | - | - | - | - |
| Jacobian (Reg) | 153.896 | $283.84 \pm 9.80$ | - | - | - | - | - | - |
| Inverse Density (Yu et al., 2025) | 153.896 | $0.03 \pm 0.00$ | $1.18 \times 10^5$ | $0.35 \pm 0.00$ | $5.62 \times 10^5$ | $0.04 \pm 0.00$ | $2.10 \times 10^3$ | $0.32 \pm 0.00$ |
| Generative (Kim et al., 2024) | 153.896 | $0.03 \pm 0.00$ | $1.18 \times 10^5$ | $0.45 \pm 0.00$ | $5.62 \times 10^5$ | $0.04 \pm 0.00$ | $2.10 \times 10^3$ | $0.29 \pm 0.01$ |
| Monge (Hartmann et al., 2022) | 152.098 | $9.90 \pm 0.05$ | $4.93 \times 10^4$ | $572.29 \pm 18.44$ | $2.21 \times 10^6$ | $47.44 \pm 0.09$ | $-1.19 \times 10^3$ | $356.90 \pm 6.55$ |

**Synthetic data.** We consider the four generative models and the corresponding data distributions in Fig. 2. Energy-based models estimate the data distribution as $\exp\left(-E_\theta(\boldsymbol{x})\right)/Z_\theta$, where $E_\theta$ denotes a neural network and $Z_\theta$ is a normalization constant (LeCun et al., 2006), while normalizing flow models estimate the log-likelihood of the data distribution applying the transformation of random variables (Papamakarios et al., 2021). Neural autoregressive models learn the conditional distribution $\log p_\theta(\boldsymbol{x}) = \sum_i \log p_\theta(\boldsymbol{x}_i|\boldsymbol{x}_{<i})$ (Uria et al., 2016). For all these models, we set $S = -\log p_\theta$, except for VAE's where we use the negative negative evidence lower bound (ELBO). We show the result of applying our framework compared to other baselines in Table 2. We see that our method, in general, obtains a higher likelihood compared to different baseline methods. Since, by construction, our method (controlled by $\lambda$) targets high-density regions, we would also expect that this is the case compared to alternative methods. For a strictly Riemannian metric, the grid points along the curve should have the same distance. However, for the Newtonian system, this does not have to be the case, since the regularization term only depends on the grid points. Therefore, our method can have a longer distance between grid points in low-density areas and a denser number of grid points in high-density regions. For lower values of $\lambda$, smoothness will play a bigger role, while a higher value of $\lambda$ will have higher-likelihood but more rapidly changing curves. In Appendix E, we illustrate for the normalizing flow example in Fig. 2 how the distance between grid points increases in low-density areas as $\lambda$ increases. Note also that it is possible within our framework to combine a likelihood-based Riemannian metric with the likelihood regularization term.

**Energy-based model.** Table 2 shows that our method traverses regions of higher-likelihood. To test whether our method also gives higher quality in images, we compare with the methods proposed in (Béthune et al., 2025). Similarly to (Béthune et al., 2025), we train an energy-based model (LeCun et al., 2006) on the Animal Faces High Quality dataset (AFHQ) (Choi et al., 2020a) on the latent encodings by Stable Diffusion v1 VAE (Rombach et al., 2022). We train the energy-based model for 24 hours corresponding to 3,000 epochs. We learn a neural network, $\phi_\theta$, to compute geodesics by minimizing the energy of the respective methods, which

Table 3: Comparison of methods using FID and KID metrics (lower is better).

| Method | FID $\downarrow$ | KID $\downarrow$ |
|---|---|---|
| $\boldsymbol{G}_{1/p_\theta}$ (conformal) | 187.25 | $0.0800 \pm 0.0028$ |
| $\boldsymbol{G}_{\log p_\theta}$ (conformal) | 189.47 | $0.0819 \pm 0.0033$ |
| $\boldsymbol{G}_{1/p_\theta}$ (diagonal) | 198.06 | $0.0828 \pm 0.0032$ |
| $\boldsymbol{G}_{\log p_\theta}$ (diagonal) | 189.32 | $0.0745 \pm 0.0031$ |
| $\boldsymbol{G}_{\text{LAND}}$ | 186.87 | $0.0726 \pm 0.0028$ |
| $\boldsymbol{G}_{1/p_\theta}$ (full) | 179.62 | $0.0676 \pm 0.0024$ |
| $\boldsymbol{G}_{\log p_\theta}$ (full) | 197.63 | $0.0802 \pm 0.0033$ |
| Ours $(-1/p_\theta)$ | **149.52** | $0.0375 \pm 0.0018$ |
| Ours $(-\log p_\theta)$ | 150.50 | $\mathbf{0.0359 \pm 0.0020}$ |

is trained for up to 24 hours for each method similar to (Béthune et al., 2025). Béthune et al. (2025) considers metrics on the following form

$$\boldsymbol{G}(\boldsymbol{z}) = (\alpha h(\boldsymbol{z}) + \beta)\,\boldsymbol{I} \quad \text{or} \quad \boldsymbol{G}(\boldsymbol{z}) = (\alpha h(\boldsymbol{z}) + \beta)^{-1}\,\boldsymbol{I},$$

where $\alpha, \beta \in \mathbb{R}$ are constants and $h$ is either based on the log-likelihood $\log p_\theta$ or the inverse density $1/p_\theta$ learned by the energy based model. In Appendix E, we describe the metrics used in (Béthune et al., 2025).

We show the result in Table 3, where we consider our method with both $S(x) = -\log p_\theta$ and $S(x) = -1/p_\theta$, since Béthune et al. (2025) finds that using the inverse density gives better results. We set $\lambda = 20.0$ for our method and normalize $\lambda$ as described Appendix. F. We see that our method gives better Fréchet inception distance (FID) (Heusel et al., 2017) and kernel inception distance (KID) (Bińkowski et al., 2018) on the AFHQ dataset (Choi et al., 2020a) compared to alternatives. Note that using the inverse density provides a marginally bet-

Table 4: Comparison of methods using FID and KID metrics (lower is better) for the AFHQ dataset (Choi et al., 2020b) for the BVP connecting two images.

| Method | FID $\downarrow$ | KID $\downarrow$ |
|---|---|---|
| Ours (Noise) | 182.73 | $0.12 \pm 0.04$ |
| Ours (Data) | **172.70** | $0.11 \pm 0.04$ |
| Linear | 200.62 | $\mathbf{0.08 \pm 0.04}$ |
| Spherical | 184.21 | $0.12 \pm 0.04$ |
| NoiseDiffusion (Zheng et al., 2024) | 174.46 | $0.11 \pm 0.04$ |

ter FID, which is most likely due to the inverse density penalizing low likelihood areas more. In Appendix F, we show the image transitions for our method and the benchmarks.

**Diffusion models.** To apply our framework to high-resolution diffusion models, we consider ControlNet (Zhang et al., 2023). We apply our method both directly in data space using the score proposed by Katzir

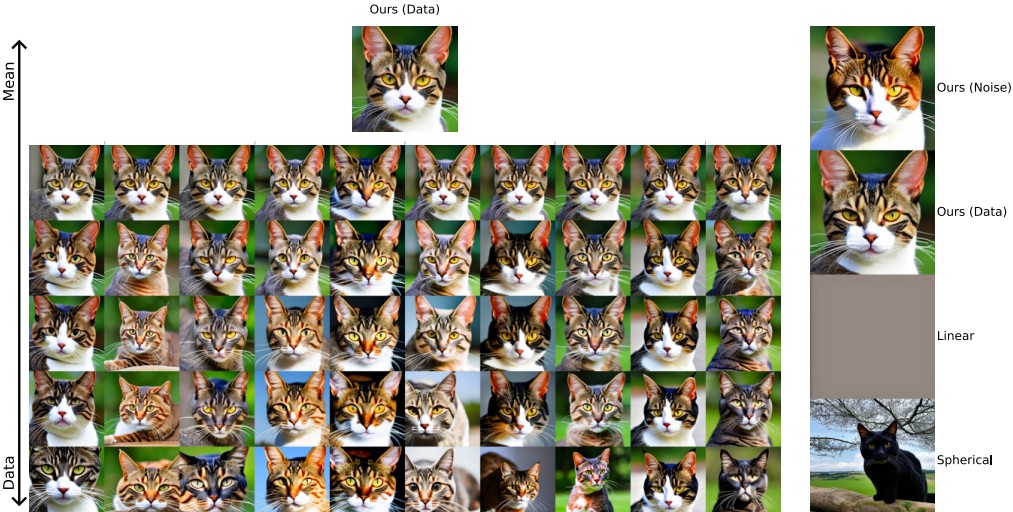

Figure 6: Left: Computed mean value using our method in data space, where the image shows the transition from the mean to the data points in the last row. Right: the estimated mean using our method in noise and data space, respectively, compared to the Fréchet mean using a linear and spherical geometry in noise space.

et al. (2024) for time 0, and in noise space using the density of a $\chi^2$-distribution on the squared norm. The latter is due to the fact that the size of the latent space is $4 \times 96 \times 96$, where the norm of an isotropic Gaussian prior is $\chi^2$-distributed and converges to a uniform distribution on a sphere. When computing curves in data space, we find that the images can get blurry. Therefore, when computing curves directly in the data space, we encode the computed curves in noise space and decode them back to data space, as we find that this gives smooth and realistic images.

In Table 4, we compute FID (Heusel et al., 2017) and KID (Bińkowski et al., 2018) on the AFHQ dataset (Choi et al., 2020b) for our method and compare it with alternative methods. In general, for interpolation, we do not find a major difference between the different methods in terms of realism, as also seen in Fig. 7 for interpolation between houses.

However, the difference can be seen more clearly when considering higher-order statistics such as the mean. In Fig. 6, we compute the mean value using our method in data space, where the last row is the images over which the mean is calculated, as well as the mean for alternative methods by applying the Euclidean and spherical geometry in noise space. We see that spherical interpolation deviates more from the data images in Fig. 7 compared to our method. In Appendix D, we show the transition from the mean to the data points for all methods, where we also see that the spherical geometry has very rapid transitions to the data from the mean. There are several plausible reasons why the spherical geometry yields a mean that decodes to a black cat, whereas our method yields a tabby cat. The primary reason is that the two methods solve different optimization problems and, therefore, estimate different means. The spherical approach computes the intrinsic Fréchet mean under the spherical Riemannian metric, whereas our method computes a Fréchet mean under a Euclidean metric regularized by the log-density of the $\chi^2$-distribution on the norms of the curve points. In high dimensions, this regularization encourages norms

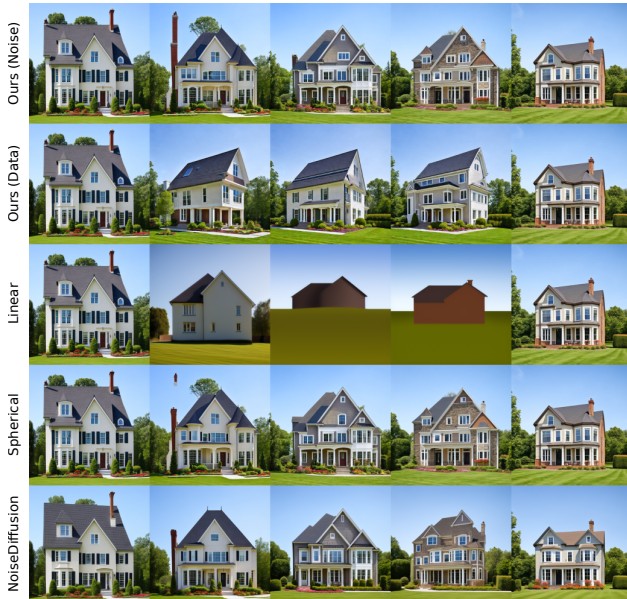

Figure 7: Interpolated images for different methods.

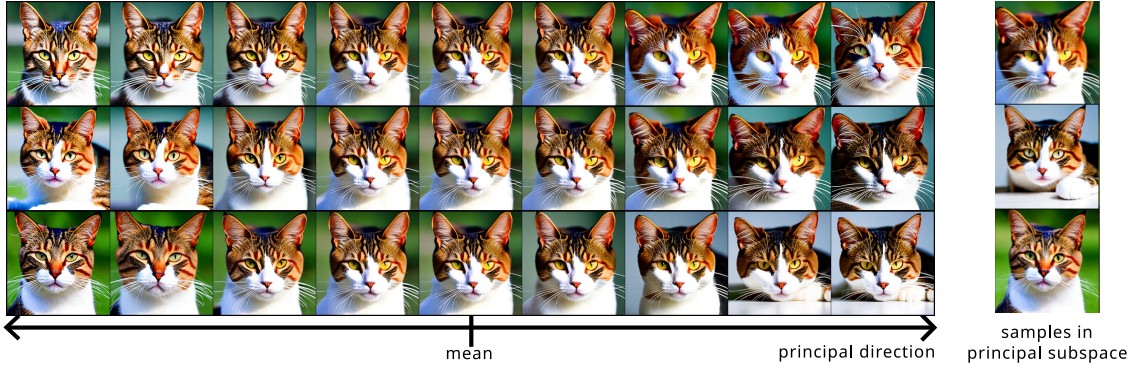

Figure 8: Three principal geodesics shown on the left. On the right, samples drawn within the principal subspace are shown.

consistent with the typical Gaussian shell while still allowing radial variation, whereas the spherical formulation assumes a fixed norm throughout the optimization. Consequently, the two approaches need not produce the same representative point in the diffusion noise space. Since the mapping from diffusion noise to images is highly nonlinear, even relatively small differences between the estimated means can lead to noticeable semantic differences in the generated images. A secondary factor is that the two methods imply different optimization procedures. For the spherical geometry, we can efficiently estimate the Fréchet mean in the embedded space iteratively using the gradient

$$\sum_{i=1}^{N} \mathrm{Log}_\mu(z_i)$$

For our method, we use *ProbGEOCRCE*. This also means that it is possible to obtain a difference due to the different optimization algorithms.

To further illustrate the potential of our method, we consider computing PGA (Fletcher et al., 2004) using our method, which generalizes principal component analysis to Riemannian manifolds. We consider the 10 images in Fig. 6 as data and display the three principal geodesics from the mean using our method in noise space and three samples using the 3 principal directions. In this way, we can sample in the principal directions of samples generated by the generative model.

In Appendix F, we provide additional qualitative results and also results for other diffusion models and score-based generative models (Song et al., 2021b), where we also show results for different regularization functions $S$.

## 5   Conclusion

We have proposed a general geometric framework to compute geometric statistics of generative models compatible with different metrics and probability distributions. We have shown that for geodesics to be within high-likelihood areas does not correspond to a Riemannian metric, but instead a Newtonian system on a Riemannian manifold. We have shown that shortest curves are characterized by a system of ODE, and that this locally along the optimal curves corresponds to geodesics under a Riemannian metric. We have derived an algorithm with fast convergence to estimate interpolation and the mean value of generative models. Empirically, we have shown our method's applicability to different generative models and demonstrated that our interpolation curves are closer to regions with high likelihood. A key benefit of our approach is that it allows for higher-order statistical calculations, where we have demonstrated means and principal components. This shows that through a rigorous geometric construction, we can incorporate contemporary generative models into more traditional data analysis pipelines.

**Limitations.**   Our method shows promising results for computing interpolation for different generative models but requires solving an optimization problem, which is more cumbersome than alternative methods

such as linear and spherical interpolation. Our method assumes a function that targets the likely areas of the generative model and that data resides on a Riemannian manifold. Furthermore, the optimal weight of the regularization $\lambda$ and the choice of the regularization function depend on the preferences of the user for smoothness versus high likelihood.

**Broader Impact Statement**

This work provides a geometrical framework for further statistical analysis of generative models enabling, e.g., interpolation. We have shown examples on images and synthetic data. Although this framework itself does not directly have any negative broader impact, it is possible to apply this framework to misuse a generative model. Since the framework is strongly dependent on the underlying generative model, any misuse can be avoided by imposing suitable restrictions on the generative model.

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

# Appendix

# A Proofs and Derivations

## A.1 Local representation of the metric

**Proposition A.1** (Local Metric). *Let $\tilde{S}$ denote the lower bound of $S$. Assume that $\lambda$ is sufficiently large such that $S(\cdot)$ is close to $\tilde{S}$ in Eq. 4. Let $\boldsymbol{\gamma}^*$ denote the optimal solution to Eq. 7. Then the regularized energy can locally along the optimal curve $\boldsymbol{\gamma}^*$ be estimated as*

$$\mathcal{E}(\boldsymbol{\gamma}) \approx \tilde{S} + \int_0^1 (\dot{\boldsymbol{\gamma}}^*)^\top (t) \left( \boldsymbol{G}\left(\boldsymbol{\gamma}^*(t)\right) + \frac{\lambda}{2} \boldsymbol{H}(S)\left(\boldsymbol{\gamma}^*(t)\right) \right) \dot{\boldsymbol{\gamma}}^*(t)\, \mathrm{d}t,$$

*where $\boldsymbol{H}(S)$ denotes the Hessian of $S$. Thus, if*

$$\boldsymbol{G}\left(\boldsymbol{\gamma}^*(t)\right) + \frac{\lambda}{2} \boldsymbol{H}(S)\left(\boldsymbol{\gamma}^*(t)\right)$$

*is positive definite, we can interpret it as a local representation of a Riemannian metric along the optimal curve.*

*Proof.* Let $\boldsymbol{\gamma}^*$ denote the optimal solution to Eq. 7. A Taylor expansion of $S$ around a point $\boldsymbol{\gamma}^*(t)$ gives the approximation

$$S\left(\boldsymbol{\gamma}^*(t + \Delta t)\right) = S\left(\boldsymbol{\gamma}^*(t)\right) + \langle \partial_{\boldsymbol{z}} S\left(\boldsymbol{\gamma}^*(t)\right), \Delta \boldsymbol{z}\rangle + \frac{1}{2}\Delta \boldsymbol{z}_i^\top \boldsymbol{H}(S)\left(\boldsymbol{\gamma}^*(t)\right) \Delta \boldsymbol{z} + \mathcal{O}\left(\Delta \boldsymbol{z}\right)\|\Delta \boldsymbol{z}\|_2^2,$$

where $\Delta \boldsymbol{z}_i = \boldsymbol{\gamma}^*(t + \Delta t) - \boldsymbol{\gamma}^*(t)$ and $\partial_{\boldsymbol{z}} S\left(\boldsymbol{\gamma}^*(t)\right)$ and $\boldsymbol{H}(S)\left(\boldsymbol{\gamma}^*(t)\right)$ denote the gradient and Hessian of $S(\cdot)$, respectively. By assumption, $S\left(\boldsymbol{\gamma}^*(t)\right)$ is close to the lower bound $\tilde{S}$, which implies that $\partial_{\boldsymbol{z}} S\left(\boldsymbol{\gamma}^*(t)\right) \approx \boldsymbol{0}$. Similarly, we set $S\left(\boldsymbol{\gamma}^*(t)\right) \approx \tilde{S}$. The regularized energy in Eq. 7 can therefore locally be written as

$$\mathcal{E}(\boldsymbol{\gamma}) \approx \tilde{S} + \int_0^1 \left( (\dot{\boldsymbol{\gamma}}^*)^\top (t) \boldsymbol{G}\left(\boldsymbol{\gamma}^*(t)\right) \dot{\boldsymbol{\gamma}}^*(t) + (\dot{\boldsymbol{\gamma}}^*)^\top (t) \frac{\lambda}{2} \boldsymbol{H}(S)(\boldsymbol{\gamma}(t))\dot{\boldsymbol{\gamma}}^*(t) \right)\, \mathrm{d}t$$

$$= \tilde{S} + \int_0^1 (\dot{\boldsymbol{\gamma}}^*)^\top (t) \left( \boldsymbol{G}\left(\boldsymbol{\gamma}^*(t)\right) + \frac{\lambda}{2} \boldsymbol{H}(S)\left(\boldsymbol{\gamma}^*(t)\right) \right) \dot{\boldsymbol{\gamma}}^*(t)\, \mathrm{d}t,$$

which completes the proof. $\qquad \square$

## A.2 ODE for minimizing curves

**Proposition A.2** (First variation). *Consider the regularized energy in Eq. 7 for a Riemannian manifold $(\mathcal{M}, g)$. Let $g_{ij}$ and $g^{ij}$ denote the local coordinates of metric matrix function and its inverse, respectively. The first variation gives the following* ODE

$$\ddot{\boldsymbol{\gamma}}^k(t) + \boldsymbol{\gamma}_{ij}^k \dot{\boldsymbol{\gamma}}^s(t)\dot{\boldsymbol{\gamma}}^j(t) = \frac{\lambda}{2} g^{kp} \partial_p S\left(\boldsymbol{\gamma}(t)\right), \tag{17}$$

*where $\boldsymbol{\gamma}_{ij}^k = \frac{1}{2} g^{kj}\left(\partial_l g_{pj} + \partial_j g_{pl} - \partial_p g_{ij}\right)$ denotes the Christoffel symbols derived from the Levi-Civita connection.*

*Proof.* Consider the energy written in Einstein notation as

$$\mathcal{E}\left(\boldsymbol{\gamma}\right) = \frac{1}{2}\int_a^b g_{ij}\left(\boldsymbol{\gamma}(t)\right) \dot{\boldsymbol{\gamma}}^i(t)\dot{\boldsymbol{\gamma}}^j(t)\, \mathrm{d}t + \lambda \int_a^b S\left(\boldsymbol{\gamma}(t)\right)\, \mathrm{d}t,$$

where $g_{ij}$ denotes the $(i, j)$th element of $\boldsymbol{G}$. We implicitly assume that $g_{ij}$ depends on $\boldsymbol{\gamma}(t)$. We aim to derive the corresponding system of ordinary differential equations (ODE) governing the shortest curves using calculus of variation. The corresponding Lagrangian is given by

$$L\left(\boldsymbol{x}, \dot{\boldsymbol{x}}\right) = g_{ij}\dot{\boldsymbol{x}}^i \dot{\boldsymbol{x}}^j + \lambda S,$$

where we to shorten notation set $\boldsymbol{x} := \boldsymbol{\gamma}(t)$ and $S := S\left(\boldsymbol{\gamma}(t)\right)$ and implicitly assume the dependence on $t$. From calculus of variation we get the Euler-Lagrange equation

$$\frac{\mathrm{d}}{\mathrm{d}t}\left(\frac{\partial L}{\partial \dot{\boldsymbol{x}}^p}\right) - \frac{\partial L}{\partial \boldsymbol{x}^p} = 0.$$

Since $\boldsymbol{G}$ is symmetric, we see that

$$\begin{aligned}
\frac{\mathrm{d}}{\mathrm{d}t}\left(\frac{\partial L}{\partial \dot{\boldsymbol{x}}^p}\right) &= \frac{\mathrm{d}}{\mathrm{d}t}\left(\frac{\partial}{\partial \dot{\boldsymbol{x}}^p}\left(\frac{1}{2}g_{ij}\dot{\boldsymbol{x}}^i\dot{\boldsymbol{x}}^j + \lambda S\right)\right) \\
&= \frac{\mathrm{d}}{\mathrm{d}t}\left(2g_{pj}\left(\boldsymbol{x}\right)\dot{\boldsymbol{x}}^j\right) \\
&= 2\partial_l g_{pj}\left(\boldsymbol{x}\right)\dot{\boldsymbol{x}}^l\dot{\boldsymbol{x}}^j + 2g_{pj}\ddot{\boldsymbol{x}}^j.
\end{aligned}$$

$$\begin{aligned}
\frac{\partial L}{\partial \boldsymbol{x}^p} &= \frac{\partial}{\partial \boldsymbol{x}^p}\left(g_{ij}\dot{\boldsymbol{x}}^i\dot{\boldsymbol{x}}^j + \lambda S\right) \\
&= \partial_p g_{ij}\dot{\boldsymbol{x}}^i\dot{\boldsymbol{x}}^j + \lambda\partial_p S.
\end{aligned}$$

Combining this, we get that

$$2\partial_l g_{pj}\dot{\boldsymbol{x}}^l\dot{\boldsymbol{x}}^j + 2g_{pj}\ddot{\boldsymbol{x}}^j - \partial_p g_{ij}\dot{\boldsymbol{x}}^i\dot{\boldsymbol{x}}^j - \lambda\partial_p S = 0$$

Let $g^{pk}$ denote the elements of the inverse to $\boldsymbol{G}$. Multiplying $g^{pk}$ on both sides, we get

$$2g^{pk}\partial_l g_{pj}\dot{\boldsymbol{x}}^l\dot{\boldsymbol{x}}^j + 2\ddot{\boldsymbol{x}}^k - g^{pk}\partial_p g_{ij}\left(\boldsymbol{x}\right)\dot{\boldsymbol{x}}^i\dot{\boldsymbol{x}}^j = \lambda g^{pk}\partial_p S,$$

where identity the Christoffel symbols as $\boldsymbol{\gamma}_{ij}^k = \frac{1}{2}g^{kj}\left(\partial_l g_{pj} + \partial_j g_{pl} - \partial_p g_{ij}\right)$ such that

$$2\ddot{\boldsymbol{x}}^k + 2\boldsymbol{\gamma}_{ij}^k\dot{\boldsymbol{x}}^i\dot{\boldsymbol{x}}^j = \lambda g^{kp}\partial_p S,$$

$\square$

## A.3 Necessary conditions for ProbGEORCE

The optimal control problem in Eq. 11 gives rise to the following necessary conditions for a minimum point.

**Proposition A.3.** *The necessary conditions for a minimum in Eq. 11 are*

$$\begin{aligned}
&2\boldsymbol{G}(\boldsymbol{z}_s)\boldsymbol{u}_s + \boldsymbol{\mu}_s = 0, \quad s = 0, \ldots, N_{\text{grid}} - 1, \\
&\boldsymbol{z}_{s+1} = \boldsymbol{z}_s + \boldsymbol{u}_s, \quad s = 0, \ldots, N_{\text{grid}} - 1, \\
&\nabla_y\left[\boldsymbol{u}_s^\top\boldsymbol{G}(y)\boldsymbol{u}_s + \lambda S(y)\right]\big|_{y=\boldsymbol{z}_s} + \boldsymbol{\mu}_s = \boldsymbol{\mu}_{s-1}, \quad s = 1, \ldots, N_{\text{grid}} - 1. \\
&\boldsymbol{z}_0 = \boldsymbol{a}, \boldsymbol{z}_{N_{\text{grid}}} = \boldsymbol{b},
\end{aligned} \tag{18}$$

*where $\boldsymbol{\mu}_s \in \mathbb{R}^d$ for $s = 0, \ldots, N_{\text{grid}} - 1$.*

*Proof.* We prove the necessary conditions using the same approach as in (Rygaard & Hauberg, 2025) for the regularized energy function by exploiting Pontryagin's maximum principle. Define the Hamiltonian of the control problem in eq. 11 as

$$H_s(\boldsymbol{z}_s, \boldsymbol{u}_s, \boldsymbol{\mu}_s) = \boldsymbol{u}_s^\top\boldsymbol{G}(\boldsymbol{z}_s)\boldsymbol{u}_s + \lambda S(\boldsymbol{z}_s) + \boldsymbol{\mu}_s^\top(\boldsymbol{z}_s + \boldsymbol{u}_s),$$

which by the time discrete version of Pontryagin's maximum principle gives the following optimization problem

$$\begin{aligned}
\min_{\boldsymbol{u}_s} \quad &\sum_{s=0}^{N_{\text{grid}}-1} H_s(\boldsymbol{z}_s, \boldsymbol{u}_s, \boldsymbol{\mu}_s) \\
\text{s.t.} \quad &\boldsymbol{z}_{s+1} = \boldsymbol{z}_s + \boldsymbol{u}_s, \quad s = 0, \ldots, N_{\text{grid}} - 1 \\
&\nabla_{\boldsymbol{z}_s} H_s(\boldsymbol{z}_s, \boldsymbol{u}_s, \boldsymbol{\mu}_s) = \boldsymbol{\mu}_{s-1}, \quad s = 0, \ldots, N_{\text{grid}} - 1 \\
&\boldsymbol{z}_0 = \boldsymbol{a}, \ \boldsymbol{z}_T = \boldsymbol{b}.
\end{aligned} \tag{19}$$

Since $\boldsymbol{G}(\boldsymbol{z}_s)$ is positive definite, $H_s(\boldsymbol{z}_s, \boldsymbol{u}_s, \boldsymbol{\mu}_s)$ is convex in $\boldsymbol{u}_s$, and therefore the stationary point $\boldsymbol{u}_s$ is also a global minimum point for $s = 0, \dots, N_{\text{grid}} - 1$. This gives the following equations for the control problem

$$
\begin{aligned}
&2\boldsymbol{G}(\boldsymbol{z}_s)\boldsymbol{u}_s + \boldsymbol{\mu}_s = 0, \quad s = 0, \dots, N_{\text{grid}} - 1, \\
&\boldsymbol{z}_{s+1} = \boldsymbol{z}_s + \boldsymbol{u}_s, \quad s = 0, \dots, N_{\text{grid}} - 1, \\
&\nabla_{\boldsymbol{z}_s}\left(\boldsymbol{u}_s^\top \boldsymbol{G}_s\left(\boldsymbol{z}_s\right)\boldsymbol{u}_s + \lambda S(\boldsymbol{z}_s)\right) + \boldsymbol{\mu}_s = \boldsymbol{\mu}_{s-1}, \quad s = 1, \dots, N_{\text{grid}} - 1, \\
&\boldsymbol{z}_0 = \boldsymbol{a}, \, \boldsymbol{z}_{N_{\text{grid}}} = \boldsymbol{b}.
\end{aligned}
$$

$\square$

The necessary conditions in Eq. 12 can generally not be solved with respect to $\boldsymbol{u}_{0:(N_{\text{grid}}-1)}$ and $\boldsymbol{\mu}_{0:(N_{\text{grid}}-1)}$. However, we can circumvent this iteratively. In iteration $k$ consider the state and control variables $\boldsymbol{z}^{(k)}_{0:N_{\text{grid}}}$ and $\boldsymbol{u}^{(k)}_{0:(N_{\text{grid}}-1)}$. We fix $\boldsymbol{G}(\cdot)$ and the gradient term in iteration $k$ to define the following variables.

$$
\begin{aligned}
&\boldsymbol{\nu}_s := \nabla_y \left(\boldsymbol{u}_s^\top \boldsymbol{G}(y)\boldsymbol{u}_s + \lambda S(y)\right)\big|_{y=\boldsymbol{z}^{(k)}_s, \boldsymbol{u}_s = \boldsymbol{u}^{(k)}_s}, \quad s = 1, \dots, N_{\text{grid}} - 1, \\
&\boldsymbol{G}_s := \boldsymbol{G}\left(\boldsymbol{z}^{(k)}_s\right), \quad s = 0, \dots, N_{\text{grid}} - 1.
\end{aligned}
\tag{20}
$$

With these variables fixed, the system of equations in Eq. 12 reduces to

$$
\begin{aligned}
&2\boldsymbol{G}_s\boldsymbol{u}_s + \boldsymbol{\mu}_s = 0, \quad s = 0, \dots, N_{\text{grid}} - 1, \\
&\boldsymbol{\nu}_s + \boldsymbol{\mu}_s = \boldsymbol{\mu}_{s-1}, \quad s = 1, \dots, N_{\text{grid}} - 1, \\
&\sum_{s=0}^{N_{\text{grid}}-1} \boldsymbol{u}_s = \boldsymbol{b} - \boldsymbol{a},
\end{aligned}
\tag{21}
$$

We see that this system is identical to the one in Rygaard & Hauberg (2025), from which we immediately have the update scheme in Eq. 14 with the modified version of $\boldsymbol{\nu}$.

### A.4 Convergence for ProbGEORCE

We will assume the same regularity of the modified energy in eq. 5 as in (Rygaard & Hauberg, 2025) with extra conditions on the regularizing function $S$. We state these below following the same outline as in (Rygaard & Hauberg, 2025).

**Assumption A.4** (Local convergence). *We assume the following regarding the discretized energy in Eq. 11 for the proof of local quadratic convergence.*

- *We assume that the discretized energy functional $E(\boldsymbol{z})$ is locally strictly convex in the (local) minimum point $\boldsymbol{x}^* = (\boldsymbol{z}^*, \boldsymbol{u}^*)$ in the sense that*

$$
\exists \epsilon > 0 : \forall \boldsymbol{x} \in B_\epsilon\left(\boldsymbol{x}^*\right), \boldsymbol{x} \neq \boldsymbol{x}^* : \forall \alpha]0, 1[: E\left((1-\alpha)\boldsymbol{x} + \alpha\boldsymbol{x}^*\right) < (1-\alpha)E(\boldsymbol{x}) + \alpha E\left(\boldsymbol{x}^*\right),
$$

  *where $B_\epsilon = \{\boldsymbol{x} \mid \|\boldsymbol{x} - \boldsymbol{x}^*\| < \epsilon\}$.*

- *Assume that the discretized energy functional $E(\boldsymbol{x})$ is locally Lipschitz, and consider the first order Taylor approximation of the discretized energy functional*

$$
\Delta E = \langle \nabla E(\boldsymbol{x}_0), \Delta\boldsymbol{x}\rangle + \mathcal{O}\left(\Delta\boldsymbol{x}\right)\|\Delta\boldsymbol{x}\|,
$$

- *We assume that the boundary value points are not conjugate points with respect to the Riemannian metric $g$. We assume a critical point of Eq. 7 is a (local) minimum point.*

In this section, we generalize the proofs for convergence in (Rygaard & Hauberg, 2025) to include the regularization of the energy.

**Proposition A.5** (Global Convergence). *Under the assumptions in Assumptions A.4, then ProbGEORCE in algorithm 3 has global convergence to a (local) minimum assuming $E(\boldsymbol{z})$ at the critical is locally strictly convex.*

*Proof.* *ProbGEORCE* fulfills that

$$\begin{aligned}
\nabla_{\boldsymbol{z}_s} E(\boldsymbol{z}, \boldsymbol{u}) \left( \boldsymbol{z}_s^{(k)}, \boldsymbol{u}_s^{(k)} \right) &= \boldsymbol{\mu}_{s-1} - \boldsymbol{\mu}_s, \quad s = 1, \ldots, N_{\text{grid}} - 1, \\
\nabla_{\boldsymbol{u}_s} E(\boldsymbol{z}, \boldsymbol{u}) \left( \boldsymbol{z}_s^{(k)}, \boldsymbol{u}_s^{(k+1)} \right) &= -\boldsymbol{\mu}_s, \quad s = 0, \ldots, N_{\text{grid}} - 1,
\end{aligned} \tag{22}$$

where

$$E(\boldsymbol{z}, \boldsymbol{u}) := \sum_{s=0}^{N_{\text{grid}} - 1} \left( \boldsymbol{u}_s^\top \boldsymbol{G}(\boldsymbol{z}_s) \boldsymbol{u}_s + \lambda S(\boldsymbol{z}_s) \right),$$

and $\boldsymbol{z}_s^{(k)}, \boldsymbol{u}_s^{(k)}$ are the state and control variables in iteration $k$, respectively. These properties are identical to the properties for the *GEORCE*-algorithm in (Rygaard & Hauberg, 2025), and therefore the proof for global convergence for the *GEORCE* algorithm also holds for *ProbGEORCE*. $\square$

In general, the local asymptotic quadratic convergence can not be extended to the regularized geodesic case unless $\lambda \boldsymbol{H}_S$ is the zero matrix, where $\boldsymbol{H}_S$ is the Hessian of $S$. This means in practice that the asymptotic local quadratic convergence of *ProbGEORCE* will only hold approximately for low values of $\lambda$, or if the Hessian of $S$ is small.

In the following, we generalize the results in (Rygaard & Hauberg, 2025) to *ProbGEORCE* to show that *ProbGEORCE* has local asymptotic quadratic convergence when $\lambda \boldsymbol{H}_s$ is close to the zero matrix.

**Lemma A.6.** *Let $z_t := (\boldsymbol{x}_t, \boldsymbol{u}_t)$ and $\|\boldsymbol{u}_t\|_{\boldsymbol{x}_t}^2 = \boldsymbol{u}_t^\top \boldsymbol{G}(\boldsymbol{x}_t) \boldsymbol{u}_t + \lambda S(\boldsymbol{x}_t)$, then the solution to the following optimal control problem is identical to a ProbGEORCE step without line-search, i.e. $\alpha = 1$.*

$$\begin{aligned}
\min_{(\boldsymbol{x}_t, \boldsymbol{u}_t)} \sum_{t=0}^{T-1} & \left\| \boldsymbol{u}_t^{(i)} \right\|_{\boldsymbol{x}_t^{(i)}}^2 + \left( \nabla \left\| \boldsymbol{u}_t^{(i)} \right\|_{\boldsymbol{x}_t^{(i)}}^2 \right)^\top \begin{pmatrix} \boldsymbol{x}_t - \boldsymbol{x}_t^{(i)} \\ \boldsymbol{u}_t - \boldsymbol{u}_t^{(i)} \end{pmatrix} \\
& + \frac{1}{2} \begin{pmatrix} \boldsymbol{x}_t - \boldsymbol{x}_t^{(i)} & \boldsymbol{u}_t - \boldsymbol{u}_t^{(i)} \end{pmatrix} Q(\boldsymbol{x}_t^{(i)}, \boldsymbol{u}_t^{(i)}) \begin{pmatrix} \boldsymbol{x}_t - \boldsymbol{x}_t^{(i)} \\ \boldsymbol{u}_t - \boldsymbol{u}_t^{(i)} \end{pmatrix}
\end{aligned} \tag{23}$$

$$\begin{aligned}
s.t. \quad & \boldsymbol{x}_{t+1} = \boldsymbol{x}_t + \boldsymbol{u}_t, \quad t = 0, \ldots, T-1, \\
& \boldsymbol{x}_0 = a, \ \boldsymbol{x}_T = b,
\end{aligned}$$

*where*

$$Q(\boldsymbol{x}_t, \boldsymbol{u}_t) = \begin{pmatrix} \boldsymbol{0} & \boldsymbol{0} \\ \boldsymbol{0} & 2G(\boldsymbol{x}_t) \end{pmatrix}$$

*Proof.* The Hamiltonian function for Eq. 23 is

$$\begin{aligned}
H_t(\boldsymbol{x}_t, \boldsymbol{u}_t, \mu_t) = & \left\| \boldsymbol{u}_t^{(i)} \right\|_{\boldsymbol{x}_t^{(i)}}^2 + \left( \nabla \left\| \boldsymbol{u}_t^{(i)} \right\|_{\boldsymbol{x}_t^{(i)}}^2 \right)^\top \begin{pmatrix} \boldsymbol{x}_t - \boldsymbol{x}_t^{(i)} \\ \boldsymbol{u}_t - \boldsymbol{u}_t^{(i)} \end{pmatrix} \\
& + \frac{1}{2} \begin{pmatrix} \boldsymbol{x}_t - \boldsymbol{x}_t^{(i)} & \boldsymbol{u}_t - \boldsymbol{u}_t^{(i)} \end{pmatrix} Q(\boldsymbol{x}_t^{(i)}, \boldsymbol{u}_t^{(i)}) \begin{pmatrix} \boldsymbol{x}_t - \boldsymbol{x}_t^{(i)} \\ \boldsymbol{u}_t - \boldsymbol{u}_t^{(i)} \end{pmatrix} + \mu_t^\top (\boldsymbol{x}_t + \boldsymbol{u}_t)
\end{aligned}$$

Applying Pontryagins' maximum principle to Eq. 23 gives

$$\begin{aligned}
\min_{\boldsymbol{u}_t} \quad & H_t(\boldsymbol{x}_t, \boldsymbol{u}_t, \mu_t) \\
s.t. \quad & \boldsymbol{x}_{t+1} = \boldsymbol{x}_t + \boldsymbol{u}_t, \quad t = 0, \ldots, T-1, \\
& \nabla_{\boldsymbol{x}_t} H_t(\boldsymbol{x}_t, \boldsymbol{u}_t, \mu_t) = \mu_{t-1}, \quad t = 1, \ldots, T-1.
\end{aligned}$$

As $H_t(\boldsymbol{x}_t, \boldsymbol{u}_t, \mu_t)$ is strictly convex in $\boldsymbol{u}_t$, then the optimal control $\boldsymbol{u}_t^*$ is the stationary point to the Hamiltonian function, i.e.

$$\nabla_{\boldsymbol{u}_t} H_t(\boldsymbol{x}_t, \boldsymbol{u}_t, \mu_t) = \nabla_{\boldsymbol{u}_t} E(\boldsymbol{x}_t^{(i)}, \boldsymbol{u}_t^{(i)}) + 2G(\boldsymbol{x}_t^{(i)})(\boldsymbol{u}_t - \boldsymbol{u}_t^{(i)}) + \mu_t = 0$$

$$\Leftrightarrow 2G(\boldsymbol{x}_t^{(i)})u_s^{(i)} + \mu_t = 0, \quad t = 0, \ldots, T-1,$$

since $\nabla_{\boldsymbol{u}_t^{(i)}} \left\| \boldsymbol{u}_t^{(i)} \right\|_2^2 = 2G(\boldsymbol{x}_t^{(i)})\boldsymbol{u}_t^{(i)}$. This is identical to the first equation in the equations for *ProbGEORCE*. The state equation is unchanged, and the co-state equation for the problem becomes

$$\nabla_{\boldsymbol{x}_t} H_t(\boldsymbol{x}_t, \boldsymbol{u}_t, \mu_t) = \nabla_{\boldsymbol{x}_t} \|\boldsymbol{u}_t\|_{\boldsymbol{x}_t}^2 + \mu_t = \mu_{t-1}, \quad t = 1, \ldots, T-1.$$

which is identical to the co-state equation of *ProbGEORCE*. $\qquad\square$

The optimality conditions can also be formulated as a linear system of equations similar to the original proof in (Rygaard & Hauberg, 2025), but the result for local quadratic convergence in *GEORCE* will generally not hold with regularization. This can be seen from the following error term between the Hessian for the regularized energy functional and $Q(\boldsymbol{x}_t, \boldsymbol{u}_t)$ in Lemma A.6

$$\nabla \|\boldsymbol{u}_t^*\|_{\boldsymbol{x}_t^*} - Q(\boldsymbol{x}_t^*, \boldsymbol{u}_t^*) = \begin{pmatrix} \nabla_{\boldsymbol{x}_t, \boldsymbol{x}_t}^2 \|\boldsymbol{u}_t^*\|_{\boldsymbol{x}_t^*} & \nabla_{\boldsymbol{x}_t, \boldsymbol{u}_t}^2 \|\boldsymbol{u}_t^*\|_{\boldsymbol{x}_t^*} \\ \nabla_{\boldsymbol{u}_t, \boldsymbol{x}_t}^2 \|\boldsymbol{u}_t^*\|_{\boldsymbol{x}_t^*} & \boldsymbol{0} \end{pmatrix}.$$

Consider the $\ell^\infty$ norm of the Hessian

$$\left\| \nabla_{\boldsymbol{x}_t, \boldsymbol{x}_t}^2 \|\boldsymbol{u}_t^*\|_{\boldsymbol{x}_t^*} \right\|_\infty = \left\| \nabla_{\boldsymbol{x}_t, \boldsymbol{x}_t}^2 \boldsymbol{u}_t^\top \boldsymbol{G}(\boldsymbol{x}_t)\boldsymbol{u}_t + \lambda S(\boldsymbol{x}_t) \big|_{(\boldsymbol{x}_t, \boldsymbol{u}_t)=(\boldsymbol{x}_t^*, \boldsymbol{u}_t^*)} \right\|_\infty$$

The local asymptotic convergence now follows from the exactly same approach as in (Rygaard & Hauberg, 2025), but only if the penalty term in the Hessian $\left\| \lambda \nabla_{\boldsymbol{x}_t, \boldsymbol{x}_t}^2 S(\boldsymbol{x}_t) \right\|_\infty$ converges to 0 when $T \to \infty$, since then $\left\| \nabla_{\boldsymbol{x}_t, \boldsymbol{x}_t}^2 S(\boldsymbol{x}_t) \right\|_\infty$ will only approach zero in the case that $\left\| \lambda \nabla_{\boldsymbol{x}_t, \boldsymbol{x}_t}^2 S(\boldsymbol{x}_t) \right\|_\infty = 0$. Thus, in practice the asymptotic local quadratic convergence will only be "approximately" observed if $S$ is approximately an affine function or if $\lambda$ is low.

Note that for a general function $S$, it cannot be ruled out that the solution to *ProbGEORCE* has converged to a saddle point. However, the regularized energy will obtain a value lower or equal to the value of the starting point. If the regularized energy is locally strictly convex at the solution, *ProbGEORCE* will converge to a local minimum point.

## A.5 Adaptive update scheme for ProbGEORCE

*ADAM* adaptively updates the step-size in gradient descent using higher-order variance of the gradient (Kingma & Ba, 2014). This results in the following adaptive scheme in iteration $k$

$$\boldsymbol{g}_k \leftarrow \nabla_{\boldsymbol{\theta}} f_k(\boldsymbol{\theta}_{k-1}),$$

$$\boldsymbol{m}_k \leftarrow \beta_1 \boldsymbol{m}_{k-1} + (1 - \beta_1)\boldsymbol{g}_k,$$

$$\boldsymbol{v}_k \leftarrow \beta_2 \boldsymbol{v}_{k-1} + (1 - \beta_2)\boldsymbol{g}_k^2,$$

$$\hat{\boldsymbol{m}}_k \leftarrow \frac{\boldsymbol{m}_k}{1 - \beta_2^k},$$

$$\hat{\boldsymbol{v}}_k \leftarrow \frac{\boldsymbol{v}_k}{1 - \beta_2^k},$$

$$\boldsymbol{\theta}_k \leftarrow \boldsymbol{\theta}_{k-1} - \gamma \frac{\hat{\boldsymbol{m}}_k}{\sqrt{\hat{\boldsymbol{v}}_k} + \epsilon},$$

where $f_k$ denotes the loss function in iteration $k$ and $\beta_1, \beta_2 > 0$ and $\gamma > 0$ are parameters, while $0 \leq \epsilon << 1$ is used for numerical stability. Now assume that we have a stochastic estimate of the energy in Eq. 11

$$\tilde{\boldsymbol{G}}(\boldsymbol{x}) + \lambda \tilde{S}(\boldsymbol{x}), \tag{24}$$

such that we allow $\tilde{G}$ or $\tilde{S}$ to be stochastic samples of $G$ and $S$. For $\tilde{\boldsymbol{\nu}}_s^{(k)}\left(\boldsymbol{z}_s^{(k)}\right) \leftarrow \nabla_{\boldsymbol{y}}\left(\boldsymbol{u}_s^{(k)}\tilde{G}\left(\boldsymbol{y}\right)\boldsymbol{u}_s^{(k)} + \lambda S\left(\boldsymbol{y}\right)\right)\Big|_{\boldsymbol{y}=\boldsymbol{z}_s^{(k)}}$ for $s = 1,\ldots,N_{\text{grid}}-1$, we can interpret $\sum_{s=0}^{N_{\text{grid}}-1}\|\tilde{\boldsymbol{\nu}}\|^2$ as the variance of the estimator in *ProbGEORCE*. By adaptively updating $\tilde{\boldsymbol{\nu}}$ and $\tilde{G}$, we can directly apply a similar update scheme as in *ADAM* on the form:

$$\boldsymbol{z}_s^{(k+1)} \leftarrow \boldsymbol{z}_s^{(k)} + \kappa\left(\boldsymbol{z}_s^{(k+1)} - \boldsymbol{z}_s^{(k)}\right),$$

$$\boldsymbol{u}_s^{(k+1)} \leftarrow \boldsymbol{u}_s^{(k)} + \kappa\left(\boldsymbol{u}_s^{(k+1)} - \boldsymbol{u}_s^{(k)}\right),$$

$$\tilde{G}_s^{(k+1)} \leftarrow (1-\beta_1)\tilde{G}_s\left(\boldsymbol{z}_s^{(k+1)}\right) + \beta_1\tilde{G}_s^{(k+1)},$$

$$\tilde{\boldsymbol{\nu}}_s^{(k+1)} \leftarrow (1-\beta_1)\tilde{\boldsymbol{\nu}}_s\left(\boldsymbol{z}_s^{(k+1)}\right) + \beta_1\tilde{\boldsymbol{\nu}}_s^{(k+1)},$$

$$\tilde{\boldsymbol{g}}^{(k+1)} \leftarrow (1-\beta_2)\left(\sum_{s=0}^{N_{\text{grid}}-1}\left\|\tilde{\boldsymbol{\nu}}_s^{(k+1)}\right\|^2\right) + \beta_2\tilde{\boldsymbol{g}}^{(k+1)},$$

$$\hat{G}_s^{(k+1)} = \frac{\tilde{G}_s^{(k+1)}}{1-\beta_1^{k+1}},$$

$$\hat{\boldsymbol{\nu}}_s^{(k+1)} = \frac{\tilde{\boldsymbol{\nu}}_s^{(k+1)}}{1-\beta_1^{k+1}},$$

$$\hat{\boldsymbol{g}}^{(k+1)} = \frac{\tilde{\boldsymbol{g}}^{(k+1)}}{1-\beta_2^{k+1}},$$

$$\kappa = \min\left\{\frac{\gamma}{\sqrt{1+\tilde{\boldsymbol{g}}^{(k+1)}}+\epsilon}, 1\right\},$$

where we cap the step size $\kappa$, since this can at most be one. We show the adaptive scheme in Algorithm 1, where ProbGEORCE denotes a step using Proposition 3.5 or Corollary 3.6. Note that if $\boldsymbol{G} = I$, we do not have to update $I$ in the algorithm. Below we apply the update scheme of *ADAM* due to its high-convergence in practice.

## A.6 Estimation of mean value

In this part, we show how to efficiently compute the Fréchet mean and minimize the connecting curves simultaneously by extending the results in Rygaard et al. (2025) to take into account the regularization term. We will follow the same approach as in Rygaard et al. (2025), and consider the control formulation of Eq. 10

$$\min_{(\boldsymbol{z}_{s,i},\boldsymbol{u}_{s,i})} E(\boldsymbol{x}) := \min_{(\boldsymbol{z}_{s,i},\boldsymbol{u}_{s,i})}\left\{\sum_{i=1}^{N_{\text{data}}} w_i\left(\sum_{s=0}^{N_{\text{grid}}-1}\boldsymbol{u}_{s,i}^{\top}\boldsymbol{G}(\boldsymbol{z}_{s,i})\boldsymbol{u}_{s,i} + \lambda S(\boldsymbol{z}_{s,i})\right)\right\}$$

$$\boldsymbol{z}_{s+1,i} = \boldsymbol{z}_{s,i} + \boldsymbol{u}_{s,i}, \quad s = 0,\ldots,N_{\text{grid}}-1, \, i = 1,\ldots,N_{\text{data}},$$

$$\boldsymbol{z}_{0,i} = \boldsymbol{a}_i, \boldsymbol{z}_{N_{\text{grid}},i} = \boldsymbol{y}, \quad i = 1,\ldots,N_{\text{data}}.$$

$$(25)$$

By a discrete-time version of Pontryagin's maximum principle, we arrive at the following

---

**Algorithm 1** Adaptive update scheme for ProbGEORCE

---

1: **Input**: tol, $N_{\text{grid}}$, $\gamma$, $\beta_1$, $\beta_2$, $\epsilon$
2: **Output**: Geodesic estimate $\boldsymbol{x}_{0:N_{\text{grid}}}$
3: Set $\boldsymbol{z}_s^{(0)} \leftarrow a + \frac{b-a}{N_{\text{grid}}} s$ for $s = 0., \ldots, N_{\text{grid}}$, $\boldsymbol{u}_s^{(0)} \leftarrow \frac{b-a}{N_{\text{grid}}}$ for $s = 0., \ldots, N_{\text{grid}} - 1$, $\kappa = \gamma$ and $k \leftarrow 0$
4: **while** stop criteria > tol **do**
5: $\quad \tilde{\boldsymbol{G}}_s^{(k)} \leftarrow \boldsymbol{G}_s\left(\boldsymbol{z}_s^{(k)}\right)$ for $s = 0, \ldots, N_{\text{grid}} - 1$
6: $\quad \tilde{\boldsymbol{\nu}}_s^{(k)}\left(\boldsymbol{z}_s^{(k)}\right) \leftarrow \nabla_{\boldsymbol{y}}\left(\boldsymbol{u}_s^{(k)}\tilde{\boldsymbol{G}}(\boldsymbol{y})\boldsymbol{u}_s^{(k)} + \lambda S(\boldsymbol{y})\right)\Big|_{\boldsymbol{y}=\boldsymbol{z}_s^{(k)}}$ for $s = 1, \ldots, N_{\text{grid}} - 1$
7: $\quad$ **if** $k = 0$ **then**
8:

$$\left\{\boldsymbol{z}_s^{(k+1)}\right\}_{s=0}^{N_{\text{grid}}-1}, \left\{\boldsymbol{u}_s^{(k+1)}\right\}_{s=0}^{N_{\text{grid}}-1}$$
$$= \text{ProbGEORCE}\left(\left\{\tilde{\boldsymbol{G}}_s^{(k)}\right\}_{s=0}^{N_{\text{grid}}-1}, \left\{\left(\tilde{\boldsymbol{G}}^{(k)}\right)^{-1}\right\}_{s=0}^{N_{\text{grid}}-1}, \left\{\tilde{\boldsymbol{\nu}}_s^{(k)}\right\}_{s=0}^{N_{\text{grid}}-1}, \left(\sum_{s=0}^{N_{\text{grid}}-1}\left(\tilde{\boldsymbol{G}}_s^{(k)}\right)^{-1}\right)^{-1}\right)$$

9: $\quad$ **else**
10:

$$\left\{\boldsymbol{z}_s^{(k+1)}\right\}_{s=0}^{N_{\text{grid}}-1}, \left\{\boldsymbol{u}_s^{(k+1)}\right\}_{s=0}^{N_{\text{grid}}-1}$$
$$= \text{ProbGEORCE}\left(\left\{\hat{\boldsymbol{G}}_s^{(k)}\right\}_{s=0}^{N_{\text{grid}}-1}, \left\{\left(\hat{\boldsymbol{G}}^{(k)}\right)^{-1}\right\}_{s=0}^{N_{\text{grid}}-1}, \left\{\hat{\boldsymbol{g}}^{(k)}\right\}_{s=0}^{N_{\text{grid}}-1}, \left(\sum_{s=0}^{N_{\text{grid}}-1}\left(\hat{\boldsymbol{G}}_s^{(k)}\right)^{-1}\right)^{-1}\right)$$

11: $\quad$ **end if**
12: $\quad \boldsymbol{z}_s^{(k+1)} \leftarrow \boldsymbol{z}_s^{(k)} + \kappa\left(\boldsymbol{z}_s^{(k+1)} - \boldsymbol{z}_s^{(k)}\right)$
13: $\quad \boldsymbol{u}_s^{(k+1)} \leftarrow \boldsymbol{u}_s^{(k)} + \kappa\left(\boldsymbol{u}_s^{(k+1)} - \boldsymbol{u}_s^{(k)}\right)$
14: $\quad \tilde{\boldsymbol{G}}_s^{(k+1)} \leftarrow (1 - \beta_1)\tilde{\boldsymbol{G}}_s\left(\boldsymbol{z}_s^{(k+1)}\right) + \beta_1 \tilde{\boldsymbol{G}}_s^{(k+1)}$
15: $\quad \tilde{\boldsymbol{\nu}}_s^{(k+1)} \leftarrow (1 - \beta_1)\tilde{\boldsymbol{\nu}}_s\left(\boldsymbol{z}_s^{(k+1)}\right) + \beta_1 \tilde{\boldsymbol{\nu}}_s^{(k+1)}$
16: $\quad \tilde{\boldsymbol{g}}^{(k+1)} \leftarrow (1 - \beta_2)\left(\sum_{s=0}^{N_{\text{grid}}-1}\left\|\tilde{\boldsymbol{\nu}}_s^{(k+1)}\right\|^2\right) + \beta_2 \tilde{\boldsymbol{g}}^{(k+1)}$
17: $\quad \hat{\boldsymbol{G}}_s^{(k+1)} = \frac{\tilde{\boldsymbol{G}}_s^{(k+1)}}{1-\beta_1^{k+1}}, \hat{\boldsymbol{\nu}}_s^{(k+1)} = \frac{\tilde{\boldsymbol{\nu}}_s^{(k+1)}}{1-\beta_1^{k+1}}, \hat{\boldsymbol{g}}^{(k+1)} = \frac{\tilde{\boldsymbol{g}}^{(k+1)}}{1-\beta_2^{k+1}}$
18: $\quad \kappa = \min\left\{\frac{\gamma}{\sqrt{1+\tilde{\boldsymbol{g}}^{(k+1)}}+\epsilon}, 1\right\}$
19: $\quad k \leftarrow k + 1$
20: **end while**
21: return $\boldsymbol{z}_s$ for $s = 0, \ldots, N_{\text{grid}} - 1$

---

**Proposition A.7.** *The necessary conditions for a minimum in Eq. 25 are*

$$2w_i \boldsymbol{G}(\boldsymbol{z}_{s,i})\boldsymbol{u}_{s,i} + \boldsymbol{\mu}_{s,i} = 0, \quad s = 0, \ldots, N_{\text{grid}} - 1, \, i = 1, \ldots, N_{\text{data}},$$

$$\boldsymbol{z}_{s+1,i} = \boldsymbol{z}_{s,i} + \boldsymbol{u}_{s,i}, \quad s = 0, \ldots, N_{\text{grid}} - 1, \, i = 1, \ldots, N_{\text{data}},$$

$$\nabla_{\boldsymbol{z}}\left[w_i \boldsymbol{u}_{s,i}^\top \boldsymbol{G}(\boldsymbol{z})\boldsymbol{u}_{s,i} + \lambda S(\boldsymbol{z})\right]\Big|_{\boldsymbol{z}=\boldsymbol{z}_{s,i}} + \boldsymbol{\mu}_{s,i} = \boldsymbol{\mu}_{s-1,i}, \quad s = 1, \ldots, N_{\text{grid}} - 1, \, i = 1, \ldots, N_{\text{grid}},$$

$$\tag{26}$$

$$0 = \sum_{i=1}^N \boldsymbol{\mu}_{N_{\text{grid}}-1,i},$$

$$\boldsymbol{z}_{0,i} = \boldsymbol{a}_i, \boldsymbol{z}_{N_{\text{grid}},i} = \boldsymbol{y}, \quad i = 1, \ldots, N_{\text{data}},$$

where $\boldsymbol{\mu}_{s,i} \in \mathbb{R}^d$ denotes the dual prices for the control problem in Eq. 25 for $s = 0, \ldots, N_{\text{grid}} - 1$ and $i = 1, \ldots, N_{\text{data}}$.

*Proof.* The Hamiltonian function of Eq. 25 is

$$H_s\left(\boldsymbol{z}_{s,i}, \boldsymbol{u}_{s,i}, \boldsymbol{\mu}_{s,i}\right) = \sum_{i=1}^{N_{\text{grid}}-1} H_{s,i}\left(\boldsymbol{z}_{s,i}, \boldsymbol{u}_{t,i}, \boldsymbol{\mu}_{t,i}\right),$$

$$H_{s,i}\left(\boldsymbol{z}_{s,i}, \boldsymbol{u}_{s,i}, \boldsymbol{\mu}_{s,i}\right) = w_i \boldsymbol{u}_{s,i}^\top \boldsymbol{G}(\boldsymbol{z}_{s,i}) \boldsymbol{u}_{s,i} + \boldsymbol{\mu}_{s,i}^\top \left(\boldsymbol{z}_{s,i} + \boldsymbol{u}_{s,i}\right). \tag{27}$$

Applying the time-discrete version of Pontryagin's maximum problem to Eq. 26, we get the following necessary conditions, where the endpoint $x_{N_{\text{grid}},i}$ is replaced by the mean point $\boldsymbol{y}$.

$$\min_{\boldsymbol{u}_{s,i}} \sum_{i=1}^{N_{\text{data}}} \sum_{s=0}^{N_{\text{grid}}-1} H_{s,i}(\boldsymbol{z}_{s,i}, \boldsymbol{u}_{s,i}, \boldsymbol{\mu}_{s,i}) \tag{28}$$

$$\text{s.t.} \quad \boldsymbol{z}_{s+1,i} = \boldsymbol{z}_{s,i} + \boldsymbol{u}_{s,i}, \quad s = 0, \ldots, N_{\text{grid}} - 1, \, i = 1, \ldots, N_{\text{data}}, \qquad \text{(state equation)}, \tag{29}$$

$$\boldsymbol{y} = \boldsymbol{z}_{N_{\text{grid}}-1,i} + \boldsymbol{u}_{N_{\text{grid}}-1,i}, \quad i = 1, \ldots, N_{\text{data}}, \qquad \text{(state equation)}, \tag{30}$$

$$\nabla_{\boldsymbol{z}_{s,i}} H_{s,i}(\boldsymbol{z}_{s,i}, \boldsymbol{u}_{s,i}, \boldsymbol{\mu}_{s,i}) = \boldsymbol{\mu}_{s-1,i}, \quad s = 1, \ldots, N_{\text{grid}} - 1, \, i = 1, \ldots, N_{\text{data}} \quad \text{(co-state equation)}, \tag{31}$$

$$0 = \sum_{i=1}^{N_{\text{data}}} \boldsymbol{\mu}_{N_{\text{grid}}-1,i}, \quad i = 1, \ldots, N_{\text{data}}, \qquad \text{(co-state equation)}, \tag{32}$$

$$x_{0,i} = \boldsymbol{a}_i, \quad i = 1, \ldots, N_{\text{data}}. \tag{33}$$

We can decompose the minimization problem into the following sub-problems for each $i \in \{1, \ldots, N_{\text{data}}\}$

$$\min_{\boldsymbol{u}_{s,i}} H_{s,i}\left(\boldsymbol{z}_{s,i}, \boldsymbol{u}_{s,i}, \boldsymbol{\mu}_{s,i}\right), \quad s = 0, \ldots, N_{\text{grid}} - 1, \, i = 1, \ldots, N_{\text{data}}.$$

Since $\boldsymbol{G}(\boldsymbol{z}_{s,i})$ is positive definite, $H_{t,i}(x_{t,i}, \boldsymbol{u}_{t,i}, \boldsymbol{\mu}_{t,i})$ is strictly convex in $\boldsymbol{u}_{t,i}$. Thus, the stationary point in $\boldsymbol{u}_{t,i}$ is also a global minimum minimum. We therefore get the following necessary conditions for a minimum.

$$2 w_i \boldsymbol{G}(\boldsymbol{z}_{s,i}) \boldsymbol{u}_{s,i} + \boldsymbol{\mu}_{s,i} = 0, \quad s = 0, \ldots, N_{\text{grid}} - 1, \, i = 1, \ldots, N_{\text{data}},$$

$$\boldsymbol{z}_{s+1,i} = \boldsymbol{z}_{s,i} + \boldsymbol{u}_{s,i}, \quad s = 0, \ldots, N_{\text{grid}} - 1, \, i = 1, \ldots, N_{\text{data}},$$

$$\nabla_{\boldsymbol{z}} \left[ w_i \boldsymbol{u}_{s,i}^\top \boldsymbol{G}(\boldsymbol{z}) \boldsymbol{u}_{s,i} + \lambda S(\boldsymbol{z}) \right] \big|_{\boldsymbol{z}=\boldsymbol{z}_{s,i}} + \boldsymbol{\mu}_{s,i} = \boldsymbol{\mu}_{s-1,i}, \quad s = 1, \ldots, N_{\text{grid}} - 1, \, i = 1, \ldots, N_{\text{data}}, \tag{34}$$

$$0 = \sum_{i=1}^{N} \boldsymbol{\mu}_{N_{\text{grid}}-1,i},$$

$$\boldsymbol{z}_{0,i} = \boldsymbol{a}_i, \boldsymbol{z}_{N_{\text{grid}},i} = \boldsymbol{y}, \quad i = 1, \ldots, N_{\text{data}}.$$

$\square$

We fix the following variables in iteration $k$ similar to Eq. 13 and Rygaard et al. (2025):

$$\boldsymbol{\nu}_{s,i} := \nabla_{\boldsymbol{z}} \left( w_i \boldsymbol{u}_{s,i}^\top \boldsymbol{G}(\boldsymbol{z}) \boldsymbol{u}_{s,i} + \lambda S(\boldsymbol{z}) \right) \big|_{\boldsymbol{z}=\boldsymbol{z}_{s,i}^{(k)}, \boldsymbol{u}_{s,i}=\boldsymbol{u}_{s,i}^{(k)}}, \quad s = 1, \ldots, N_{\text{grid}} - 1, \, i = 1, \ldots, N_{\text{data}},$$

$$\boldsymbol{G}_{s,i} := \boldsymbol{G}\left(\boldsymbol{z}_{s,i}^{(k)}\right), \quad s = 0, \ldots, N_{\text{grid}} - 1, \, i = 1, \ldots, N_{\text{data}}. \tag{35}$$

In this way, the necessary conditions reduce to:

$$2 w_i \boldsymbol{G}_{s,i} \boldsymbol{u}_{s,i} + \boldsymbol{\mu}_{s,i} = 0, \quad s = 0, \ldots, N_{\text{grid}} - 1, \, i = 1, \ldots, N_{\text{data}},$$

$$\boldsymbol{z}_{s+1,i} = \boldsymbol{z}_{s,i} + \boldsymbol{u}_{s,i}, \quad s = 0, \ldots, N_{\text{grid}} - 1, \, i = 1, \ldots, N_{\text{data}},$$

$$\boldsymbol{\nu}_{s,i} + \boldsymbol{\mu}_{s,i} = \boldsymbol{\mu}_{s-1,i}, \quad s = 1, \ldots, N_{\text{grid}} - 1, \, i = 1, \ldots, N_{\text{data}}, \tag{36}$$

$$0 = \sum_{i=1}^{N} \boldsymbol{\mu}_{N_{\text{grid}}-1,i},$$

$$\boldsymbol{z}_{0,i} = \boldsymbol{a}_i, \boldsymbol{z}_{N_{\text{grid}},i} = \boldsymbol{y}, \quad i = 1, \ldots, N_{\text{data}}.$$

Eq. 36 is completely similar to (Rygaard et al., 2025) with the modification that $\boldsymbol{\nu}_{s,i}$ depends on $S$, which has the closed-form solution.

**Proposition A.8.** *The update scheme of* $\boldsymbol{u}_{s,i}, \boldsymbol{\mu}_{s,i}$ *and* $\boldsymbol{z}_{s,i}$ *to minimize Eq. 25 is*

$$
\begin{aligned}
\boldsymbol{y} &= W^{-1}V, \\
\boldsymbol{\mu}_{N_{\text{grid}}-1,i} &= \left( \sum_{s=0}^{N_{\text{grid}}-1} \boldsymbol{G}_{s,i}^{-1} \right)^{-1} \left( 2w_i(\boldsymbol{a}_i - \boldsymbol{y}) - \sum_{s=0}^{N_{\text{grid}}-1} \boldsymbol{G}_{s,i}^{-1} \sum_{j>s}^{N_{\text{grid}}-1} \boldsymbol{\nu}_{j,i} \right), \quad i = 1, \ldots, N_{\text{data}}, \\
\boldsymbol{u}_{s,i} &= -\frac{1}{2w_i} \boldsymbol{G}_{s,i}^{-1} \left( \boldsymbol{\mu}_{N_{\text{data}}-1,i} + \sum_{j>s}^{N_{\text{grid}}-1} \boldsymbol{\nu}_{j,i} \right), \quad s = 0, \ldots, N_{\text{grid}} - 1, \, i = 1, \ldots, N_{\text{data}} \\
\boldsymbol{z}_{s+1,i} &= \boldsymbol{z}_{s,i} + \boldsymbol{u}_{s,i}, \quad s = 0, \ldots, N_{\text{grid}} - 1, \, i = 1, \ldots, N_{\text{data}}, \\
\boldsymbol{z}_{0,i} &= \boldsymbol{a}_i \quad i = 1, \ldots, N_{\text{data}},
\end{aligned}
\tag{37}
$$

*where*

$$
\begin{aligned}
W &= \sum_{i=1}^{N_{\text{data}}} w_i \left( \sum_{s=0}^{N_{\text{grid}}-1} \boldsymbol{G}_{i,s}^{-1} \right)^{-1}, \\
V &= \sum_{i=1}^{N_{\text{data}}} w_i \left( \sum_{s=0}^{N_{\text{grid}}-1} \boldsymbol{G}_{s,i}^{-1} \right)^{-1} \boldsymbol{a}_i - \frac{1}{2} \sum_{i=1}^{N_{\text{data}}} \left( \sum_{s=0}^{N_{\text{grid}}-1} \boldsymbol{G}_{s,i}^{-1} \right)^{-1} \sum_{s=0}^{N_{\text{grid}}-1} \boldsymbol{G}_{s,i}^{-1} \sum_{j>s}^{N_{\text{grid}}-1} \boldsymbol{\nu}_{j,i}.
\end{aligned}
\tag{38}
$$

If $\boldsymbol{G} = I$ as in diffusion models, we get the following simplified update scheme.

**Corollary A.9.** *The update scheme of* $\boldsymbol{u}_{s,i}, \boldsymbol{\mu}_{s,i}$ *and* $\boldsymbol{z}_{s,i}$ *to minimize Eq. 10 is*

$$
\begin{aligned}
\boldsymbol{y} &= \frac{1}{\sum_{i=1}^{N_{\text{data}}} w_i} \left( \sum_{i=1}^{N_{\text{data}}} w_i \boldsymbol{a}_i - \frac{1}{2} \sum_{i=1}^{N_{\text{data}}} \sum_{s=0}^{N_{\text{grid}}-1} \sum_{j>s} \boldsymbol{\nu}_{j,i} \right), \\
\boldsymbol{u}_{s,i} &= \frac{\boldsymbol{y} - \boldsymbol{a}_i}{N_{\text{grid}}} + \frac{1}{2w_i} \left( \frac{1}{N_{\text{grid}}} \sum_{k=0}^{N_{\text{grid}}-1} \sum_{j>k} \boldsymbol{\nu}_{j,i} - \sum_{j>s} \boldsymbol{\nu}_{j,i} \right) \quad s = 0, \ldots, N_{\text{grid}} - 1, \, i = 1, \ldots, N_{\text{data}} \\
\boldsymbol{z}_{s+1,i} &= \boldsymbol{z}_{s,i} + \boldsymbol{u}_{s,i}, \quad s = 0, \ldots, N_{\text{grid}} - 1, \, i = 1, \ldots, N_{\text{data}}, \\
\boldsymbol{z}_{0,i} &= \boldsymbol{a}_i \quad i = 1, \ldots, N_{\text{data}},
\end{aligned}
\tag{39}
$$

Note that by the same argument as in Appendix A.4, the computation of the mean and minimizing curves will also have global convergence and local quadratic convergence. Similarly, to the adaptive update for *ProbGEORCE* in Appendix A.5, we can update the solution for the mean computation by the adaptive

scheme.

$$
\boldsymbol{z}_{s,i}^{(k+1)} \leftarrow \boldsymbol{z}_{s,i}^{(k)} + \kappa \left( \boldsymbol{z}_{s,i}^{(k+1)} - \boldsymbol{z}_{s,i}^{(k)} \right),
$$

$$
\boldsymbol{u}_{s,i}^{(k+1)} \leftarrow \boldsymbol{u}_{s,i}^{(k)} + \kappa \left( \boldsymbol{u}_{s,i}^{(k+1)} - \boldsymbol{u}_{s,i}^{(k)} \right),
$$

$$
\tilde{\boldsymbol{G}}_{s,i}^{(k+1)} \leftarrow (1 - \beta_1) \tilde{\boldsymbol{G}}_{s,i} \left( \boldsymbol{z}_{s,i}^{(k+1)} \right) + \beta_1 \tilde{\boldsymbol{G}}_{s,i}^{(k+1)},
$$

$$
\tilde{\boldsymbol{\nu}}_{s}^{(k+1)} \leftarrow (1 - \beta_1) \tilde{\boldsymbol{\nu}}_{s,i} \left( \boldsymbol{z}_{s,i}^{(k+1)} \right) + \beta_1 \tilde{\boldsymbol{\nu}}_{s,i}^{(k+1)},
$$

$$
\tilde{g}^{(k+1)} \leftarrow (1 - \beta_2) \left( \sum_{i=1}^{N_{\mathrm{data}}} \sum_{s=0}^{N_{\mathrm{grid}}-1} \left\| \tilde{\boldsymbol{\nu}}_{s,i}^{(k+1)} \right\|^2 \right) + \beta_2 \tilde{g}^{(k+1)},
$$

$$
\hat{\boldsymbol{G}}_{s,i}^{(k+1)} = \frac{\tilde{\boldsymbol{G}}_{s,i}^{(k+1)}}{1 - \beta_1^{k+1}},
$$

$$
\hat{\boldsymbol{\nu}}_{s,i}^{(k+1)} = \frac{\tilde{\boldsymbol{\nu}}_{s}^{(k+1)}}{1 - \beta_1^{k+1}},
$$

$$
\hat{g}^{(k+1)} = \frac{\tilde{g}^{(k+1)}}{1 - \beta_2^{k+1}},
$$

$$
\kappa = \min \left\{ \frac{\gamma}{\sqrt{1 + \tilde{g}^{(k+1)}} + \epsilon}, 1 \right\},
$$

We show the update in pseudo-code in Algorithm 2, where the line-search can be replaced by the update scheme above similar to Algorithm 1.

---

**Algorithm 2** ProbGEORCE for means

---

1: **Input**: tol, $\boldsymbol{a}_{1:N_{\text{data}}}$, $N_{\text{grid}}$

2: **Output**: Geodesic estimate $\boldsymbol{z}_{0:N_{\text{grid}}}$

3: Set $\boldsymbol{y}^{(0)} \leftarrow \boldsymbol{a}_0$, $\boldsymbol{z}_{s,i}^{(0)} \leftarrow \boldsymbol{a}_i + \frac{\boldsymbol{y}^{(0)} - \boldsymbol{a}_i}{N_{\text{grid}}} s$ for $s = 0., \ldots, N_{\text{grid}}$ and $\boldsymbol{u}_{s,i}^{(0)} \leftarrow \frac{\boldsymbol{y}^{(0)} - \boldsymbol{a}_i}{N_{\text{grid}}}$ for $s = 0., \ldots, N_{\text{grid}} - 1$ and $i = 1, \ldots, N_{\text{data}}$.

4: **while** $\frac{1}{N_{\text{data}}} \left\| \nabla_{\boldsymbol{y}} E(\boldsymbol{y}) \big|_{\boldsymbol{y} = \boldsymbol{z}_{s,i}^{(k)}} \right\|_2 > \text{tol}$ **do**

5:      $\boldsymbol{G}_{s,i} \leftarrow \boldsymbol{G}\left(\boldsymbol{z}_{s,i}^{(k)}\right)$ for $s = 0, \ldots, N_{\text{grid}} - 1$ and $i = 1, \ldots, N_{\text{data}}$.

6:      $\boldsymbol{\nu}_{s,i} \leftarrow \nabla_{\boldsymbol{z}}\left(\boldsymbol{u}_{s,i}^{(k)}\boldsymbol{G}(\boldsymbol{z})\boldsymbol{u}_{s,i}^{(k)} + \lambda S\left(\boldsymbol{z}_{s,i}^{(k)}\right)\right)\Big|_{\boldsymbol{z} = \boldsymbol{z}_{s,i}^{(k)}}$ for $s = 1, \ldots, N_{\text{grid}} - 1$ and $i = 1, \ldots, N_{\text{data}}$.

7:      $\boldsymbol{y} \leftarrow W^{-1}V$ with $W, V$ given by eq. 38.

8:      $\boldsymbol{\mu}_{N_{\text{grid}}-1,i} \leftarrow \left(\sum_{s=0}^{N_{\text{grid}}-1}\boldsymbol{G}_{s,i}^{-1}\right)^{-1}\left(2w_i(\boldsymbol{a}_i - \boldsymbol{y}) - \sum_{s=0}^{N_{\text{grid}}-1}\boldsymbol{G}_{s,i}^{-1}\sum_{j>s}^{N_{\text{grid}}-1}\boldsymbol{\nu}_{j,i}\right)$ for $i = 1, \ldots, N_{\text{data}}$ and $s = 1, \ldots, N_{\text{grid}} - 1$.

9:      $\boldsymbol{u}_{s,i} \leftarrow -\frac{1}{2w_i}\boldsymbol{G}_{s,i}^{-1}\left(\boldsymbol{\mu}_{N_{\text{grid}}-1,i} + \sum_{j>s}^{N_{\text{grid}}-1}\boldsymbol{\nu}_{j,i}\right)$ for $t = 0, \ldots, N_{\text{grid}} - 1$ and $i = 1, \ldots, N_{\text{data}}$.

10:      $\boldsymbol{z}_{s+1,i} \leftarrow \boldsymbol{z}_{s,i} + \boldsymbol{u}_{s,i}$ for $s = 0, \ldots, N_{\text{grid}} - 2$ and $i = 1, \ldots, N_{\text{data}}$.

11:      Using line search find $\alpha^*$ for the following optimization problem for the discrete sum of energy $E$

$$\min_{\alpha} \quad E\left(\boldsymbol{z}_{0:N_{\text{grid}}, 1:N_{\text{data}}}\right) \quad \text{(exact line search)}$$

$$\text{s.t.} \quad \boldsymbol{z}_{s+1,i} = \boldsymbol{z}_{s,i} + \alpha\tilde{\boldsymbol{u}}_{s,i} + (1-\alpha)\boldsymbol{u}_{s,i}^{(k)}, \quad s = 0, \ldots, N_{\text{grid}} - 1, \ i = 1, \ldots, N_{\text{data}}.$$

$$\tilde{\boldsymbol{u}}_{s,i} = \alpha\boldsymbol{u}_{s,i} + (1-\alpha)\boldsymbol{u}_{s,i}^{(k)}, \quad s = 0, \ldots, N_{\text{grid}} - 1, \ i = 1, \ldots, N_{\text{data}}.$$

$$x_{0,i} = \boldsymbol{a}_i.$$

12:      Set $\boldsymbol{u}_{s,i}^{(k+1)} \leftarrow \alpha^*\boldsymbol{u}_{s,i} + (1-\alpha^*)\boldsymbol{u}_{s,i}^{(k)}$ for $s = 0, \ldots, N_{\text{grid}} - 1$ and $i = 1, \ldots, N_{\text{data}}$.

13:      Set $\boldsymbol{z}_{s+1,i}^{(k+1)} \leftarrow \boldsymbol{z}_{s,i}^{(k+1)} + \boldsymbol{u}_{s,i}^{(k+1)}$ for $s = 0, \ldots, N_{\text{grid}} - 1$ and $i = 1, \ldots, N_{\text{data}}$.

14: **end while**

15: return $\boldsymbol{z}_{s,i}$ for $s = 0, \ldots, N_{\text{grid}} - 1$ for $i = 1, \ldots, N_{\text{data}}$.

---

## B  Algorithms

In Algorithm 3, we state *ProbGEORCE* in pseudo-code for solving the boundary value problem for curves on the Newtonian system on a Riemannian manifold.

---
**Algorithm 3** ProbGEORCE: Probabilistic Geodesics
---
1: **Input**: tol, $N_{\text{grid}}$, $\rho$
2: **Output**: Constrained geodesic estimate $\boldsymbol{z}_{0:N_{\text{grid}}}$
3: Set $\boldsymbol{z}_s^{(0)} \leftarrow \boldsymbol{a} + \frac{\boldsymbol{b}-\boldsymbol{a}}{N_{\text{grid}}}s$ for $s = 0., \ldots, N_{\text{grid}}$, $\boldsymbol{u}_s^{(0)} \leftarrow \frac{\boldsymbol{b}-\boldsymbol{a}}{N_{\text{grid}}}$ for $s = 0., \ldots, N_{\text{grid}} - 1$ and $k \leftarrow 0$
4: **while** $\left\| \nabla_{\boldsymbol{y}} E(\boldsymbol{y}) \big|_{\boldsymbol{y}=\boldsymbol{z}_s}^{(k)} \right\|_2 > \text{tol}$ **do**
5:    $\boldsymbol{G}_s \leftarrow \boldsymbol{G}\left(\boldsymbol{z}_s^{(k)}\right)$ for $s = 0, \ldots, N_{\text{grid}} - 1$
6:    $\boldsymbol{\nu}_s \leftarrow \nabla_{\boldsymbol{y}}\left(\boldsymbol{u}_s^{(k)}\boldsymbol{G}\left(\boldsymbol{y}\right)\boldsymbol{u}_s^{(k)} + S(\boldsymbol{y})\right)\Big|_{\boldsymbol{y}=\boldsymbol{z}_s^{(k)}}$ for $s = 1, \ldots, N_{\text{grid}} - 1$
7:    $\boldsymbol{\mu}_{N_{\text{grid}}-1} \leftarrow \left(\sum_{s=0}^{N_{\text{grid}}-1}\boldsymbol{G}_s^{-1}\right)^{-1}\left(2(\boldsymbol{a}-\boldsymbol{b}) - \sum_{s=0}^{N_{\text{grid}}-1}\boldsymbol{G}_s^{-1}\sum_{j>s}^{N_{\text{grid}}-1}\boldsymbol{\nu}_j\right)$
8:    $\boldsymbol{u}_s \leftarrow -\frac{1}{2}\boldsymbol{G}_s^{-1}\left(\boldsymbol{\mu}_{N_{\text{grid}}-1} + \sum_{j>s}^{N_{\text{grid}}-1}\boldsymbol{\nu}_j\right)$ for $s = 0, \ldots, N_{\text{grid}} - 1$
9:    $\boldsymbol{z}_{s+1} \leftarrow \boldsymbol{z}_s + \boldsymbol{u}_s$ for $s = 0, \ldots, N_{\text{grid}} - 1$
10:    $j \leftarrow 0$
11:    **while** $E\left(\boldsymbol{z}_{0:N_{\text{grid}}}\right) < E\left(\tilde{\boldsymbol{z}}_{0:N_{\text{grid}}}\right)$ **do**
12:      $\tilde{\boldsymbol{z}}_{s+1} = \tilde{\boldsymbol{z}}_s + \rho^j\boldsymbol{u}_s + (1-\rho^j)\boldsymbol{u}_s^{(k)}, \quad s = 0, \ldots, N_{\text{grid}} - 1, \quad \tilde{\boldsymbol{z}}_0 = \boldsymbol{a}.$
13:      $j \leftarrow j + 1$
14:    **end while**
15:    Set $\boldsymbol{u}_s^{(k+1)} \leftarrow \rho^{j-1}\boldsymbol{u}_s + (1-\rho^{j-1})\boldsymbol{u}_s^{(k)}$ for $s = 0, \ldots, N_{\text{grid}} - 1$
16:    Set $\boldsymbol{z}_{s+1}^{(k+1)} \leftarrow \boldsymbol{z}_s^{(k+1)} + \boldsymbol{u}_s^{(k+1)}$ for $s = 0, \ldots, N_{\text{grid}} - 1$
17:    $k \leftarrow k + 1$
18: **end while**
19: return $\boldsymbol{z}_s$ for $s = 0, \ldots, N_{\text{grid}}$.
---

## C  Existence and uniqueness of the Fréchet mean

Let $(\mathcal{M}, g)$ be a complete Riemannian manifold. Consider a probability space $(\Omega, \mathbb{B}, \mathbb{P})$, where $\mathbb{B}$ denotes the Borel $\sigma$-algebra, then the Fréchet mean is defined as (Fréchet, 1948; Pennec, 2006)

$$\boldsymbol{\mu} = \underset{\boldsymbol{y} \in \mathcal{M}}{\arg\min} \int_{\mathcal{M}} \text{dist}(\boldsymbol{z}, \boldsymbol{y})^2 p_{\boldsymbol{x}}(\boldsymbol{z}) \mathrm{d}\mathcal{M}(\boldsymbol{z}), \tag{40}$$

where $\boldsymbol{x}$ is a random variable on $\mathcal{M}$ with density $p_{\boldsymbol{x}}$ and $\mathrm{d}\mathcal{M}(\boldsymbol{z})$ is the Riemannian volume measure. The discrete version of Eq. 40 is (Pennec, 2006)

$$\boldsymbol{\mu} = \underset{\boldsymbol{y} \in \mathcal{M}}{\arg\min} \sum_{i=1}^{N_{\text{data}}} \text{dist}(\boldsymbol{z}_i, \boldsymbol{y})^2. \tag{41}$$

For an in-depth analysis of the properties of Eq. 41 as an estimator, and the general properties of the Fréchet mean, we refer to (Ziezold, 1977; Bhattacharya & Patrangenaru, 2003). In general, the existence and uniqueness of the Fréchet mean is not guaranteed. Let $\mathcal{B}_r(p) = \{\boldsymbol{y} \in \mathcal{M} \,|\, \text{dist}(\boldsymbol{x}, p) < r, \,\forall \boldsymbol{x} \in \mathcal{M}\}$ be an open ball on $\mathcal{M}$. If there exists a unique minimizing geodesic from the center, $p \in \mathcal{M}$, to any other point in the open ball, then the open ball is said to be regular (Pennec, 2006). If $\mathcal{M}$ is a complete Riemannian manifold with sectional curvature bounded by $\kappa$, and the support of the data distribution is within an open regular ball $\mathcal{B}_r(p)$ for a point $p \in \mathcal{M}$ with radius

$$r < \pi/(2\sqrt{\kappa}),$$

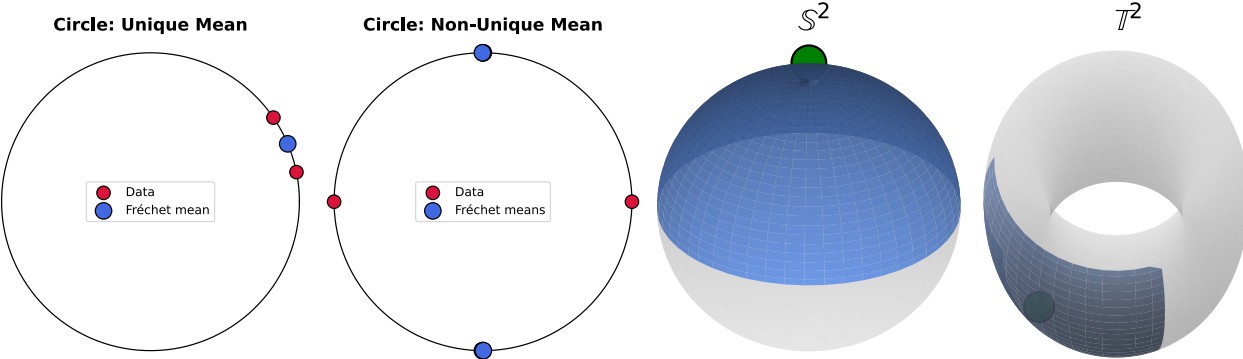

Figure 9: From left to right: The Fréchet mean (blue) for two data points (red) on the circle with uniqueness of the Fréchet mean, two points on the circle, where there are multiple Fréchet mean, the area (blue) for the data support around the north pole (green) of $\mathbb{S}^2$ for which the existence and uniqueness is guaranteed and similarly for the torus embedded in $\mathbb{R}^3$.

then the Fréchet mean exists and is unique (Kendall, 1990). Note that there also exist other results of existence and uniqueness that require a certain regularity of the variance (Karcher, 1977). To illustrate the existence and uniqueness, we consider the circle, unit-sphere and torus embedded in $\mathbb{R}^3$ in Fig. 9. The two left-most figures illustrate that if the data points are sufficiently 'close', then the Fréchet mean is unique, while if, for example, there are antipodal data points on the circle, then there will be multiple Fréchet means. The two figures on the right illustrate an area in the unit sphere and torus embedded in $\mathbb{R}^3$, where the Fréchet mean is unique if the support of the data distribution is within these areas. For the unit sphere, the sectional curvature is constant and hence $\kappa = 1$, and therefore there exists a unique Fréchet mean if the data support is within an open ball with $r < \pi/2$. Thus, if the data distribution is within a hemisphere, then the Fréchet mean is unique, as illustrated in Fig. 9. For the torus embedded in $\mathbb{R}^3$ parametrized by

$$\left( (R + r\cos\theta)\cos\phi, (R + r\cos\theta)\sin\phi, r\sin\theta \right), \quad \theta, \phi \in [0, 2\pi),$$

with $R > r > 0$, the sectional curvature is given by

$$\frac{\cos\theta}{r\,(R + r\cos\theta)}.$$

Thus, the curvature and the corresponding area with the uniqueness of the Fréchet mean guaranteed (Kendall, 1990) depends on $\theta$.

In general, the conditions for uniqueness and existence of the Fréchet mean may not hold for the generalized version of the Fréchet mean in Eq. 10. For the discretized version in Eq. 25, define

$$f_{\text{Frechet}} = \sum_{i=1}^{N_{\text{data}}} w_i \sum_{s=0}^{N_{\text{grid}}-1} \left( \boldsymbol{x}_{t+1,i} - \boldsymbol{x}_{t,i} \right)^\top \boldsymbol{G}(\boldsymbol{z}_{s,i}) \left( \boldsymbol{x}_{t+1,i} - \boldsymbol{x}_{t,i} \right),$$

$$\lambda f_S = \lambda \sum_{i=1}^{N_{\text{data}}} w_i \sum_{s=0}^{N_{\text{grid}}-1} \lambda S(\boldsymbol{z}_{s,i}),$$

Assume that the number of grid points $N_{\text{grid}}$ is sufficiently high to approximate the continuous integrals. If all data points are within a certain set $A \subseteq \mathcal{M}$, such that the Fréchet mean in Eq. 41 is unique and exists, then it follows that if

$$\lambda \rho_{\min} \left( \text{Hess}_{\boldsymbol{x}} \left( f_S \right) \right) > -\rho_{\min} \left( \text{Hess}_{\boldsymbol{x}} f_{\text{Frechet}} \right), \quad \forall \boldsymbol{x} \in A$$

where $\rho_{\min}(\cdot)$ denotes the smallest eigenvalues of the Riemannian Hessian (Afsari, 2011). We illustrate the negative loss landscape of the Fréchet mean in Fig. 10 for the energy-based model for checkerboard data used in Fig. 2. In this case $\boldsymbol{G} = \boldsymbol{I}$, and therefore for $\lambda = 0$ the Fréchet mean is unique and exists for any discrete data distribution. It can be seen that as $\lambda$ increases, two Fréchet means appear, and when $\lambda = 100$ it seems that there are multiple Fréchet means.

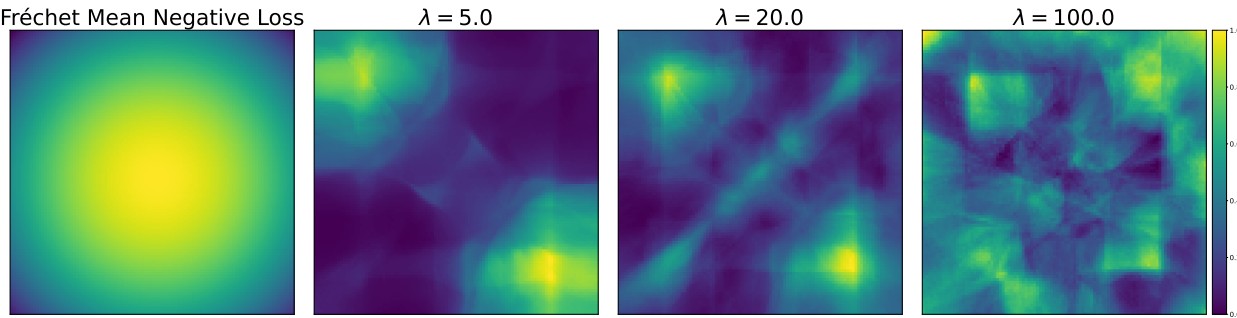

Figure 10: The negative loss of mean, where we for each grid point computes the energy from all other data points using Eq. 7 for different values of $\lambda$. The right-most figure correspond to $\lambda = 0$. Note that each loss is normalized for each plot to be between 0 and 1, and therefore the absolute scale between the plots can not be compared.

## D  Application to diffusion models and image editing

Score-based diffusion models (Song et al., 2021b) approximate the data distribution by transforming the data using forward dynamics:

$$\mathrm{d}\boldsymbol{x}_t = \mu(\boldsymbol{x}_t, t)\mathrm{d}t + \sigma(t)\mathrm{d}W_t, \quad \boldsymbol{x}_0 \sim p_{\mathrm{data}}, \tag{42}$$

such that $\mu : \mathbb{R}^D \times \mathbb{R}_+ \to \mathbb{R}^D$ and $\sigma : \mathbb{R}_+ \to \mathbb{R}^D$ are suitable functions for eq. 42 to converge to a known limiting distribution $\pi$ for a sufficiently large time $T > 0$ (Song et al., 2021b). Samples of the data distribution can then be generated using the time-reversal process $\boldsymbol{y}_t := \boldsymbol{x}_{T-t}$ (Anderson, 1982):

$$dy_t = \left(\mu(\boldsymbol{y}_t, t) - \sigma(t)^2 \nabla_{\boldsymbol{y}} \log p_t(\boldsymbol{y}_t)\right) \mathrm{d}t + \sigma(t)\mathrm{d}\overline{W}_t, \tag{43}$$

where $\boldsymbol{y}_0 \sim \pi$, and $\nabla_{\boldsymbol{y}} \log p_t(\cdot)$ denotes the *score*, which can be learned using score matching (Hyvärinen, 2005; Vincent, 2011; Song et al., 2020). Samples can also be generated deterministically using the probability flow ODE (Chen et al., 2018)

$$\mathrm{d}\boldsymbol{y}_t = \left(\mu(\boldsymbol{y}_t, t) - \frac{1}{2}\sigma(t)^2 \nabla_{\boldsymbol{y}_t} \log p_t(\boldsymbol{y}_t)\right) \mathrm{d}t, \tag{44}$$

Score-based diffusion models can be seen as the continuous version of denoising diffusion probabilistic models DDPM (Ho et al., 2020; Song et al., 2021b) and have also been extended to data distributions on Riemannian manifolds (Huang et al., 2022; Bortoli et al., 2022; Jo & Hwang, 2024). We illustrate in Fig. 11 the difference between computing curves in noise space and data space as described in Section 3.

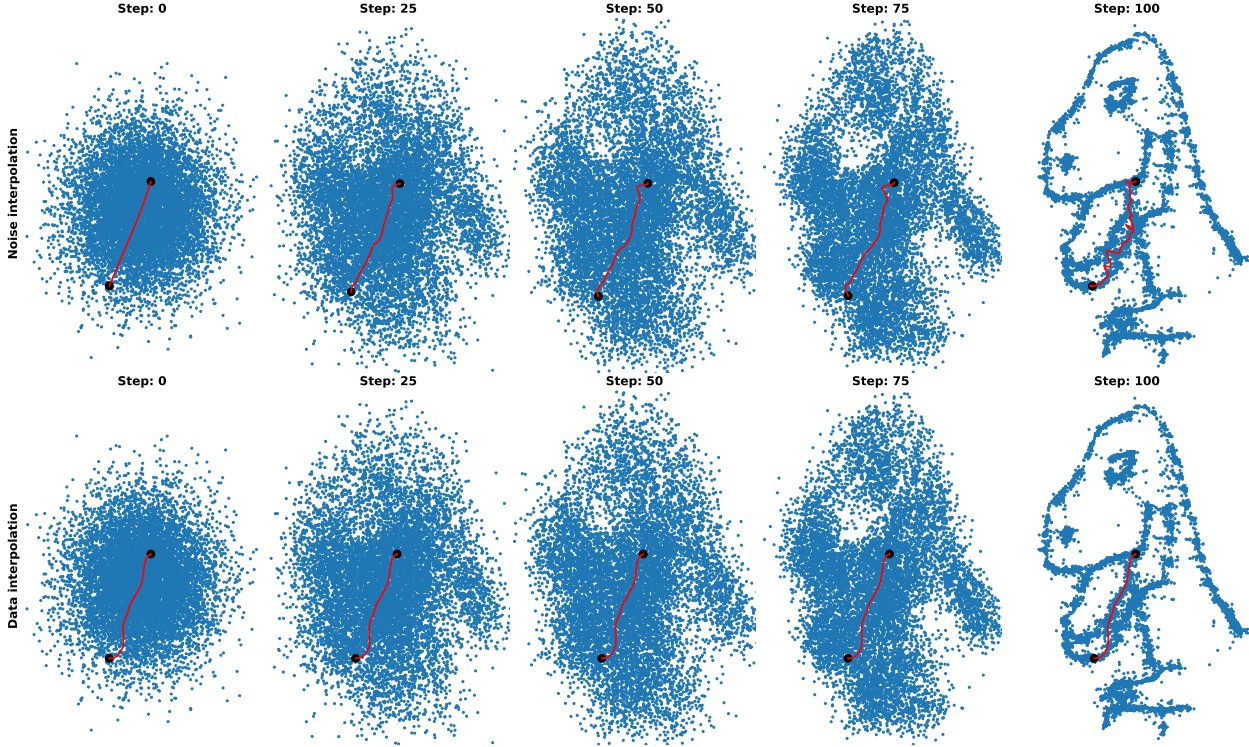

Figure 11: The first row shows interpolation in noise space (red) using the isotropic Gaussian limiting distribution and transports the curve to data space using a DDPM-model for a dino-dataset. The second row computes the curve directly in data space using the score of the DDPM, which, for illustrative purposes, is repeated in each step of the sampling process.

**Connection to image editing** Our method regularizes the geodesic interpolation and for $\lambda = 0$ our method will compute the connecting geodesic. In the Euclidean case, geodesic interpolation corresponds to linear interpolation, whereas spherical interpolation corresponds to a geodesic on a sphere. NoiseDiffusion (Zheng et al., 2024) is designed for interpolation in diffusion models with image editing. Let $f$ denote the forward diffusion process that takes data to noise in eq. 42 and $f^{-1}$ denote the reverse process that generates data samples using noise in eq. 44, then NoiseDiffusion (Zheng et al., 2024) proposes the following interpolation between two images $\boldsymbol{x}, \boldsymbol{y}$.

$$
\begin{aligned}
a &= \mathrm{clip}\left(f\left(\boldsymbol{x}, t\right)\right), \\
b &= \mathrm{clip}\left(f\left(\boldsymbol{y}, t\right)\right), \\
\boldsymbol{z}_s &= \alpha(s)a + \beta(s)b + (\mu(s) - \alpha(s))\,\boldsymbol{x} + (\nu(s) - \beta(s))\,\boldsymbol{y} + \gamma(s)\epsilon, \\
\boldsymbol{x}_s &= f^{-1}\left(\mathrm{clip}\left(\boldsymbol{z}_s, t\right)\right),
\end{aligned}
\tag{45}
$$

where clip denotes an element-wise operation that restricts a value to be between $[-\mathrm{boundary}, \mathrm{boundary}]$, while $\alpha(s), \beta(s)$ and $\gamma(s)$ are functions depending on $s$ with $\alpha^2(s) + \beta^2(s) + \gamma(s)^2 = 1$ and $\epsilon \sim \mathcal{N}_{\mathrm{grid}}\left(0, I\right)$. The functions $\mu(s)$ and $\nu(s)$ serve as compensation for lost information. Note that

$$
\boldsymbol{z}_s = \alpha(s)a + \beta(s)b - \alpha(s)\boldsymbol{x} - \beta(s)\boldsymbol{y} + \mu(s)\boldsymbol{x} + \nu(s)\boldsymbol{y} + \gamma(s)\epsilon.
$$

Inspired by eq. 45 we can easily extend our method to compute similar interpolants in noise space by

$$
\boldsymbol{z}_s = \mathrm{ProbGEORCE}_1(a, b) - \mathrm{ProbGEORCE}_2(\boldsymbol{x}, \boldsymbol{y}) + \mu(s)\boldsymbol{x} + \nu(s)\boldsymbol{y} + \gamma(s)\epsilon,
\tag{46}
$$

where $\{\mathrm{ProbGEORCE}\}_{s=1}^2$ denotes the interpolations using Algorithm 3 using any metric or regularization function. Thus, we can obtain near-identical results for NoiseDiffusion with any original images to modify the interpolation curve as illustrated in Fig. 12. Note that NoiseDiffusion assumes that the limiting distribution of the diffusion model is isotropic Gaussian, which our modification in eq. 46 does not assume. Note also that NoiseDiffusion (Zheng et al., 2024) only defines interpolation and cannot be used to compute mean values or other statistics for diffusion models.

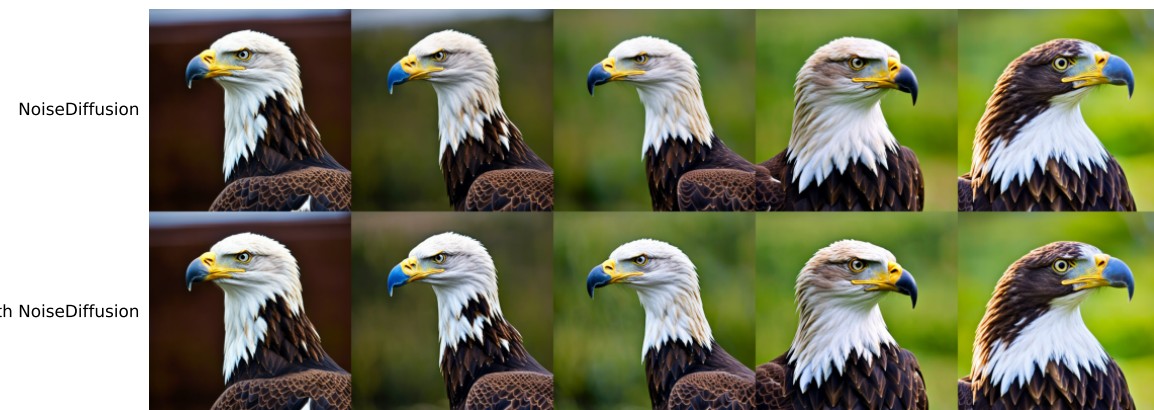

Figure 12: Computed interpolations for ControlNet (Zhang et al., 2023) for eagle images of size $768 \times 768 \times 3$ similar to the experiment by Zheng et al. (2024). We set the regularizer function in Eq. 7 as the density of the $\chi^2$-distribution on the squared norm of the grid points $\boldsymbol{z}_{0:N}$. We plot the using our method incorporated into *NoiseDiffusion* using Eq. 46 and compare to NoiseDiffusion (Zheng et al., 2024).

# E   Experimental details

The following contains experimental details and hyper-parameters. The code for reproducing the results can be found at [LINK REMOVED DURING REVIEW].

**Normalizing $\lambda$**   Since the energy and regularization function $S$ in Eq. 7 might have very different scales, we will re-normalize $\lambda$ when computing interpolation and the mean value. We do it in the following sense: Let $E^{(0)}$ denote the energy $\int_0^1 \dot{\gamma}(t)G(\gamma(t))\dot{\gamma}(t)\,dt$, where $\gamma$ is a simple straight line connecting any given start and end point. Let $S^{(0)}$ denote the corresponding value of $\int_0^1 S(\gamma(t))\,dt$. We propose to use a normalized version, $\tilde{\lambda}$, for all computations

$$\tilde{\lambda} := \lambda \frac{E^{(0)}}{S^{(0)}}.$$

This is to ensure that the energy and regularization terms are approximately on the same scale before performing any computations. We illustrate in Fig. 13 the effect of varying values of $\lambda$ with the proposed normalization, where we see that higher values of $\lambda$ increases the distance between grid points in low-density areas.

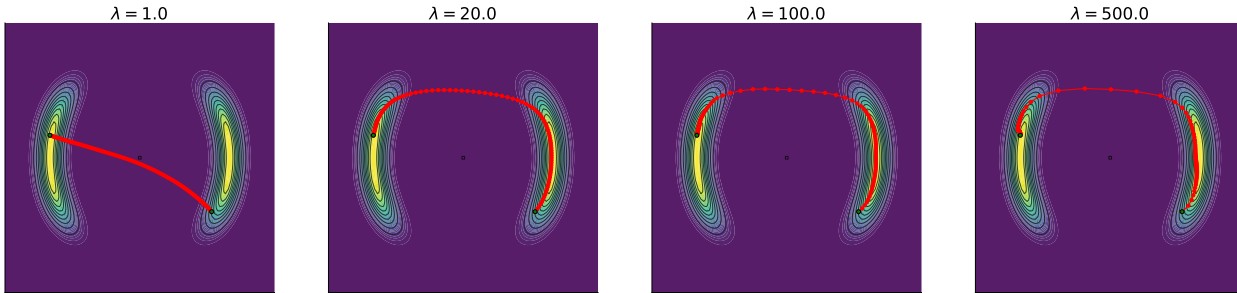

Figure 13: The distance between grid points for the normalizing flow model in Fig. 2 for varying values of $\lambda$.

**Benchmarks**   In the following table, we describe the metrics and hyper-parameters used in Table 9. We consider the density $p$ as described in Section 4. For the methods that apply $p$ directly, we compute it as $\exp \log p$, although this may give a non-normalized density.

Table 5: Benchmark methods and hyper-parameters

| Method | Metric | Hyper-parameters |
|---|---|---|
| Linear | $G(\boldsymbol{z}) = I$ | - |
| Spherical | Riemannian geometry of a sphere embedded in Euclidean space | - |
| Fisher-Rao | $G(\boldsymbol{z}) = \nabla_{\boldsymbol{z}} \log p(\boldsymbol{z}) \nabla_{\boldsymbol{z}} \log p(\boldsymbol{z})^{\top}$ | - |
| Jacobian | $G(\boldsymbol{z}) = J_{\nabla_{\boldsymbol{z}} \log p}^{\top}(\boldsymbol{z}) J_{\nabla_x \log p}(\boldsymbol{z})$ | - |
| Inverse Density | $G(\boldsymbol{z}) = \frac{1}{p(\boldsymbol{z})^2} I$ | - |
| Generative | $G(\boldsymbol{z}) = \left(\frac{p(\boldsymbol{z})+\lambda}{p_0+\lambda}\right)^2 I$ | $p_0 = \lambda = 1$ |
| Monge | $G(\boldsymbol{z}) = I + \alpha^2 \nabla_{\boldsymbol{z}} \log p(\boldsymbol{z}) \nabla_{\boldsymbol{z}} \log p(\boldsymbol{z})^{\top}$ | $\alpha = 1$ |

When we write "Reg", we consider the metric described in Table 5 added with the identity matrix $\alpha I$, i.e., $\tilde{G}(x) = G(x) + \alpha I$ with $\alpha = 1$. Note that we did not add this to the Fisher-Rao metric as this would correspond to the Monge metric for $\alpha = 1$. For all methods, except linear and spherical, we apply *ProbGEORCE* with adaptive update of the step-size in Algorithm 1. For Linear, we apply the closed-form expression for Euclidean geometry of the initial and boundary value problem as well as the closed-form expression of the mean. For spherical, we apply the closed-form expression for the initial value and boundary value problem. Note that IVP are started from the same point as the first point for the BVP. For the mean,

we use the Logarithmic map and compute the mean using gradient descent (Pennec, 2006). For the energy-based model on AFHQ, we give a brief description of the benchmarks in (Béthune et al., 2025) in Table 6. Note that the constants $\alpha$ and $\beta$ are scaled to target high-density areas. We refer to (Béthune et al., 2025) for the approach. Note also that $1/p_\theta$ refers to using the inverse density, while $\log p_\theta$ refers to using the log-likelihood learned the energy based model. The $\boldsymbol{G}_{\mathrm{LAND}}$ is described in (Béthune et al., 2025).

Table 6: Kinetic energy for different metric types.

| Metric type | Kinetic Energy |
|---|---|
| Conformal | $E_{\mathrm{kin}}(x, \dot{x}) = (\lambda + h(x)) \|\dot{x}\|^2$, where $h$ is a scalar function. |
| Diagonal | $E_{\mathrm{kin}}(x, \dot{x}) = (\lambda + h(x)) \|\dot{x}\|^2$, where $h$ is not a scalar function. |
| Full | $E_{\mathrm{kin}}(x, \dot{x}) = \lambda \|\dot{x}\|^2 + (\langle \dot{x}, \nabla h(x) \rangle)^2$ |

**Optimizers**  For the results in Table 5, Fig. 11 and Fig. 4, we apply *ProbGEORCE* with adaptive update of the step-size in Algorithm 1 with $\beta_1 = \beta_2 = 0.5$, $\epsilon = 10^{-8}$ and $\gamma = 0.01$ with a maximum of 1000 iterations and a tolerance of $10^{-4}$ to our method and all Riemannian methods with $\lambda = 0$. For ControlNet, we use *ProbGEORCE* with adaptive update of the step-size in Algorithm 1 with $\beta_1 = \beta_2 = 0.5$, $\epsilon = 10^{-8}$ and $\gamma = 0.001$ in data space, and in noise space we use *ProbGEORCE* with line-search and $\rho = 0.5$. For the runtime results for other methods, we set the learning rate to 0.01.

**FID and KID**  We compute the FID and KID scores for the AFHQ dataset based on 10 boundary value curves with $N_{\mathrm{grid}} = 10$ grid points for ControlNet. We set the initial prompt in ControlNet to "A photo of a cat". For the energy-based model in (Béthune et al., 2025), we use 6 boundary points.

**Manifolds**  For the manifolds used for the runtime results in Table 10, we apply the same local chart and metric as in Rygaard & Hauberg (2025), where we add a three isotropic Gaussian with random means in the local charts. For the conceptual figure in Fig. 1, we consider a chart on the form

$$\left\{ \left( u, v, \exp\left( -\left( u^2 + v^2 \right) \right) + 0.3 \sin\left( 3u \right) \cos\left( 3v \right) \right) \mid (u, v) \in \mathbb{R}^2 \right\}.$$

where the regularization corresponds to

$$S(u, v) = -\exp\left( -\frac{\|u - v\|^2}{0.15} \right).$$

**Architecture**  Table 7 contains a list of all trained models and networks used in the paper. All other models are pre-trained.

**Hardware**  The interpolation for GMM and KDE as well as plots have been computed on a: *HP* computer with Intel Core i9-11950H 2.6 GHz 8C, 15.6" FHD, 720P CAM, 32 GB (2×16GB) DDR4 3200 So-Dimm, Nvidia Quadro TI2000 4GB Discrete Graphics, 1TB PCle NVMe SSD, 150W PSU, 8cell, W11Home 64 Advanced.

The interpolation for ControlNet, runtime tables, and benchmarks have been computed on a GPU for at most 24 hours with a maximum memory of 20 GB. The *GPU* consists of 4 nodes on a *Tesla V100*.

The interpolation of CIFAR10 and CelebAHQ with SGM (Song et al., 2021b) used in Appendix F is conducted on a single *Nvidia RTX A6000* GPU with 48G memory.

The interpolation of OASIS3 with LDM used in Appendix F is conducted on four *Nvidia RTX A6000* GPUs.

Table 7: Summary of models, their architectures, and training details.

| Model | Architecture | Training |
|---|---|---|
| Energy-Based model | We use a MLP with one hidden layer with 128 neurons and ReLU activations using the package Ghaderi & Contributors (2025). | Trained for $1,000$ epochs with a batch size of 128 for fixed $50,000$ samples. |
| Normalizing flow | We use 32 affine coupling blocks of MLPs with two hidden layers of 64 neurons using the package (Stimper et al., 2023). | Trained for $4,000$ epochs with a batch size of $2^9$. |
| AR | We use a MLP with two hidden layers of 64 neurons and TanH activations using the package. | Trained for $10,000$ epochs with a batch size of 512 for fixed $50,000$ samples. |
| VAE | We use the same architecture for a noisy circle embedded in $\mathbb{R}^3$ as in Shao et al. (2018) with minor modifications. | Trained for $5,000$ epochs with a batch size of 512 for fixed $50,000$ samples. |
| DDPM dinosaur | We use the architecture from github.com/tanelp/tiny-diffusion. | Trained identical to github.com/tanelp/tiny-diffusion. |

## F  Additional experiments

**Energy-based model**  In the following, we show the image transitions for the images used to compute FID and KID in Table 3. We show the result using our method and the benchmarks from (Béthune et al., 2025) in Fig. 14, Fig. 15, Fig. 16, Fig. 18, Fig. 18, Fig. 21 and Fig. 22.

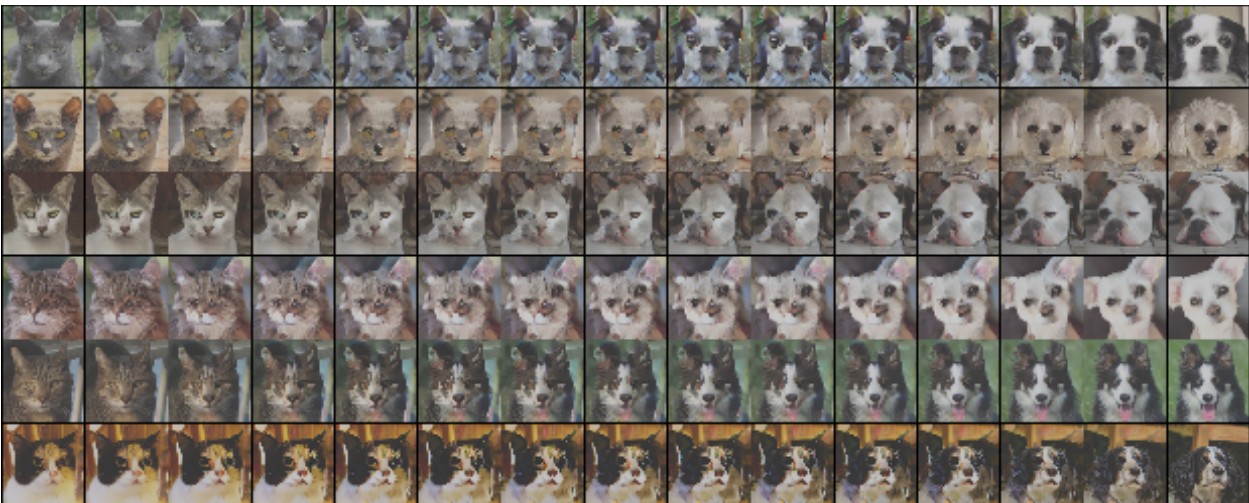

Figure 14: Image transitions on the AFHQ dataset (Choi et al., 2020a) with $\boldsymbol{G}_{1/p_\theta}$ (conformal).

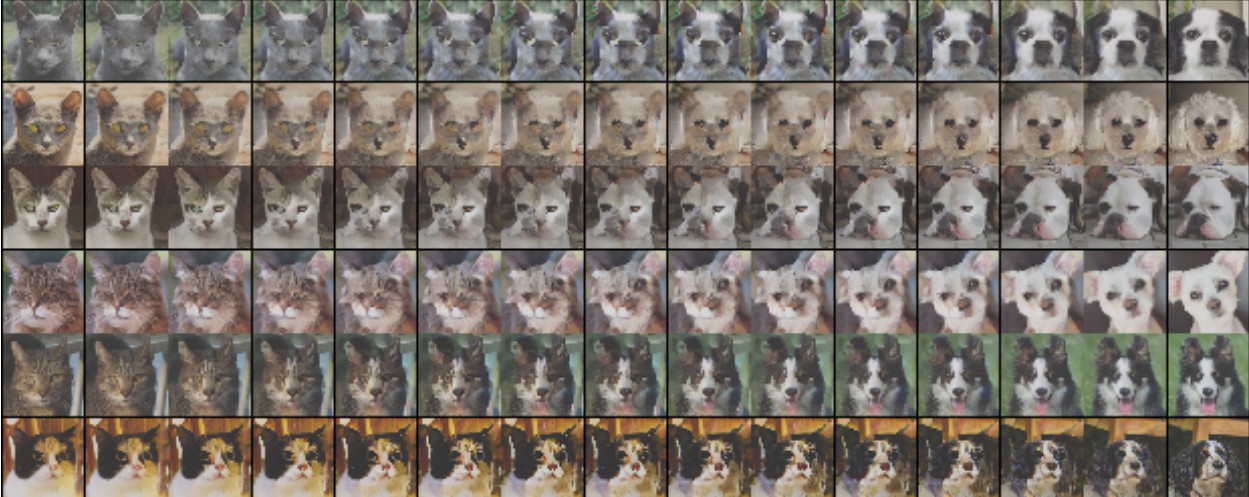

Figure 15: Image transitions on the AFHQ dataset (Choi et al., 2020a) with $\boldsymbol{G}_{\log p_\theta}$ (conformal).

**ControlNet**  We provide additional qualitative examples for the ControlNet diffusion model (Zhang et al., 2023) using the regularization function as described in Section 4. In Fig. 23 and  24, we show our interpolation method in noise space in the direction of a normalized standard normally distributed direction. We interpolate between an initial prompt and a target prompt linearly along the curve. We state the target prompt under the images.

In Fig. 25, 26 and  27, we show our interpolation method and other methods for curves connecting two images.

In Fig. 28,  29 and  30 we show the estimated mean and their transition for different methods.

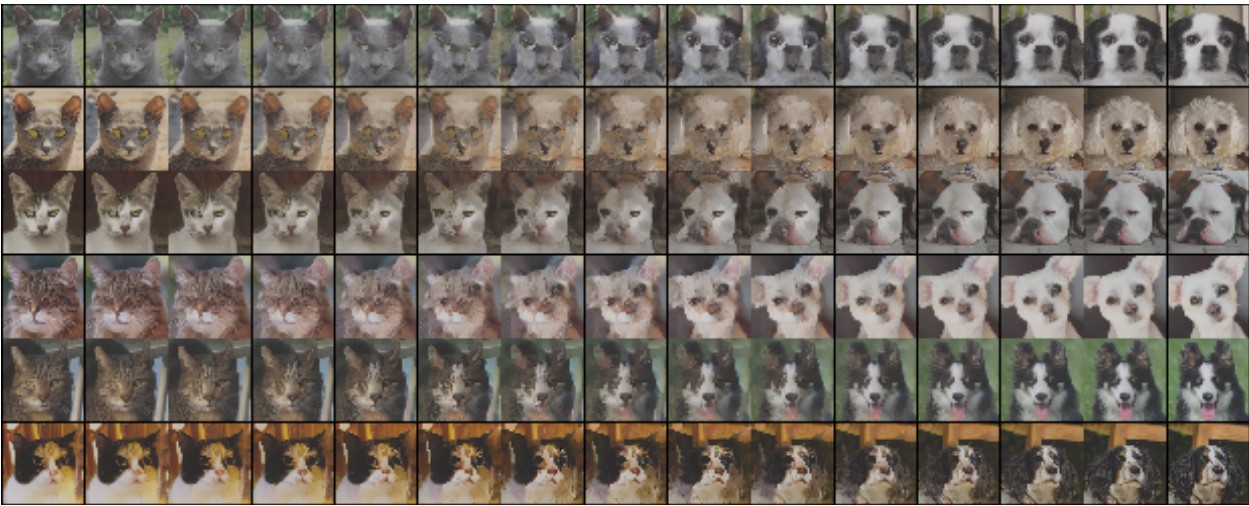

Figure 16: Image transitions on the AFHQ dataset (Choi et al., 2020a) with $\boldsymbol{G}_{1/p_\theta}$ (diagonal).

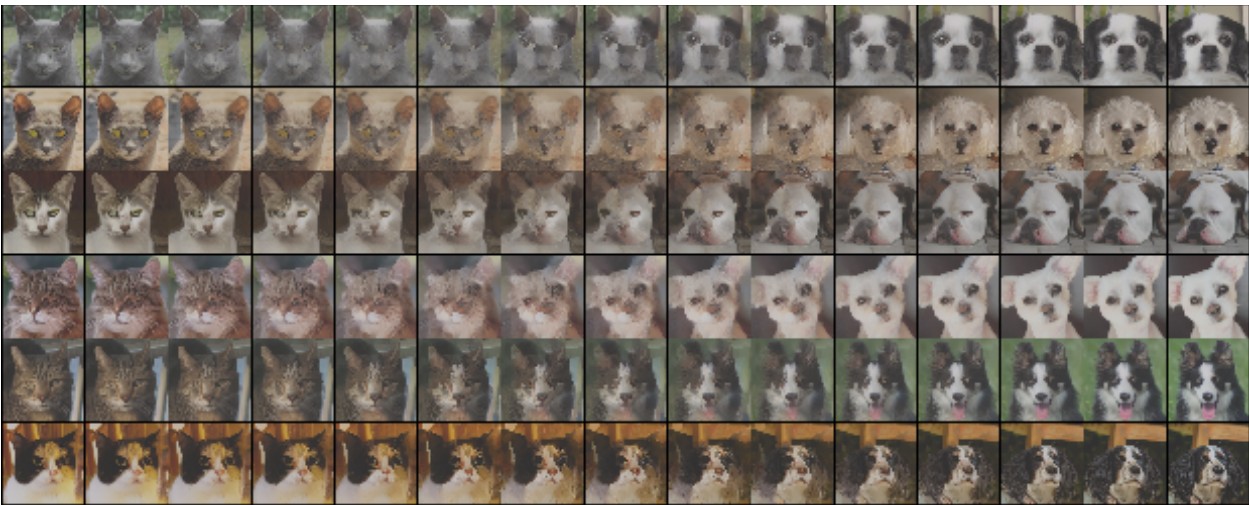

Figure 17: Image transitions on the AFHQ dataset (Choi et al., 2020a) with $\boldsymbol{G}_{\log p_\theta}$ (diagonal).

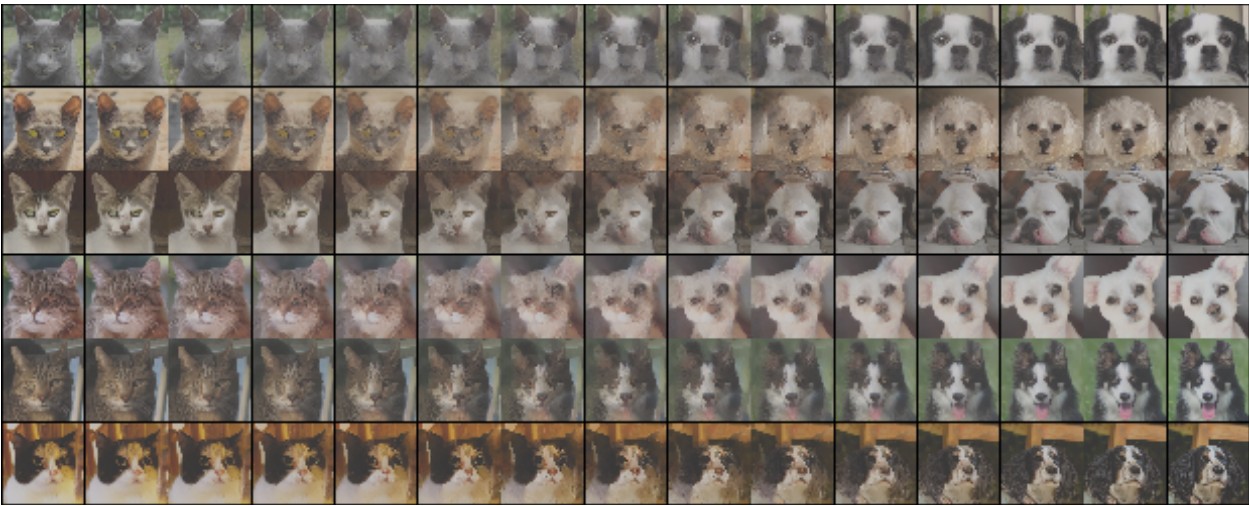

Figure 18: Image transitions on the AFHQ dataset (Choi et al., 2020a) with $\boldsymbol{G}_{\text{LAND}}$.

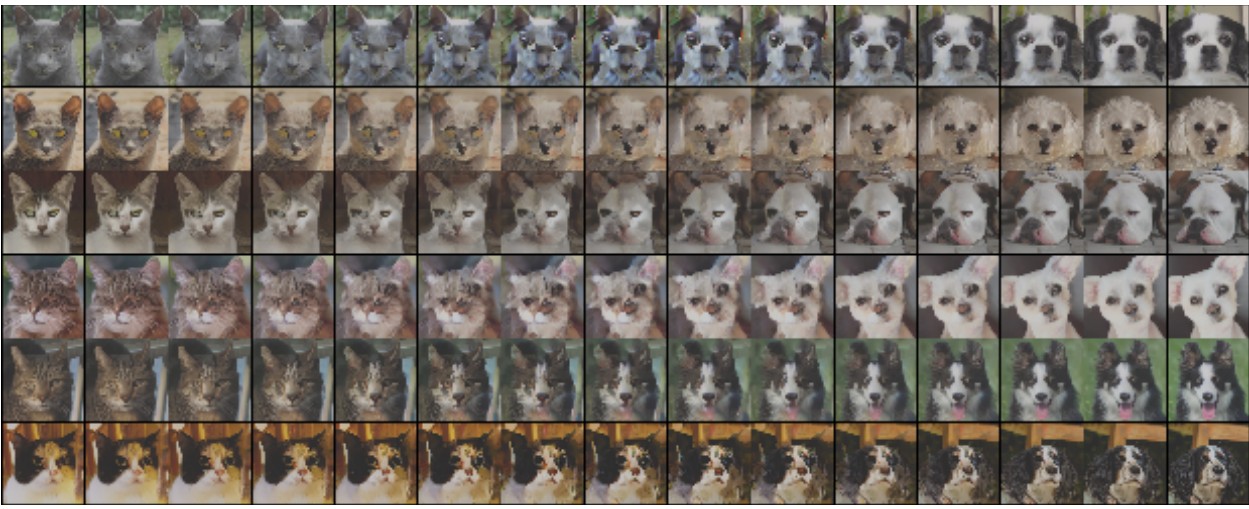

Figure 19: Image transitions on the AFHQ dataset (Choi et al., 2020a) with $\boldsymbol{G}_{1/p_\theta}$ (full).

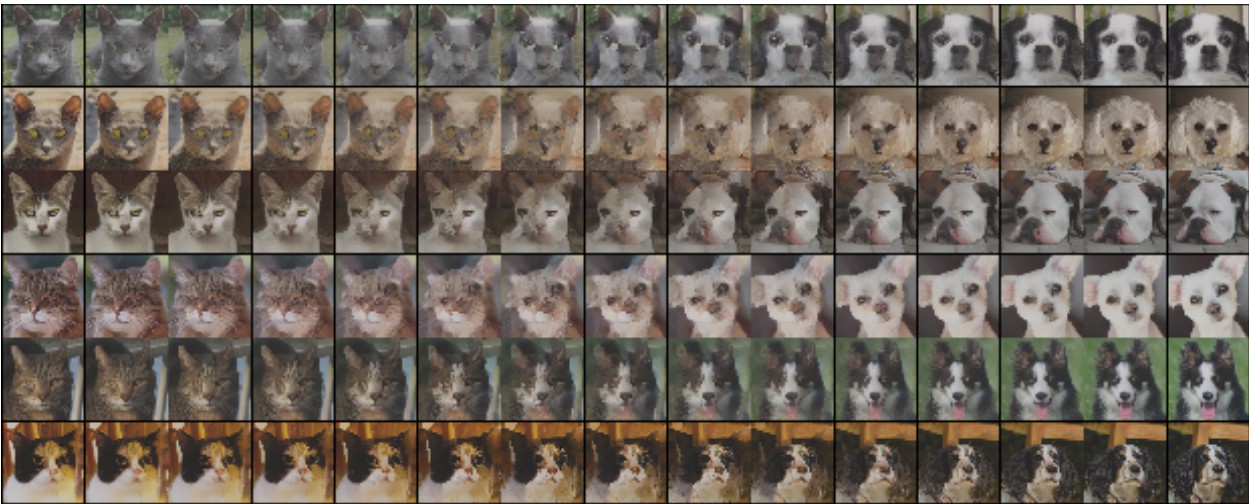

Figure 20: Image transitions on the AFHQ dataset (Choi et al., 2020a) with $\boldsymbol{G}_{\log p_\theta}$ (full).

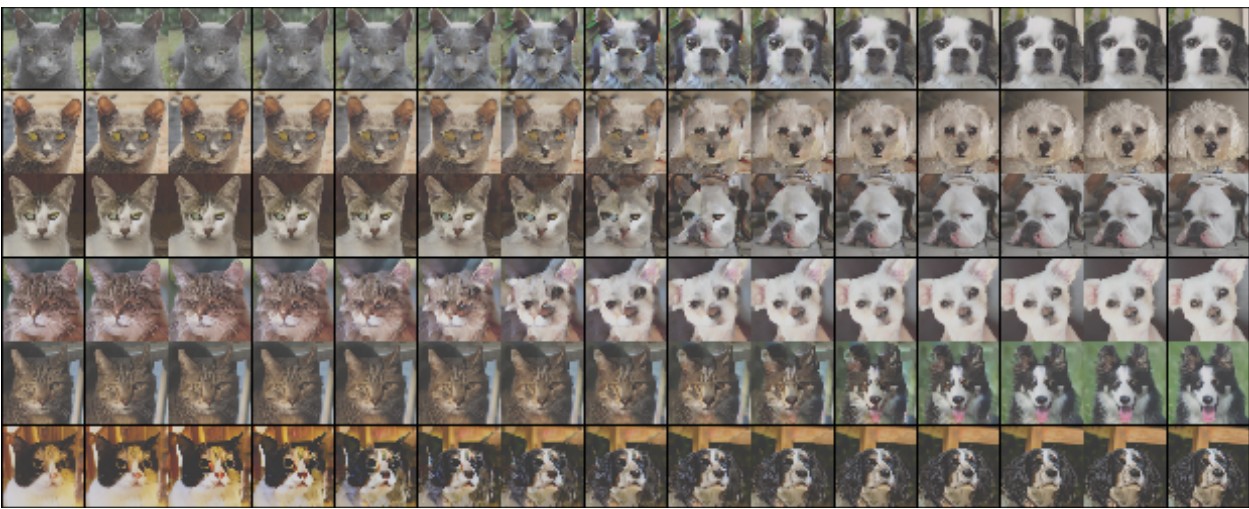

Figure 21: Image transitions on the AFHQ dataset (Choi et al., 2020a) with our method $(S = -1/p_\theta)$.

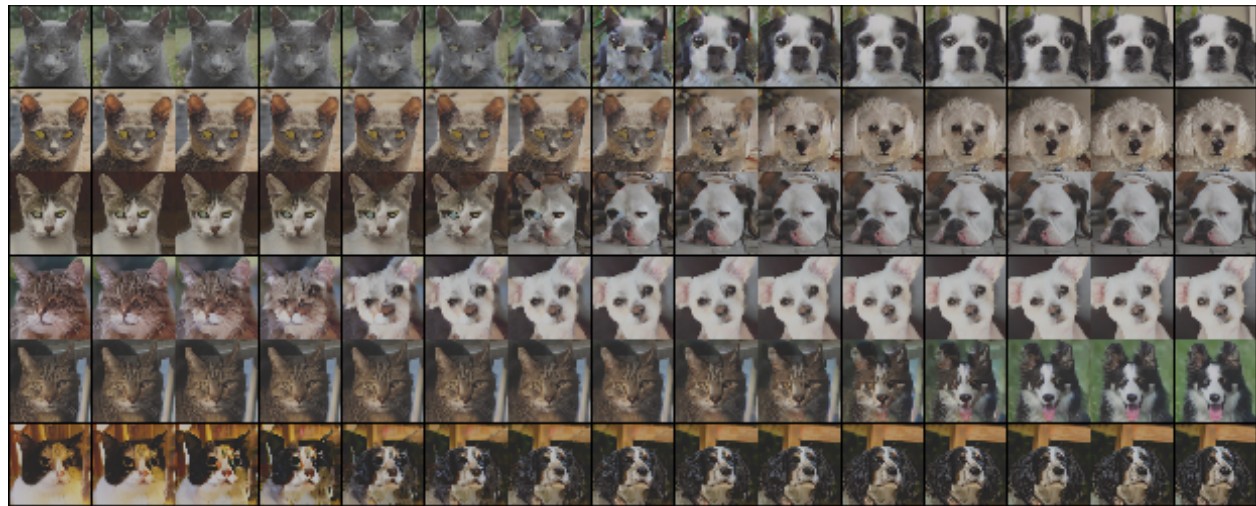

Figure 22: Image transitions on the AFHQ dataset (Choi et al., 2020a) with our method ($S = -\log p_\theta$).

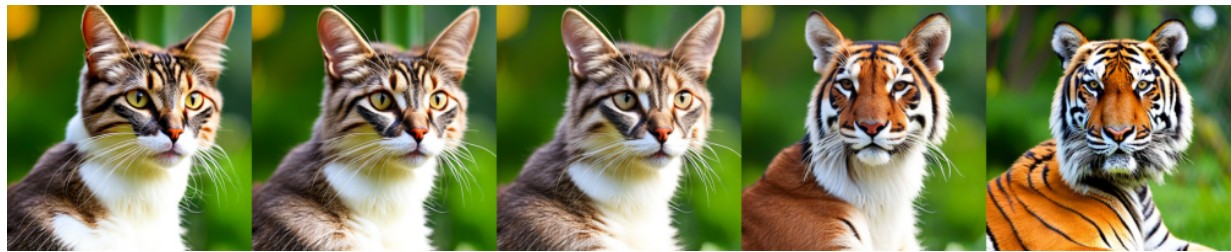

A photo of a tiger

Figure 23: A transition corresponding to an initial value problem using our method in noise space for ControlNet, where we state the target prompt under the images. The initial prompt was "A photo of a cat".

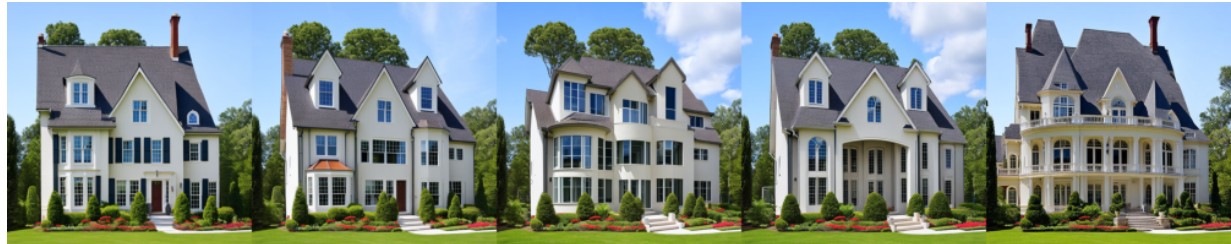

A photo of an old-fashioned mansion

Figure 24: A transition corresponding to an initial value problem using our method in noise space for ControlNet, where we state the target prompt under the images. The initial prompt was "A photo of a house".

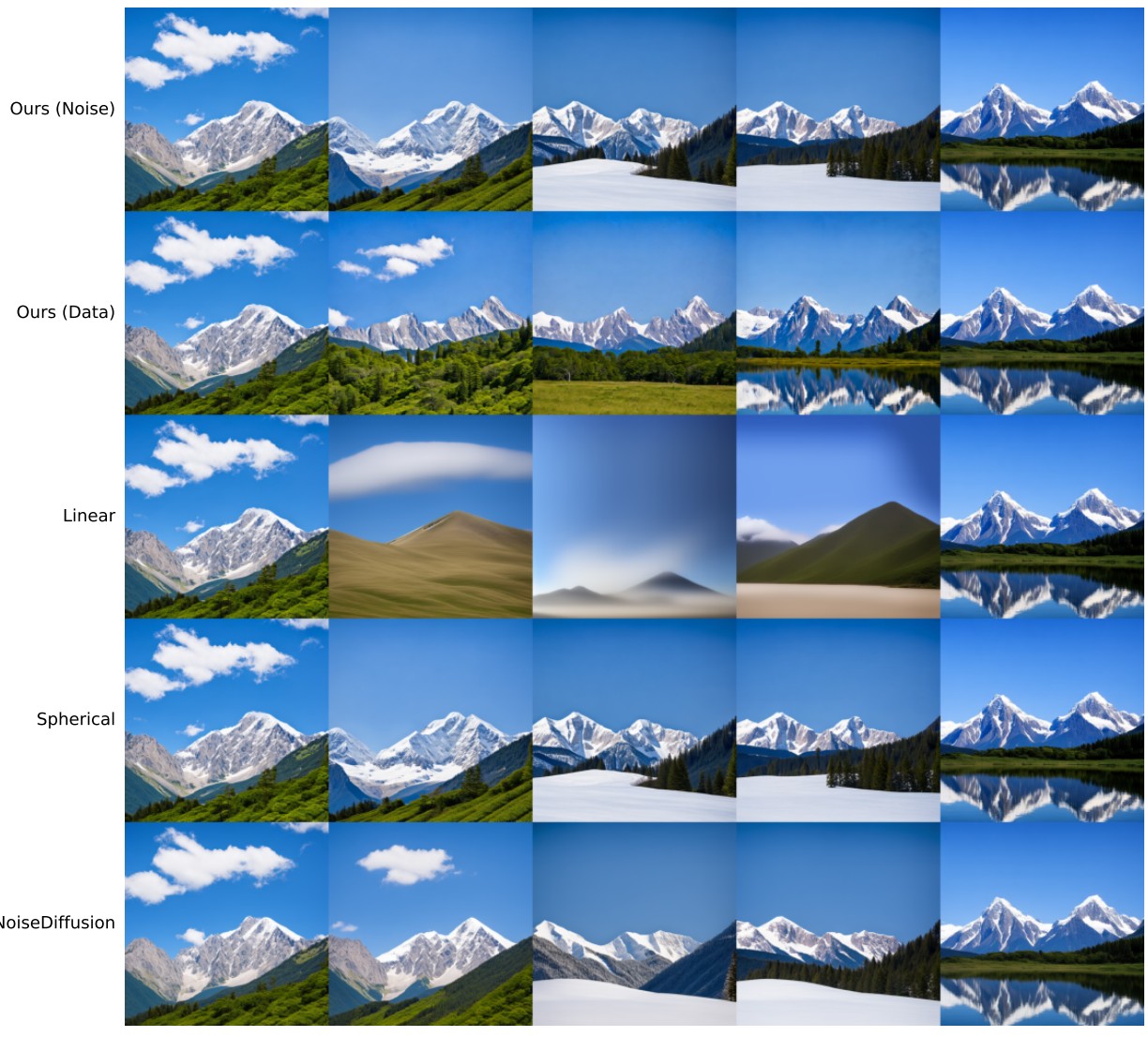

Figure 25: A display of different boundary value methods for ControlNet. The prompt was "A photo of a mountain".

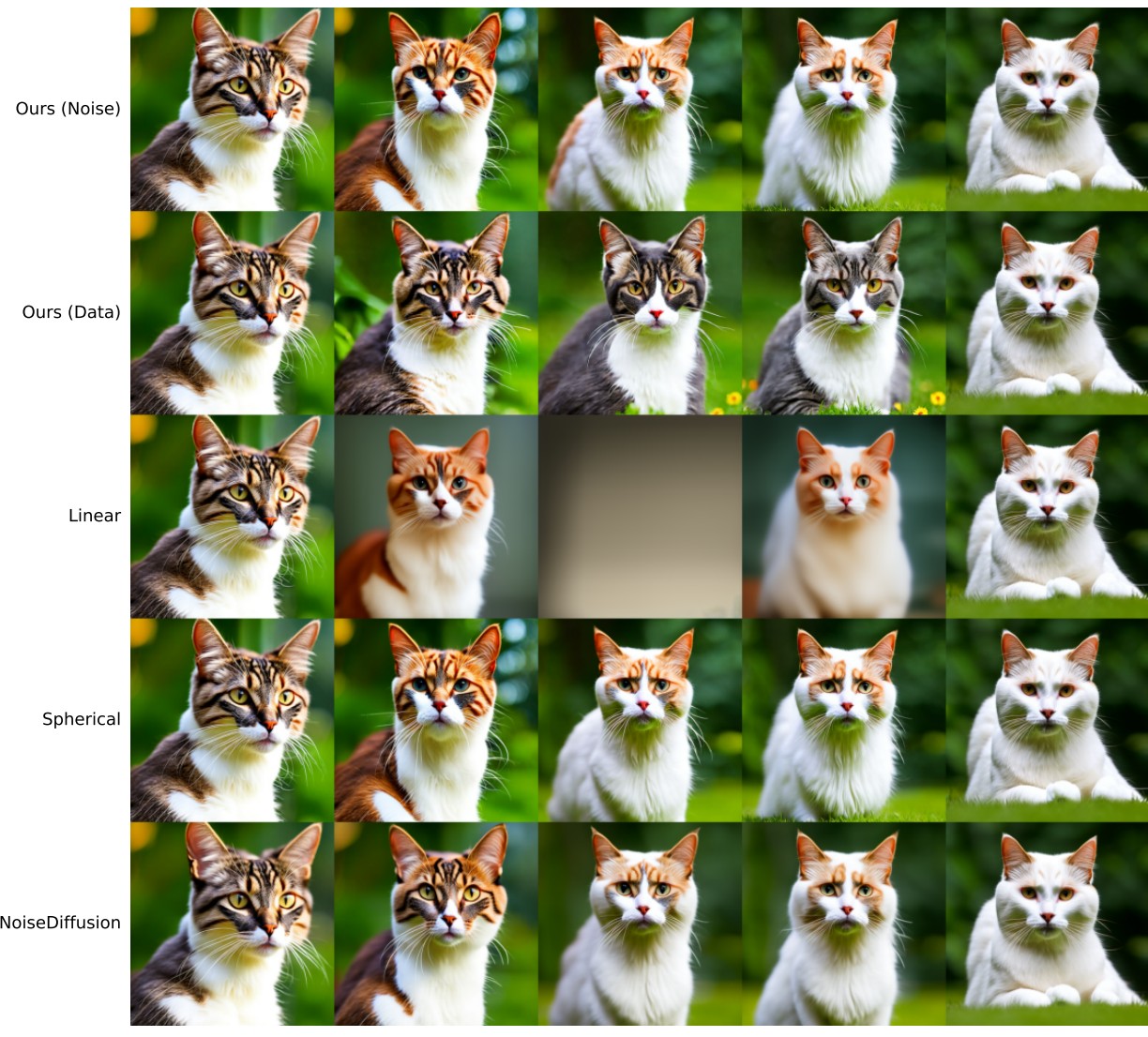

Figure 26: A display of different boundary value methods for ControlNet. The prompt was "A photo of a cat".

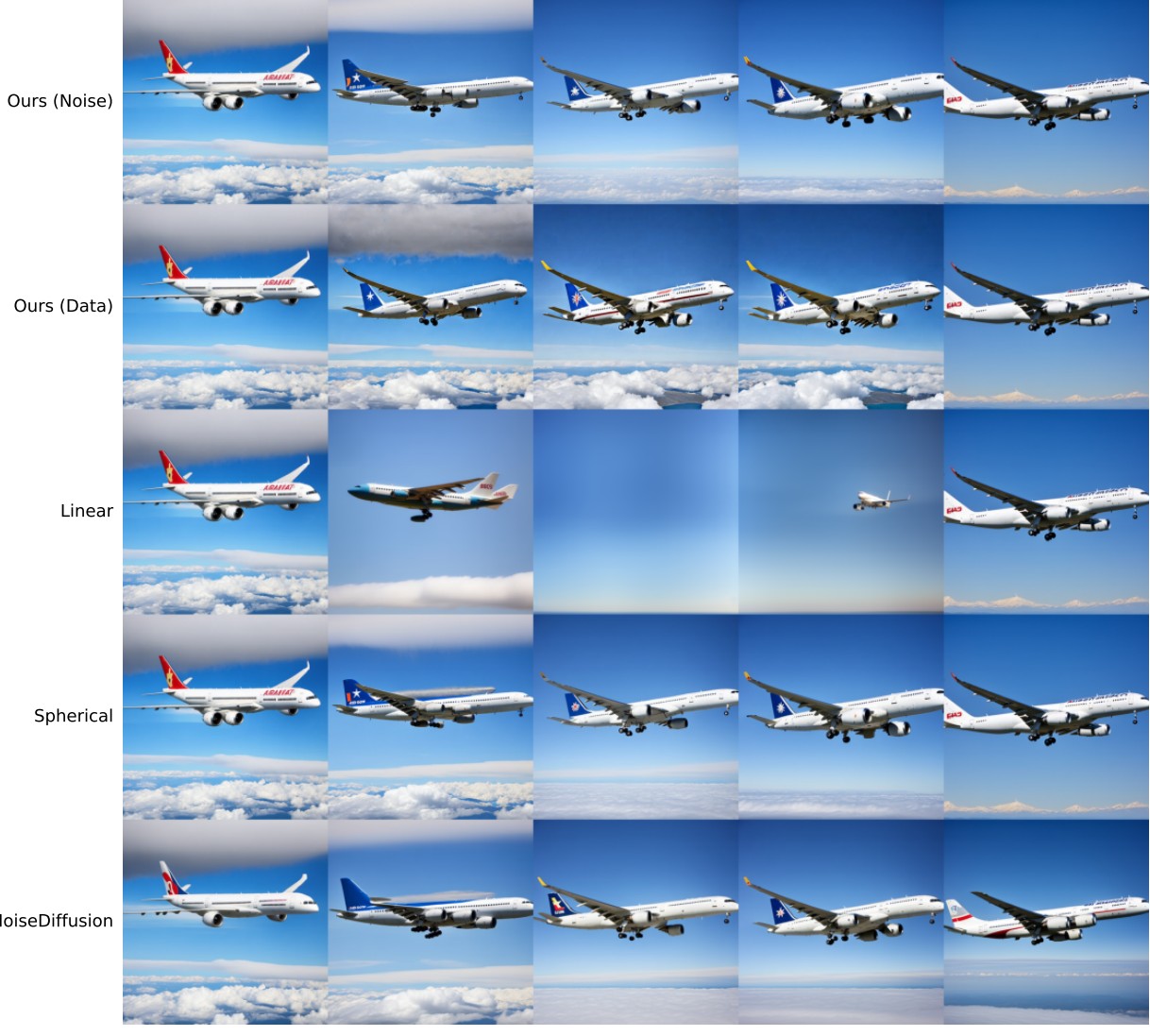

Figure 27: A display of different boundary value methods for ControlNet. The prompt was "A photo of an aircraft".

Spherical

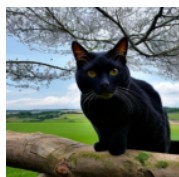

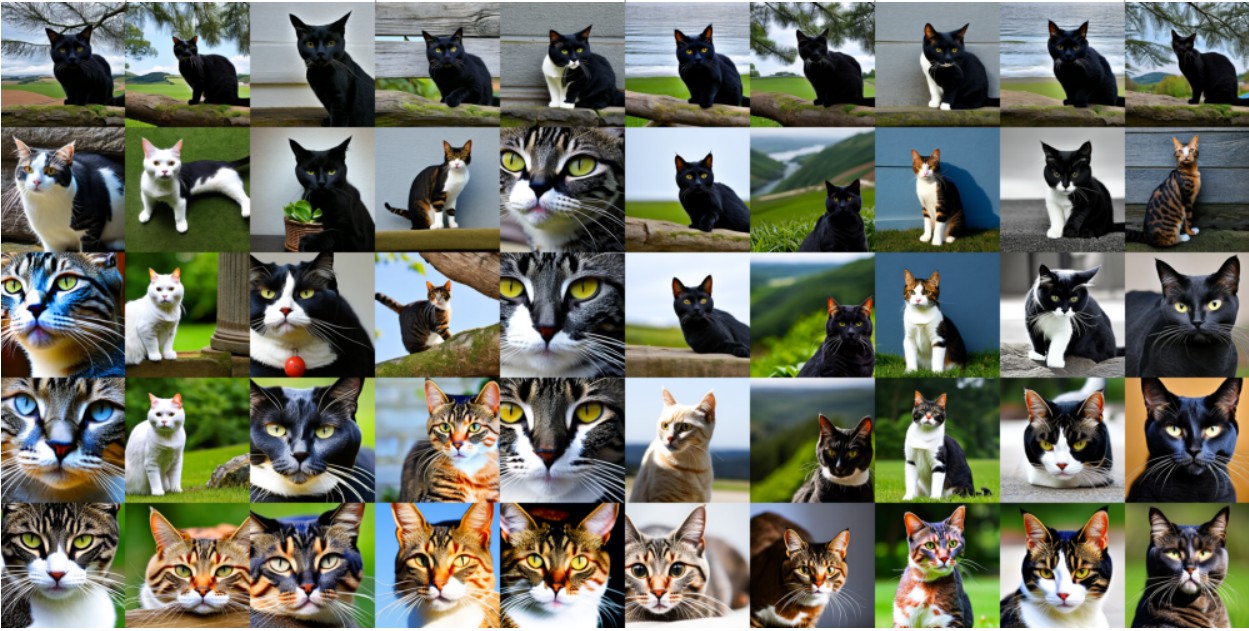

Data

Figure 28: The estimated mean using spherical geometry for the data in Fig. 6. The prompt was "A photo of a cat".

Linear

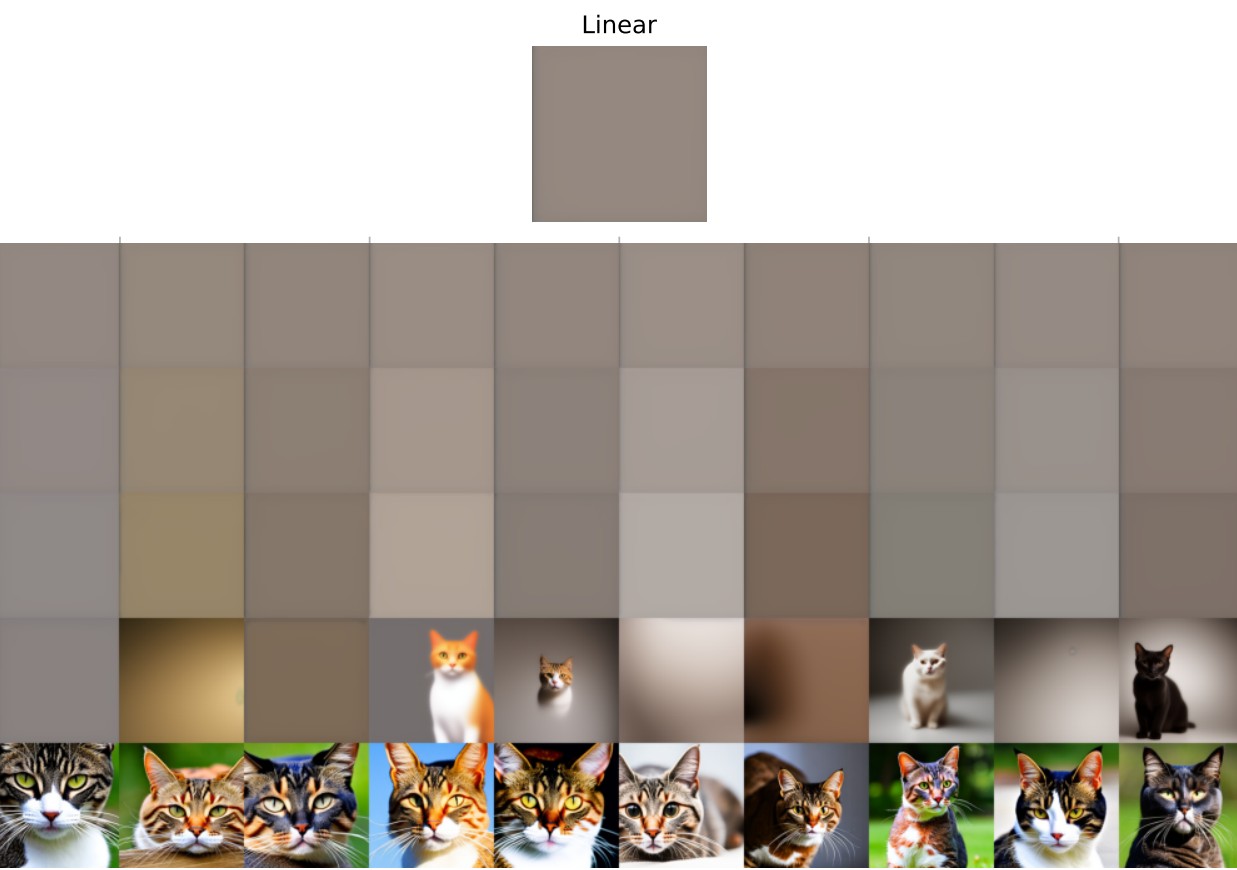

Figure 29: The estimated mean using linear geometry for the data in Fig. 6. The prompt was "A photo of a cat".

Ours (Noise)

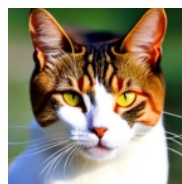

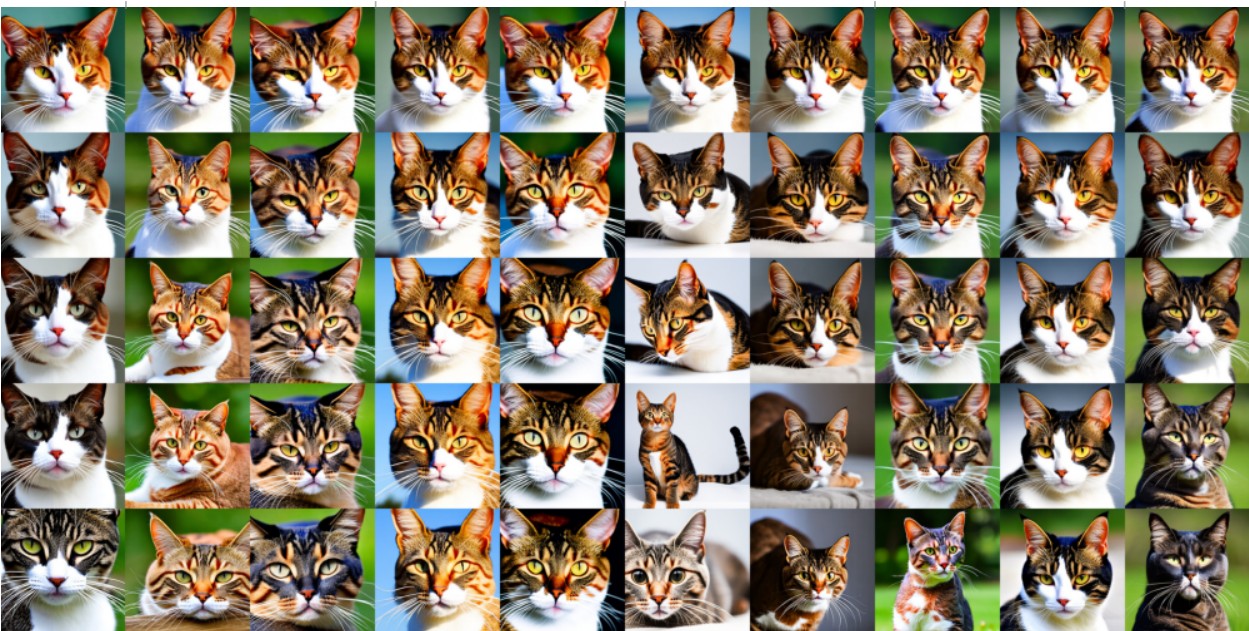

Data

Figure 30: The estimated mean using our method in noise space for the data in Fig. 6. The prompt was "A photo of a cat".

**Other diffusion models**   We consider the score-based generative model (SGM. (Song et al., 2021b), where we use the variance exploding SDE (VESDE) for both CIFAR10 and CelebAHQ datasets with SGM. For CIFAR10 interpolation, the minimum $\sigma$ and maximum $\sigma$ are set to 0.01 and 50 respectively, with the number of scales being 1000. For CelebAHQ interpolation, we use 0.01 and 348 as the minimum and maximum $\sigma$. The sampling $\epsilon$ is set to $1e-5$ for both cases. We also consider the latent score-based generative model (Vahdat et al., 2021) for the mentioned datasets as well as OASIS3 (LaMontagne et al., 2019) with LDM (Rombach et al., 2022), where the linear $\beta$ time schedule with 1000 time steps is adopted with the initial $\beta$ value being 0.0015 and the last $\beta$ value being 0.0205. For the score-based models, we sample using the probability flow ODE, while for the OASIS3 we use DDIM sampling from noise space (Song et al., 2021a). For our method, we set $\lambda = 1.0$ for both datasets and consider the regularization function.

$$S(x) = -\log p\left(\|x\|^2\right) + 0.1\left(\|x\|^2 - r^2\right),$$

where $r$ denotes the data dimension. Note that we apply this slightly different regularization function compared to ControlNet to account for the dimension of the noise space being lower. We compute the likelihood of the interpolation curves and FID for our method compared to linear and spherical interpolation in Table 8. In Fig. 31, 32 and 33, we provide interpolation paths.

Table 8: Mean log-likelihood / FID for interpolation in different diffusion models and datasets. SGM is score-based generative models (Kingma & Ba, 2014), while LSGM is latent score-based generative model (Vahdat et al., 2021). OASIS3 (LaMontagne et al., 2019) was trained using latent diffusion model (Rombach et al., 2022) used by Pinaya et al. (2022).

| Interpolation | OASIS3 | CIFAR10 (SGM) | CelebAHQ (SGM) | CIFAR10 (LSGM) | CelebAHQ (LSGM) |
|---|---|---|---|---|---|
| Linear | 27.83 / 85.94 | 0.33 / 391.64 | -1.61 / 285.32 | 26.49 / 254.39 | 1.51 / 149.53 |
| Spherical | **28.46** / 68.97 | **4.30** / 237.64 | -1.56 / 142.14 | **30.00** / **217.10** | **1.70** / 145.02 |
| ProbGEORCE | 28.41 / **66.54** | 4.12 / **233.84** | **-0.46** / **138.69** | 29.99 / 217.52 | **1.70** / **143.79** |

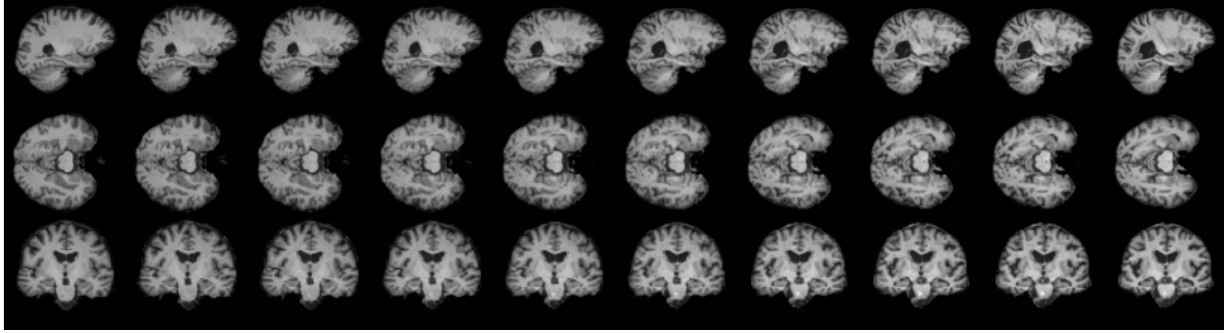

Figure 31: Interpolation using a latent diffusion model similar to (Pinaya et al., 2022) with our proposed method. Each row represents different slices of the brain. The interpolation shows the generative transition between a healthy brain (left) and a brain with Alzheimer disease (right).

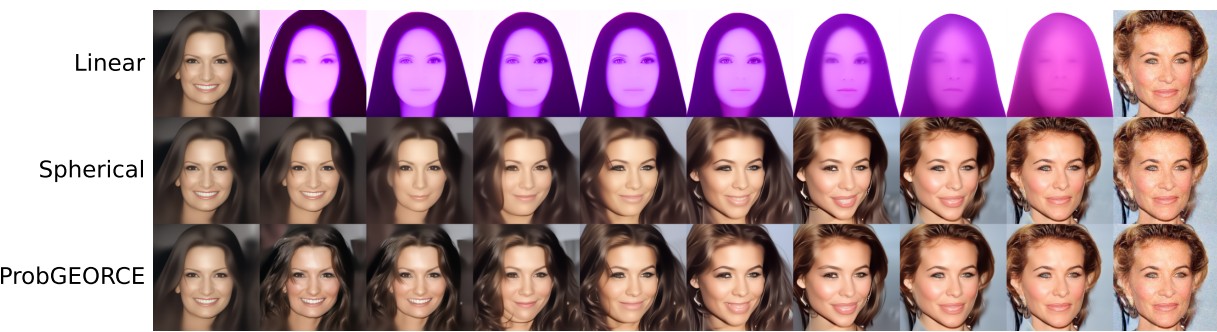

Figure 32: Comparisons of interpolation curves for CelebAHQ with SGM.

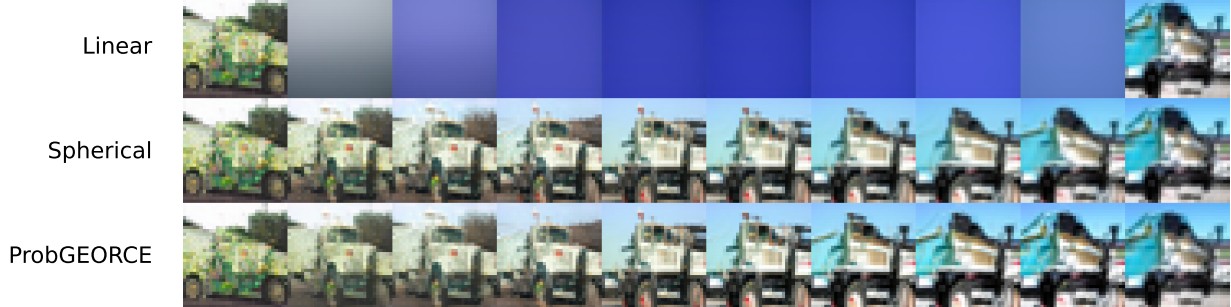

Figure 33: Comparisons of interpolation curves for CIFAR10 with SGM.

# G  Additional runtime estimates

In this section, we provide additional runtime estimates using our method compared to standard optimizers. We consider the BVP for Eq. 7 using our method *ProbGEORCE* with line-search and with adaptive update scheme. In Fig. 34, we show the estimated regularized energy and runtime for different optimizers for different values of $\lambda$. In general, we see that our method is faster for lower values of $\lambda$ and as $\lambda$ increases, the methods

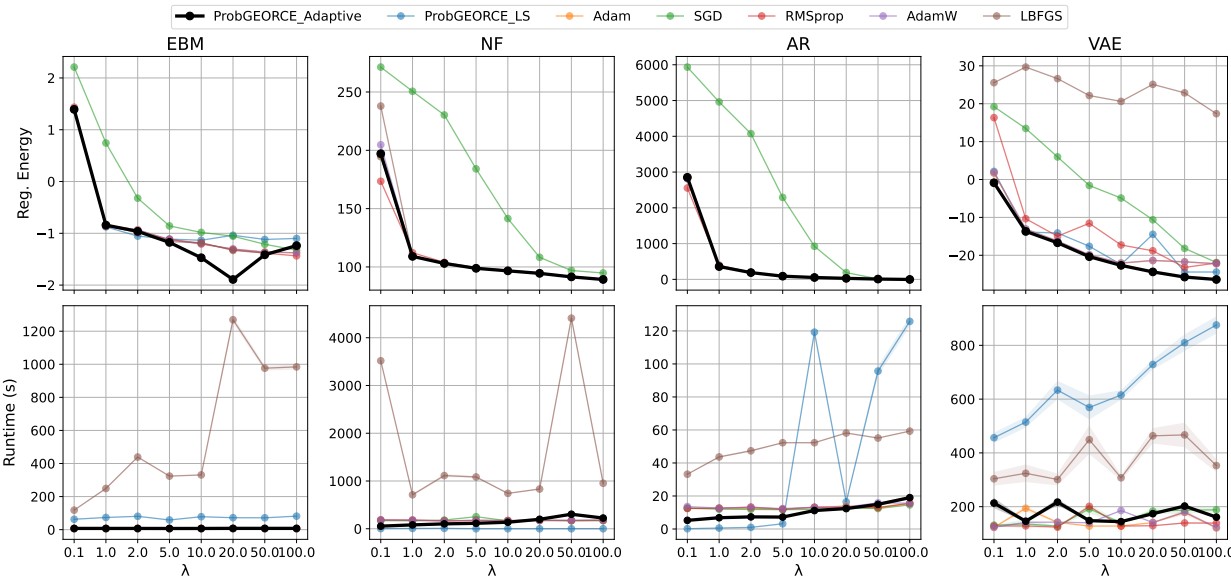

Figure 34: The regularized energy and runtime for different optimizers for the regularized energy in Eq. 7 solving the BVP for the models in Fig. 2.

converge to each other. This is expected as when $\lambda$ increases, the metric-term becomes negligible, and the regularization term dominates the expression. In this case, the minimization problem is approximately the same as minimizing the $S$ function, where *ProbGEORCE* does not exploit any specific structures of $S$ that can speed-up the computation.

We also see that the adaptive version is usually faster than line-search, especially if the regularized energy is expensive to evaluate as with the VAE, which is expected as this avoids repeated evaluations of the regularized energy functional. In Table 9, we show the corresponding runtimes for Fig. 34. Note that we terminate the methods if the gradient of the regularized energy is less than $10^{-4}$. Given the runtimes, it seems that no algorithms terminate beforehand, which is mostly likely for numerical reasons due to the neural network.

In Table 10, we consider the regularized energy and runtimes for different optimizers minimizing Eq. 7. We consider different manifolds and dimensions with $\lambda = 1.0$ and $S = -\log p$, where $p$ is the density of a mixture of three randomly weighted Gaussians with random means and the identity as covariance matrix. In this case, we see that *ProbGEORCE* with line-search is significantly faster than alternative methods and compared to the adaptive update scheme. This is expected since the regularization function is inexpensive to evaluate. In Fig. 35, we consider the $n$-sphere with $n = 10$ for increasing values of $\lambda$. In this case, we see that ProbGEORCE with line-search and the adaptive update scheme computes the smallest regularized energy, but also that the runtime increases using line-search as $\lambda$ increases, while the adaptive scheme is more stable in runtime.

Table 9: Runtime and regularized energy for different methods for different values of $\lambda$ for the methods in Fig. 2. For ProbGEORCE (LS) for NF, the neural networks estimate of the likelihood returned nans during the iterations, and therefore we set $-$.

| | EBM | | NF | | AR | | VAE | |
|---|---|---|---|---|---|---|---|---|
| Optimizer | Reg. Energy | Runtime | Reg. Energy | Runtime | Reg. Energy | Runtime | Reg. Energy | Runtime |
| $\lambda = 1.0$ | | | | | | | | |
| ProbGEORCE (Adaptive) | 0.124 | 7.45 ± 0.01 | **0.087** | 81.03 ± 0.39 | **0.115** | 6.83 ± 0.01 | 0.043 | 145.86 ± 0.53 |
| ProbGEORCE (LS) | **0.124** | 73.62 ± 0.29 | **0.087** | **6.42 ± 0.01** | 0.115 | **0.58 ± 0.00** | **0.037** | 514.56 ± 10.30 |
| Adam | 0.124 | 6.60 ± 0.00 | 0.087 | 181.43 ± 3.04 | 0.115 | 12.65 ± 0.03 | 0.045 | 194.05 ± 5.60 |
| SGD | 0.206 | **5.70 ± 0.01** | 0.111 | 166.12 ± 2.02 | 0.175 | 11.99 ± 0.02 | 0.150 | 135.86 ± 9.14 |
| RMSprop | 0.146 | 6.02 ± 0.04 | 0.107 | 178.22 ± 3.90 | 0.134 | 12.58 ± 0.02 | 0.068 | **128.54 ± 0.20** |
| AdamW | 0.124 | 6.68 ± 0.03 | 0.087 | 182.80 ± 1.65 | 0.115 | 12.67 ± 0.01 | 0.045 | 141.78 ± 0.72 |
| LBFGS | 0.124 | 248.88 ± 2.07 | 0.087 | 711.39 ± 10.12 | 0.115 | 43.66 ± 0.23 | 0.214 | 324.14 ± 16.69 |
| $\lambda = 5.0$ | | | | | | | | |
| ProbGEORCE (Adaptive) | **-0.126** | 7.11 ± 0.24 | **0.174** | 114.12 ± 1.32 | **0.125** | 7.23 ± 0.01 | **-0.289** | 147.95 ± 0.42 |
| ProbGEORCE (LS) | -0.107 | 58.47 ± 0.74 | - | **0.66 ± 0.00** | 0.125 | **3.16 ± 0.01** | -0.234 | 568.85 ± 23.03 |
| Adam | -0.101 | 6.28 ± 0.03 | 0.174 | 172.02 ± 2.80 | 0.125 | 12.02 ± 0.02 | -0.279 | **127.81 ± 0.18** |
| SGD | -0.010 | **5.61 ± 0.04** | 0.255 | 252.45 ± 0.02 | 0.283 | 11.68 ± 0.02 | 0.112 | 191.93 ± 9.31 |
| RMSprop | -0.085 | 6.28 ± 0.05 | 0.194 | 173.34 ± 2.73 | 0.146 | 11.96 ± 0.02 | -0.060 | 201.13 ± 1.66 |
| AdamW | -0.100 | 6.52 ± 0.03 | 0.174 | 179.02 ± 0.36 | 0.125 | 12.38 ± 0.07 | -0.279 | 139.85 ± 0.68 |
| LBFGS | -0.119 | 323.73 ± 0.31 | 0.174 | $1.08 \times 10^3$ ± 11.59 | 0.125 | 52.26 ± 0.05 | 0.582 | 448.98 ± 27.14 |
| $\lambda = 10.0$ | | | | | | | | |
| ProbGEORCE (Adaptive) | **-0.577** | 7.25 ± 0.05 | **0.278** | 136.70 ± 0.31 | 0.130 | **11.18 ± 0.07** | -0.789 | 144.33 ± 0.54 |
| ProbGEORCE (LS) | -0.368 | 78.16 ± 0.33 | - | **0.63 ± 0.01** | 0.130 | 119.25 ± 0.33 | **-0.794** | 615.24 ± 9.14 |
| Adam | -0.420 | 6.46 ± 0.05 | 0.278 | 172.64 ± 0.26 | 0.130 | 13.19 ± 0.02 | -0.763 | 128.05 ± 0.13 |
| SGD | -0.305 | **5.77 ± 0.02** | 0.363 | 165.85 ± 2.17 | 0.255 | 11.94 ± 0.33 | -0.028 | 140.18 ± 0.71 |
| RMSprop | -0.397 | 6.23 ± 0.01 | 0.298 | 171.90 ± 0.60 | 0.150 | 13.13 ± 0.05 | -0.480 | **127.77 ± 0.16** |
| AdamW | -0.409 | 6.51 ± 0.04 | 0.278 | 170.25 ± 0.36 | 0.130 | 13.15 ± 0.03 | -0.763 | 185.49 ± 1.18 |
| LBFGS | -0.409 | 330.91 ± 0.33 | 0.278 | 741.10 ± 2.46 | **0.130** | 52.27 ± 0.06 | 1.015 | 307.28 ± 5.25 |
| $\lambda = 20.0$ | | | | | | | | |
| ProbGEORCE (Adaptive) | **-1.791** | 7.83 ± 0.02 | **0.481** | 198.07 ± 1.98 | **0.136** | 12.38 ± 0.01 | **-1.875** | 173.77 ± 0.72 |
| ProbGEORCE (LS) | -0.903 | 72.13 ± 0.34 | - | **0.64 ± 0.00** | **0.136** | 16.41 ± 0.06 | -0.821 | 728.79 ± 9.63 |
| Adam | -1.157 | 6.35 ± 0.02 | 0.481 | 172.26 ± 1.23 | 0.136 | 13.85 ± 0.05 | -1.633 | 140.63 ± 0.78 |
| SGD | -0.930 | **5.68 ± 0.01** | 0.528 | 173.78 ± 0.46 | 0.181 | 12.71 ± 0.05 | -0.777 | 183.66 ± 5.63 |
| RMSprop | -1.149 | 7.30 ± 0.04 | 0.501 | 179.19 ± 0.30 | 0.156 | 13.08 ± 0.74 | -1.255 | **129.16 ± 0.21** |
| AdamW | -1.154 | 6.39 ± 0.02 | 0.481 | 173.33 ± 0.18 | 0.136 | 13.34 ± 0.02 | -1.630 | 141.13 ± 0.52 |
| LBFGS | -1.181 | $1.27 \times 10^3$ ± 10.42 | 0.481 | 832.08 ± 6.46 | **0.136** | 58.16 ± 0.05 | 2.360 | 463.50 ± 14.78 |
| $\lambda = 50.0$ | | | | | | | | |
| ProbGEORCE (Adaptive) | **-3.596** | 7.80 ± 0.03 | **1.074** | 304.15 ± 0.50 | 0.144 | 14.94 ± 0.27 | **-5.309** | 202.25 ± 0.22 |
| ProbGEORCE (LS) | -2.611 | 71.58 ± 0.19 | - | **0.63 ± 0.01** | 0.144 | 95.65 ± 1.56 | -5.018 | 809.95 ± 15.94 |
| Adam | -3.490 | 6.39 ± 0.04 | 1.074 | 171.42 ± 0.36 | 0.144 | 13.04 ± 0.25 | -4.231 | 178.80 ± 4.90 |
| SGD | -3.134 | **5.69 ± 0.02** | 1.110 | 168.95 ± 2.57 | 0.149 | **12.41 ± 0.13** | -3.666 | 186.83 ± 6.66 |
| RMSprop | -3.513 | 6.29 ± 0.04 | 1.094 | 171.08 ± 2.67 | 0.165 | 12.99 ± 0.22 | -4.596 | **139.37 ± 1.44** |
| AdamW | -3.487 | 6.37 ± 0.02 | 1.074 | 178.96 ± 0.58 | 0.144 | 15.84 ± 0.13 | -4.230 | 179.64 ± 4.45 |
| LBFGS | -3.318 | 976.39 ± 7.88 | - | $4.41 \times 10^3$ ± 19.60 | **0.144** | 55.11 ± 0.04 | 5.274 | 466.87 ± 23.32 |
| $\lambda = 100.0$ | | | | | | | | |
| ProbGEORCE (Adaptive) | -5.980 | 7.78 ± 0.04 | 2.034 | 224.24 ± 2.25 | 0.146 | 19.00 ± 0.31 | **-11.225** | 161.09 ± 0.76 |
| ProbGEORCE (LS) | -5.787 | 81.56 ± 0.56 | - | **0.66 ± 0.00** | 0.148 | 125.78 ± 1.56 | -10.326 | 875.75 ± 16.40 |
| Adam | -7.405 | 6.26 ± 0.03 | 2.034 | 173.19 ± 1.58 | 0.146 | 15.60 ± 0.15 | -9.053 | **120.66 ± 0.22** |
| SGD | -7.192 | **5.64 ± 0.04** | 2.108 | 169.45 ± 2.83 | 0.151 | **14.60 ± 0.13** | -9.018 | 188.26 ± 5.28 |
| RMSprop | **-7.763** | 6.17 ± 0.05 | 2.052 | 179.11 ± 0.22 | 0.168 | 15.50 ± 0.02 | -8.921 | 138.88 ± 0.60 |
| AdamW | -7.400 | 6.32 ± 0.01 | 2.034 | 182.77 ± 0.33 | 0.146 | 15.64 ± 0.12 | -9.033 | 122.84 ± 0.23 |
| LBFGS | -4.619 | 984.25 ± 8.40 | **2.034** | 952.73 ± 12.46 | **0.146** | 59.30 ± 0.07 | 7.969 | 352.84 ± 12.44 |

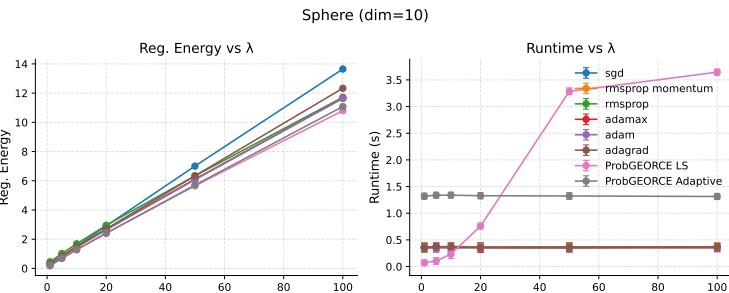

Figure 35: The regularized energy and runtimes for different optimizers minimizing Eq. 7. We consider the $n$-sphere with $n = 10$ for different values of $\lambda$ and $S = -\log p$, where $p$ is the density of a mixture of three randomly weighted Gaussians with random means and the identity as covariance matrix. We report the regularized energy and runtime (mean runtime $\pm$ standard deviation runtime over five runs repeated five times).

Table 10: The regularized energy and runtimes for different optimizers minimizing Eq. 7. We consider different manifolds and dimensions with $\lambda = 1.0$ and $S = -\log p$, where $p$ is the density of a mixture of three randomly weighted Gaussians with random means and the identity as covariance matrix. We report the regularized energy and runtime (mean runtime ± standard deviation runtime over five runs repeated five times).

| Manifold | sgd | | rmsprop momentum | | rmsprop | | adamax | | adam | | adagrad | | ProbGEORCE LS | | ProbGEORCE Adaptive | |
|---|---|---|---|---|---|---|---|---|---|---|---|---|---|---|---|---|
| | Reg. Energy | Runtime | Reg. Energy | Runtime | Reg. Energy | Runtime | Reg. Energy | Runtime | Reg. Energy | Runtime | Reg. Energy | Runtime | Reg. Energy | Runtime | Reg. Energy | Runtime |
| Cauchy (dim=2) | 0.0853 | 0.344 ± 0.037 | 539.4545 | 0.350 ± 0.033 | 0.1059 | 0.358 ± 0.033 | 0.0839 | 0.333 ± 0.033 | 0.0843 | 0.344 ± 0.035 | 0.0839 | 0.340 ± 0.033 | **0.0689** | **0.037 ± 0.029** | 0.0689 | 0.614 ± 0.034 |
| Ellipsoid (dim=2) | 0.0854 | 0.330 ± 0.035 | 0.1108 | 0.344 ± 0.036 | 0.1282 | 0.361 ± 0.045 | 0.0887 | 0.352 ± 0.033 | 0.0894 | 0.334 ± 0.033 | 0.0892 | 0.333 ± 0.041 | **0.0833** | **0.026 ± 0.027** | 0.0833 | 0.487 ± 0.026 |
| Ellipsoid (dim=3) | 0.0967 | 0.328 ± 0.036 | 0.1190 | 0.338 ± 0.030 | 0.1540 | 0.319 ± 0.036 | 0.1010 | 0.365 ± 0.027 | 0.1019 | 0.365 ± 0.027 | 0.1014 | 0.323 ± 0.036 | **0.0929** | **0.034 ± 0.028** | 0.0929 | 0.665 ± 0.031 |
| Ellipsoid (dim=5) | 0.1151 | 0.353 ± 0.030 | 0.1669 | 0.346 ± 0.027 | 0.1954 | 0.350 ± 0.039 | 0.1223 | 0.382 ± 0.033 | 0.1238 | 0.354 ± 0.036 | 0.1227 | 0.342 ± 0.033 | **0.0969** | **0.090 ± 0.031** | 0.0974 | 1.125 ± 0.032 |
| Ellipsoid (dim=10) | 0.1441 | 0.369 ± 0.036 | 0.2233 | 0.381 ± 0.032 | 0.2655 | 0.371 ± 0.036 | 0.1583 | 0.374 ± 0.039 | 0.1618 | 0.376 ± 0.036 | 0.1588 | 0.382 ± 0.034 | **0.1038** | **0.147 ± 0.034** | 0.1054 | 1.325 ± 0.030 |
| Ellipsoid (dim=20) | 0.1682 | 0.463 ± 0.030 | 0.3182 | 0.460 ± 0.028 | 0.3240 | 0.467 ± 0.035 | 0.1870 | 0.427 ± 0.032 | 0.1951 | 0.457 ± 0.038 | 0.1889 | 0.459 ± 0.028 | **0.1024** | **0.092 ± 0.031** | 0.1028 | 1.655 ± 0.029 |
| Ellipsoid (dim=50) | 0.1685 | 0.767 ± 0.030 | 0.4460 | 0.757 ± 0.031 | 0.3461 | 0.699 ± 0.029 | 0.1797 | 0.707 ± 0.033 | 0.1922 | 0.714 ± 0.031 | 0.1842 | 0.756 ± 0.032 | **0.0903** | **0.160 ± 0.035** | 0.0905 | 2.463 ± 0.032 |
| Ellipsoid (dim=100) | 0.1382 | 1.597 ± 0.030 | 0.4146 | 1.615 ± 0.031 | 0.3022 | 1.614 ± 0.031 | 0.1334 | 1.605 ± 0.031 | 0.1378 | 1.622 ± 0.030 | 0.1351 | 1.623 ± 0.033 | **0.0664** | **0.590 ± 0.031** | 0.0669 | 2.519 ± 0.030 |
| Frechet (dim=2) | 0.0291 | 0.349 ± 0.036 | 2421.0039 | 0.327 ± 0.029 | 0.0917 | 0.327 ± 0.039 | 0.0286 | 0.328 ± 0.030 | 0.0291 | 0.363 ± 0.027 | 0.0286 | 0.328 ± 0.030 | **0.0283** | **0.021 ± 0.027** | 0.0283 | 0.363 ± 0.031 |
| Gaussian (dim=2) | - | - | 1222.1246 | 0.341 ± 0.034 | 0.2471 | 0.361 ± 0.031 | 0.1752 | 0.335 ± 0.028 | 0.1758 | 0.345 ± 0.032 | 0.1754 | 0.356 ± 0.030 | **0.1541** | **0.029 ± 0.028** | 0.1541 | 0.700 ± 0.029 |
| Pareto (dim=2) | 0.0150 | 0.342 ± 0.031 | 2151.8325 | 0.353 ± 0.033 | 0.0465 | 0.356 ± 0.036 | 0.0146 | 0.331 ± 0.031 | 0.0149 | 0.359 ± 0.036 | 0.0146 | 0.331 ± 0.029 | **0.0145** | **0.021 ± 0.027** | 0.0145 | 0.338 ± 0.030 |
| SPDN (dim=2) | 0.0275 | 0.485 ± 0.029 | 0.0659 | 0.450 ± 0.032 | 0.1574 | 0.456 ± 0.030 | 0.0265 | 0.442 ± 0.033 | 0.0259 | 0.464 ± 0.033 | 0.0265 | 0.448 ± 0.030 | **0.0251** | **0.042 ± 0.029** | 0.0251 | 0.667 ± 0.030 |
| SPDN (dim=3) | 0.1197 | 0.504 ± 0.030 | 0.2602 | 0.494 ± 0.030 | 0.5051 | 0.514 ± 0.029 | 0.1185 | 0.500 ± 0.030 | 0.1196 | 0.522 ± 0.029 | 0.1184 | 0.517 ± 0.030 | **0.1126** | **0.072 ± 0.031** | 0.1126 | 0.899 ± 0.031 |
| Sphere (dim=2) | 0.1208 | 0.364 ± 0.036 | 0.1433 | 0.324 ± 0.035 | 0.1739 | 0.320 ± 0.036 | 0.1224 | 0.358 ± 0.035 | 0.1228 | 0.331 ± 0.037 | 0.1226 | 0.330 ± 0.040 | **0.1160** | **0.026 ± 0.029** | 0.1160 | 0.701 ± 0.031 |
| Sphere (dim=3) | 0.1546 | 0.314 ± 0.041 | 0.1897 | 0.313 ± 0.035 | 0.2310 | 0.335 ± 0.034 | 0.1584 | 0.339 ± 0.030 | 0.1591 | 0.327 ± 0.032 | 0.1587 | 0.341 ± 0.038 | **0.1462** | **0.042 ± 0.030** | 0.1462 | 1.031 ± 0.029 |
| Sphere (dim=5) | 0.2048 | 0.340 ± 0.040 | 0.2585 | 0.319 ± 0.038 | 0.3151 | 0.343 ± 0.036 | 0.2075 | 0.377 ± 0.031 | 0.2080 | 0.352 ± 0.030 | 0.2076 | 0.319 ± 0.048 | **0.1653** | **0.050 ± 0.026** | 0.1655 | 1.049 ± 0.032 |
| Sphere (dim=10) | 0.2749 | 0.362 ± 0.038 | 0.3606 | 0.364 ± 0.038 | 0.4526 | 0.355 ± 0.036 | 0.2820 | 0.353 ± 0.042 | 0.2837 | 0.353 ± 0.036 | 0.2822 | 0.358 ± 0.041 | **0.1873** | **0.071 ± 0.028** | 0.1879 | 1.319 ± 0.029 |
| Sphere (dim=20) | 0.3242 | 0.423 ± 0.029 | 2.8070 | 0.462 ± 0.030 | 0.5753 | 0.421 ± 0.030 | 0.3325 | 0.440 ± 0.029 | 0.3339 | 0.435 ± 0.030 | 0.3310 | 0.461 ± 0.031 | **0.1893** | **0.122 ± 0.037** | 0.1894 | 1.642 ± 0.031 |
| Sphere (dim=50) | 0.3152 | 0.700 ± 0.033 | 1.3125 | 0.687 ± 0.031 | 0.6369 | 0.706 ± 0.031 | 0.3247 | 0.691 ± 0.031 | 0.3292 | 0.712 ± 0.029 | 0.3221 | 0.685 ± 0.031 | **0.1654** | **0.243 ± 0.033** | 0.1654 | 3.050 ± 0.029 |
| Sphere (dim=100) | 0.2504 | 1.475 ± 0.032 | 2.1177 | 1.653 ± 0.028 | 0.5268 | 1.530 ± 0.030 | 0.2441 | 1.592 ± 0.032 | 0.2409 | 1.639 ± 0.029 | 0.2386 | 1.612 ± 0.030 | **0.1177** | **0.401 ± 0.030** | 0.1179 | 4.443 ± 0.030 |
| T2 (dim=2) | 2.7905 | 0.349 ± 0.036 | 2.7950 | 0.348 ± 0.038 | 2.8467 | 0.343 ± 0.038 | 2.7551 | 0.361 ± 0.027 | 2.7582 | 0.347 ± 0.035 | 2.6305 | 0.353 ± 0.027 | **2.4578** | **0.078 ± 0.029** | 2.4578 | 0.925 ± 0.029 |

