# OpenReview forum: "A Likely Geometry of Generative Models"
_TMLR — Under review for TMLR_

### Review · Reviewer_u74y · 2026-07-02

**Summary Of Contributions:**

The authors proposed ProbGEORCE, a probabilistic extension of GEORCE to find (relaxed) geodesics of a Riemannian manifold on which a generative model resides. The authors showed that the learned geodesic can be used to interpolate and define averages in that manifold without the model explicitly learning it.

**Additional Comments:**

- Why does spherical geometry in noise space lead to a black cat while everyone else is a tabby?

**Audience:**

Yes

**Audience Explanation:**

- I am clearly interested in this.
- Beyond image editing, in many scientific applications one might want to use generative models, e.g., as surrogates for expensive simulations where interpolation within the low-dimensional manifold is essential for success.

**Broader Impact Concerns:**

No such concerns.

**Claims And Evidence:**

Yes

**Claims Explanation:**

- Eq. 4 is a well-defined optimization problem that intuitively captures the author's goal of finding a geodesic within high-probability regions, and theory supports that.
- Fig. 4 provides a toy setup where the mean and geodesics seem reasonable.
- The challenge for more "real" datasets is that we do not have ground truth to compare to, so the authors showed the interpolations have good FID scores, at least showing that the interpolations are still within a high-probability region of the manifold. There are ways this logic might fail, but I cannot come up with better ways to test real data, so I would say this evidence is as good as it gets.

**Requested Changes:**

My requested changes are mostly on the presentation side

- I think the authors have a differential geometry or physics background. The paper does follow the standard notations, e.g., the Euler-Lagrange equation, but it might be a bit foreign to people from, e.g., the pure generative modeling community. It would be better to explain the notations. E.g., a dot on top means a derivative over time. Terminology like Christoffel symbols is better explained when they first appear as well.
- I am not sure I fully understood how the metric tensor $g$ is selected. Do we have to know the tensor to use the method? It appears to me that one has several choices, like Ours $-\log p$ in Table 2. Does that require us knowing the density? A clarification of what is required to be known in the method section will be helpful. The algorithm in the appendix did not help me, as, e.g., in Algo 1 line 5, to get $\tilde{G}_{s}^k $ one needs a function $\tilde{G}_s$ which is not an input...

---

> ### Author Response · Authors · 2026-07-17
>
> We are thankful for the review. In the following, we answer the questions and requested changes.
>
> **Requested changes**
> * We agree that we can explain the notation in Riemannian manifolds more clearly. We have clarified the notation in the introduction and background section in the updated paper under Eq. 1 and over Eq. 3. We have also modified the background section on Riemannian manifolds so that it is more accessible.
> * In our method, we assume that we know the metric tensor, but this can in principle also be learned. The metric is for all experiments set to the metric of the ambient space where the generative model relies. The geodesics are then controlled by a mix of the ambient metric and the likelihood term in the regularization, where the likelihood term pushes the curves to high-likelihood areas. In our paper, all experiments are with generative models in Euclidean space, so $G=I$ is the Euclidean metric. We have designed our framework, so one can easily change this and apply our method for generative models residing on Riemannian manifolds, e.g., learning a generative model on spherical data. For the regularization function, $S$, we choose this to be the negative log-likelihood of the  data distribution. This assumes that the generative model learns an estimate of the data distribution. In practice, the algorithms derived only require the gradient (score) of the data distribution. Therefore, we are also able to apply our framework for diffusion models, which does not necessarily learn the data distribution, $-\log p_{t}$, but instead the score $-\nabla_{z} \log p_{t}$. In general, we assume an underlying metric $g$ and access to the density of the data distribution (approximated by the generative model). We have added this to the beginning of section 3 in the updated version.
>
> In Algorithm 1, we compute $G_{s}$ by evaluating the metric matrix function of grid point $z_{s}$. Line 5 is therefore a typo and
>
> $$
> \tilde{G}(z_{s})
> $$
>
> should be replaced by
>
> $$
> G(z_{s})
> $$
>
> We have corrected this in the updated paper in Algorithm~1.
>
> **Additional comments**
> * There are several plausible reasons why the spherical geometry yields a mean that decodes to a black cat, whereas our method yields a tabby cat. The primary reason is that the two methods solve different optimization problems and, therefore, estimate different means. The spherical approach computes the intrinsic Fr\'echet mean under the spherical Riemannian metric, whereas our method computes a Fr\'echet mean under a Euclidean metric regularized by the log-density of the $\chi^{2}$-distribution on the norms of the curve points. In high dimensions, this regularization encourages norms consistent with the typical Gaussian shell while still allowing radial variation, whereas the spherical formulation assumes a fixed norm throughout the optimization. Consequently, the two approaches need not produce the same representative point in the diffusion noise space. Since the mapping from diffusion noise to images is highly nonlinear, even relatively small differences between the estimated means can lead to noticeable semantic differences in the generated images. A secondary factor is that the two methods imply different optimization procedures. For the spherical geometry, we can efficiently estimate the Fr\'echet mean in the embedded space iteratively using the gradient
>
> $$
> \sum_{i=1}^{N}\mathrm{Log}_{\mu}(z_i)
> $$
>
> For our method, we use ProbGEOCRCE. This also means that it is possible to obtain a difference due to the different optimization algorithms. We have added this explanation in the experiment section in the updated version on page 12.
>
> **Additional changes**
> * The original GEORCE-paper, from which our extended algorithm used in the paper is built on, claimed local quadratic convergence. We were recently made aware of an error in their proof. We have corrected this in our paper and instead have shown that ProbGEORCE has locally asymptotic quadratic convergence in the number of grid points under the assumption that $S$ is an affine function. This is in line with the runtime results shown in Table 8 in Appendix G, where the speed-up will occur for smaller values of $\lambda$, and as $\lambda$ increases, the speed-up of our method will decay. We have updated Appendix A.4 with this and also any mention of local quadratic convergence. We apologize for this.

---

### Review · Reviewer_o4MG · 2026-07-07

**Summary Of Contributions:**

This paper proposes to revist how to examine the latent spaces of generative models. The authors argue that generative models naturally lend themselves to a Riemannian geometry, and that factoring in the likelihood of specific samples induces a Newtonian structure on a Riemannian manifold.

The paper's contributions include a more principle way of interpolating between different samples or deriving a notion of mean. It is competitive compared to other approaches, such as spherical interpolation, and at face value appears to be more principled, in that it makes fewer assumptions about the model at hand.

**Audience:**

Yes

**Audience Explanation:**

The ability to provide a more principled way to explore a model's latent space is of clear value to the TMLR community. This paper has immediate implications for a significant portion of TMLR's readership, in particular those interested e.g. in interpretability work.

**Claims And Evidence:**

Yes

**Claims Explanation:**

The paper goes to great length to articulate its demonstrations, and also includes a wealth of experimental evidence to articulate the findings. This also includes selected examples. Overall, I find myself with a generally solid idea of what the paper is trying to achieve.

That being said, the paper is not extremely accessible to someone without the requisite background in Riemannian geometry. I will happily include myself in this category, and invite others to focus on my co-reviewer's assessment more than on mine for this specific point.

**Requested Changes:**

None of the requested changes are critical for my recommendation.

The key addition I would have appreciated would be a more thorough discussion of how to set $\lambda$ on more realistic settings than Fig 3 (table 8 is a start).

Fig 33 & fig 34: missing unit in the y axis.

Table 9 is extremely small.

---

> ### Author Response · Authors · 2026-07-17
>
> We are thankful for the review. In the following, we answer the questions and requested changes.
>
> **Requested changes**
> * We agree to extend the Riemannian manifold background section, so it is more accessible. In the updated version, we have corrected the background section on Riemannian manifolds to make it more accessible.
> * We describe the heuristic method chosen to scale $\lambda$ in Appendix E. With the heuristic scaling proposed in Appendix E, $\lambda$ can be directly interpreted as the weight between likelihood and smoothness. We find that a choice of $\lambda=20.0$ provides a suitable balance between models in practice. In Appendix E, we have added Figure 13 in the updated version of the paper that illustrates the effect on varying $\lambda$. If requested, we are open to adding this to the main paper.
> * Thanks for pointing out the missing unit in Fig. 33 and Fig. 34. We have updated the axis on both figures in the updated version. The unit is seconds.
> * Thanks for pointing out the small text in Table 9. We have modify Table 9 (Tabel 10 in the udpated version) to a sideways-table to make it more readable.
>
> **Additional changes**
> * The original GEORCE-paper, from which our extended algorithm used in the paper is built on, claimed local quadratic convergence. We were recently made aware of an error in their proof. We have corrected this in our paper and instead have shown that ProbGEORCE has locally asymptotic quadratic convergence in the number of grid points under the assumption that $S$ is an affine function. This is in line with the runtime results shown in Table 8 in Appendix G, where the speed-up will occur for smaller values of $\lambda$, and as $\lambda$ increases, the speed-up of our method will decay. We have updated Appendix A.4 with this and also any mention of local quadratic convergence. We apologize for this.

---

### Review · Reviewer_j584 · 2026-07-12

**Summary Of Contributions:**

## Summary

The paper proposes a general geometric framework for computing geometric statistics of generative models. Instead of modeling the data solely as a Riemannian manifold with geodesics, the framework also targets high-density regions learned by the generative model. The problem is formulated as a Newtonian system on a Riemannian manifold. The proposed approach essentially selects geodesics while encouraging them to remain within regions of high likelihood. It characterizes the shortest path as an ODE, which locally corresponds to geodesics along the optimal path. The paper also derives a novel algorithm to efficiently compute interpolation and generalized Fréchet means. Finally, the proposed method is compared against several baselines in terms of negative log-likelihood and runtime.

## Advantages

* This is a general geometric framework that does not restrict the data to lie on a particular manifold.
* Instead of modeling the geometry of the data manifold solely as a Riemannian manifold, the authors introduce a force field that points toward regions of high likelihood, resulting in a Newtonian system on a Riemannian manifold.
* Although solving these ODEs to compute mean and other statistics is computationally expensive, the authors acknowledge this drawback and propose a novel algorithm to compute these statistics more efficiently.
* The comparison with recent baselines strengthens the paper and provides a useful empirical evaluation.

I am putting the disadvantages and question in requested changes section

**Audience:**

Yes

**Audience Explanation:**

Yes the paper actually proposes a new very general framework for computing geometric statistics of generative models which lots of people will find relevant.

**Claims And Evidence:**

Yes

**Claims Explanation:**

Yes the paper make convincing arguments and it's supported by the experiments performed.

**Requested Changes:**

## Disadvantages and Questions

* I have a fundamental question regarding the formulation of the constrained optimization problem in Equation 4. As I understand it, the authors constrain the geodesic such that the overall path cost is upper bounded by a value $\bar{S}$. However, this could still allow portions of the geodesic to pass through low-likelihood regions, provided the start and end points lie in high-likelihood regions. Wouldn't it make more sense to constrain every point along the geodesic to remain within high-likelihood regions? This effect appears to be visible in Figure 2 for the Mean of NF method, where the red (mean) curve for the two-moons distribution passes through a region with very low data likelihood.
* Although the paper's title refers to the "geometry of generative models," the proposed method appears to use only the likelihood function, $S(\cdot)$, provided by the generative model. Since generative models capture much richer structure than likelihood alone, why is utilizing only the likelihood sufficient?
* Do the authors have any intuition or theoretical justification for why this approach performs better than incorporating the probabilistic structure directly into the Riemannian metric? I noticed the brief discussion at the end of Section 2, but I could not find a clear explanation of why this should be expected.
* Why is it important to show that the optimal curve admits a local representation in terms of a Riemannian metric? Please elaborate on why this property is needed and how it benefits the proposed framework.
* Could the authors provide a complexity analysis of the different methods, particularly the BVP approach and the proposed accelerated method? This would help readers better understand the computational improvements.
* The statement preceding Corollary 3.6 requires further clarification: "Some generative models, such as diffusion models, apply a Euclidean background metric of high dimension." It would be helpful to explain what this means and why.

---

> ### Comment · Reviewer_u74y · 2026-07-13
>
> - Re: "[T]his could still allow portions of the geodesic to pass through low-likelihood regions, provided the start and end points lie in high-likelihood regions. Wouldn't it make more sense to constrain every point along the geodesic to remain within high-likelihood regions?". I second this question, but also I guess there is a possibility that the data distribution is multimodal, so maybe constraining every point to have high likelihood is a bit too restrictive? But also, at that point, interpolating might not make sense to start with so I can see having all points within high-likelihood regions makes sense.

---

> ### Author Response · Authors · 2026-07-17
>
> We are thankful for the review. In the following, we answer the questions and requested changes.
>
> **Requested changes**
> * The proposed property is actually already in the current framework. Rather than using $\bar{S}$ in Eq. 4, we use the regularized version in Eq. 5. For a strictly Riemannian metric, the grid points along the curve should have the same distance. However, for the Newtonian system, this does not have to be the case, since the regularization term only depends on the grid points. Therefore, our method can have a longer distance between grid points in low-density areas and a denser number of grid points in high-density regions. For lower values of $\lambda$, smoothness will play a bigger role, while a higher value of $\lambda$ will have higher-likelihood but more rapidly changing curves as seen Fig. 3 for KDE. We have in Appendix E in the updated version added Fig. 13 that shows this effect on the bimodal NF distribution for varying $\lambda$, where the grid points are visible. In practice, we find that increasing $\lambda$ too high will give very sudden jumps in the curves as seen in Fig. 3, which is why we chose $\lambda = 20.0$ to balance smoothness and likelihood.
> * In the paper, we focused on computing high-likelihood statistics on generative models, where we apply the likelihood function as regularization function and the Euclidean background metric. However, the approach is not limited to this. Other constraints can easily be added with their own $\lambda$ parameter, such that the regularization function takes the form
>     %
>     \begin{equation*}
>         S(x) = \bar{\lambda}\sum_{i=1}^{N_\mathrm{constraints}}\lambda_{i}S_{i}(x).
>     \end{equation*}
> * Rather than incorporating the likelihood into the metric, we instead start in a different direction by stating the properties that the connecting curve should have in Eq. 4 and use this to derive the metric. Since, by construction, our method (controlled by $\lambda$) targets high-density regions, we would also expect that this is the case compared to alternative methods. Note also that it is possible within our framework to combine a likelihood-based Riemannian metric with the likelihood regularization term. Also, the varying distance between the grid points controlled by a higher value of $\lambda$ will also ''push'' our curves past lower density regions compared to strictly Riemannian methods as illustrated in Fig. 13 in Appendix E in the updated version of our paper.
> * Proposition 3.2 was derived to understand the behavior of our proposed metric when $\lambda$ becomes large. The Proposition only state the properties of the metric along the optimal curve, which makes it difficult to apply the Proposition in practice. If requested, we are open to removing Proposition 3.2 from the updated version.
> * We agree that adding the complexity of different BVP methods can improve the paper. In general, the computational complexities of solving the BVP are:
>
>   * Gradient descent: $\mathcal{O}\left(N_{\mathrm{grid}}d\right)$
>   * Quasi-Newton method: $\mathcal{O}\left(N_{\mathrm{grid}}^{2}d^{2}\right)$
>   * (Sparse) Newton method: $\mathcal{O}\left(N_{\mathrm{grid}}d^{3}\right)$
>   * ProbGEORCE: $\mathcal{O}\left(N_{\mathrm{grid}}d^{3}\right)$
> where $d$ is the dimension. Note that since the Hessian of the first order approximation of the energy with respect to all grid points is a diagonal is sparse, and that the Hessian of $S$ with respect to all grid points is a diagonal of block matrices, then the complexity of the Newton method is not $\mathcal{O}\left(N_{\mathrm{grid}}^{3}d^{3}\right)$. We have added this to the updated version of the paper in Table 1.
> * The intention behind the statement preceding Corollary 3.6 is that some generative models use the Euclidean background metric, since the data are in a Euclidean space. For diffusion models on images, the stochastic dynamics are mostly defined in the Euclidean space (neglecting the latent representation from the VAE). We have changed this in the updated version.
>
> **Additional changes**
> * The original GEORCE-paper, from which our extended algorithm used in the paper is built on, claimed local quadratic convergence. We were recently made aware of an error in their proof. We have corrected this in our paper and instead have shown that ProbGEORCE has locally asymptotic quadratic convergence in the number of grid points under the assumption that $S$ is an affine function. This is in line with the runtime results shown in Table 8 in Appendix G, where the speed-up will occur for smaller values of $\lambda$, and as $\lambda$ increases, the speed-up of our method will decay. We have updated Appendix A.4 with this and also any mention of local quadratic convergence. We apologize for this.

---

### Author Response · Authors · 2026-07-17

We have updated the paper to address the reviewers' comments and proposed changes. In the most recent version, all changes are highlighted in blue.